# TRAINING-FREE RATE-DISTORTION-PERCEPTION TRAVERSAL WITH DIFFUSION

## ABSTRACT

The rate-distortion-perception (RDP) tradeoff captures the fundamental limits of lossy compression by jointly considering bitrate, reconstruction fidelity, and perceptual quality. While recent neural compression methods have improved perceptual performance, they typically operate at a fixed point on the RDP surface, requiring retraining to target different tradeoffs. In this work, we propose a training-free framework for traversing the full RDP surface, utilizing pretrained diffusion models. Our approach integrates a reverse channel coding (RCC) encoder with a novel score-scaled probability flow ODE decoder. We theoretically prove that the proposed decoder is optimal for the distortion-perception tradeoff under AWGN observations and that the overall framework with the RCC encoder is optimal for the RDP function in the Gaussian case. Empirical results across multiple datasets demonstrate the framework's flexibility and effectiveness in navigating the ternary RDP tradeoff using pre-trained diffusion models. Our results establish a practical and theoretically grounded approach to adaptive, perception-aware compression.

## 1 INTRODUCTION

Lossy compression aims to represent data using the fewest possible bits while preserving acceptable fidelity to the original data. Traditionally, this is formalized by rate-distortion theory, which characterizes the tradeoff between compression rate and data distortion, e.g., mean squared error (MSE). However, distortion-centric metrics often fail in perceptual domains such as image and video compression. This has led to growing interest in the rate-distortion-perception (RDP) tradeoff (Blau & Michaeli, 2019), which incorporates perceptual quality into the classical framework, resulting in a ternary tradeoff that better aligns with the goals of modern compression systems.

Understanding and traversing the RDP surface is crucial for building adaptive and user-controllable compression algorithms. A line of work from the information theory community has established coding theorems for the RDP function under various perceptual constraints (Theis & Wagner, 2021; Chen et al., 2022; Yan et al., 2021; Salehkalaibar et al., 2024; Hamdi et al., 2024). Moreover, the universal RDP function proposed by Zhang et al. (2021) shows that it is possible to fix the encoder and adapt only the decoder to achieve multiple distortion-perception pairs under a given rate.

Despite the theoretical potential, existing neural compression methods fall short of flexibly traversing the RDP tradeoff. Approaches like HiFiC (Mentzer et al., 2020), which employs a generative adversarial network (GAN)-based model optimized over rate, distortion, and perception losses, and Conditional Diffusion Compression (CDC) (Yang & Mandt, 2023), which uses a latent representation and conditional diffusion for reconstruction, operate at fixed tradeoffs, yielding only a single point on the RDP surface per model. While methods such as DiffC (Theis et al., 2022; Vonderfecht & Liu, 2025), Posterior Sampling Compression (PSC) (Elata et al., 2025), and Universally Quantized Diffusion Model (UQDM) (Yang et al., 2025) offer progressive rate control via adaptive sensing or diffusion encoding, they lack mechanisms to navigate the distortion-perception (DP) axis. As a result, no existing approach enables full traversal of the RDP tradeoff using one pre-trained model.

In this work, we propose a training-free framework to traverse the RDP surface based on the DiffC algorithm. Specifically, we utilize the reverse channel coding (RCC) module (Li, 2024; Li & Gamal, 2018) as the encoder, and introduce a flexible decoder powered by pre-trained diffusion models. The framework introduces two intuitive control parameters that steer the ternary tradeoff among rate, distortion, and perception. Our contributions can be summarized as follows:

Table 1: Comparison of related schemes in image restoration and lossy compression problems. Here the rate or DP control means the ability to adjust the rate or DP tradeoff with a single pre-trained model.

| | | Rate Control | DP Control | Proved Optimality in Gaussian |
|---|---|---|---|---|
| Image Restoration | Ohayon et al. (2021) | / | ✓ | ✗ |
| | Wang et al. (2025) | / | ✓ | ✓ |
| | Freirich et al. (2021) | / | ✓ | ✓ |
| | Our ODE decoder | / | ✓ | ✓ |
| Lossy Compression | HiFiC(Mentzer et al., 2020) | ✗ | ✗ | ✗ |
| | CDC(Yang & Mandt, 2023) | ✗ | ✗ | ✗ |
| | DDCM(Ohayon et al., 2025) | ✓ | ✗ | ✗ |
| | DiffC(Theis et al., 2022) | ✓ | ✗ | at perfect realism |
| | Ours | ✓ | ✓ | scalar Gaussian |

- We introduce a novel, training-free framework that enables flexible traversal of the RDP surface using a pre-trained diffusion model based on DiffC. In particular, we propose a novel score-scaled probability flow ODE (PF-ODE) decoder, enabling single-parameter control of the distortion-perception (DP) tradeoff using a single pre-trained diffusion model. The RCC module introduces a second parameter to control the compression rate.

- We derive new theoretical guarantees for the achievability of DP and RDP functions. We prove that the score-scaled PF-ODE is optimal for the DP tradeoff under additive white Gaussian noise (AWGN) observation in multivariate Gaussian cases. The full framework with the RCC encoder achieves the optimal RDP function for scalar Gaussian sources.

- We conduct extensive experiments on CIFAR-10, Kodak, and DIV2K datasets, demonstrating superior flexibility and reconstruction quality across a wide range of rate, distortion, and perception settings. Using a pre-trained diffusion model, our framework enables full RDP traversal via two control parameters.

We present a comparison of related works concerning image restoration and lossy compression problems, focusing on DP and RDP tradeoffs, in Table 1.

**Notations:** Let $X$ be a random variable (r.v.) with distribution $p_X(\mathbf{x})$ over the alphabet $\mathcal{X}$. Realizations are denoted by lowercase letters $\mathbf{x}$. The expectation of $X$ is denoted by $\mathbb{E}[X]$. The covariance between r.v.s $X$ and $Y$ is $\mathrm{Cov}[X, Y]$. Matrices are denoted by bold uppercase letters (e.g., $\boldsymbol{\Sigma}$), with $\mathrm{Tr}(\boldsymbol{\Sigma})$ and $\boldsymbol{\Sigma}^{-1}$ representing the trace and inverse, respectively. $H(X)$ and $I(X; Y)$ denote the Shannon entropy and mutual information.

## 2 BACKGROUND

### 2.1 RATE-DISTORTION-PERCEPTION TRADEOFF

**Information RDP function:** Mathematically, the information RDP function (Blau & Michaeli, 2019) for a source $X$ is defined as

$$R(D, P) = \min_{p_{\hat{X}|X}} I(X; \hat{X}) \quad \text{s.t.} \quad \mathbb{E}[\Delta(X, \hat{X})] \leq D, \quad d(p_X, p_{\hat{X}}) \leq P, \tag{1}$$

where $\Delta : \mathcal{X} \times \hat{\mathcal{X}} \to \mathbb{R}^+$ is a data distortion measure (e.g., square-error), and $d(\cdot, \cdot)$ is a divergence between probability distributions, such as total variation (TV) divergence or Wasserstein-2 (W2) distance (Panaretos & Zemel, 2020). From an information-theoretic perspective, $R(D, P)$ serves as a *lower bound* on the one-shot achievable rate (Theis & Wagner, 2021) when unlimited common randomness is shared between encoder and decoder. Converse and achievability results have been established under various assumptions on shared randomness (Wagner, 2022) and realism constraints (Chen et al., 2022; Hamdi et al., 2024; Salehkalaibar et al., 2024).

Closed-form expressions of Eq. (1) have been derived for binary sources with Hamming distortion and TV divergence (Blau & Michaeli, 2019), as well as scalar Gaussian sources under MSE distortion and W2 distance (Zhang et al., 2021). For multivariate Gaussian cases, Qian et al. (2025)

solved the RDP function via an extended reverse water-filling algorithm. In this paper, we focus on the practical design to traverse the RDP function for general sources.

**Universal RDP function:** Zhang et al. (2021) further reveals the potential to fix an encoder and only adapt the decoder to meet multiple distortion-perception pairs $(D, P) \in \Theta$. For example, $\Theta$ could be the set of all $(D, P)$ pairs associated with a given rate along the information RDP function. Zhang et al. (2021) demonstrated that for the scalar Gaussian distribution, the theoretical rate required by a fixed encoder to achieve all $(D, P)$ pairs in $\Theta$ is exactly $\sup_{(D,P)\in\Theta} R(D, P)$, where $R(D, P)$ is the information RDP function defined in Eq. (1).

**Distortion-perception tradeoff in image restoration:** A related problem to the RDP function is the distortion-perception (DP) tradeoff in image restoration (Blau & Michaeli, 2018; Freirich et al., 2021). Given a noisy observation $Y$ of the source $X$, the DP function is defined as

$$D = \min_{p_{\hat{X}|Y}} \mathbb{E}[\Delta(X, \hat{X})] \quad \text{s.t.} \quad d(p_X, p_{\hat{X}}) \leq P. \tag{2}$$

Note that denoising problem is fundamentally different from lossy compression problem, as the observation is fixed, and there is no rate constraint.

## 2.2 DIFFUSION MODELS AND PROBABILITY FLOW ODE:

Diffusion models (or score-based generative models) define a forward process $(\overrightarrow{Z}_\tau)_{\tau\in[0,T_c]}$ that progressively perturbs data $X \in \mathbb{R}^d$ with Gaussian noise, governed by the stochastic differential equation (SDE) (Song et al., 2021):

$$d\overrightarrow{Z}_\tau = -\frac{1}{2}\beta(\tau)\overrightarrow{Z}_\tau d\tau + \sqrt{\beta(\tau)}dW_\tau, \quad \overrightarrow{Z}_0 = X \sim p_{\text{data}}, \tag{3}$$

where $(W_\tau)_{\tau\in[0,T_c]}$ is the standard Brownian motion and $\beta(\tau)$ is the noise schedule. To generate samples from $p_{\text{data}}$, one can reverse the SDE and discretize the resulting process. According to Anderson (1982) and Song et al. (2021), the reverse SDE associated with Eq. (3) is

$$d\overleftarrow{Z}_\tau = \left[-\frac{1}{2}\beta(\tau)\overleftarrow{Z}_\tau - \beta(\tau)\nabla \log p_{Z_\tau}(\overleftarrow{Z}_\tau)\right]d\tau + \sqrt{\beta(\tau)}d\tilde{W}_\tau, \ \overleftarrow{Z}_{T_c} \sim p_{T_c}, \tag{4}$$

where $(\tilde{W}_\tau)_{\tau\in[0,T_c]}$ is an independent Brownian motion. The score function $\nabla_{\mathbf{z}_\tau} \log p_{Z_\tau}(\mathbf{z}_\tau)$ is approximated by a neural network $s_\theta(\mathbf{z}_\tau, \tau)$ via denoising score matching (Vincent, 2011). The reverse process $\overleftarrow{Z}_\tau$ has the same distribution with $\overrightarrow{Z}_\tau$ for $\tau \in [0, T_c]$.

We can discretize the reverse SDE into $T$ intervals with Euler-Maruyama discretization (Särkkä & Solin, 2019) and index the time sequence as $k \in \{0, 1, \ldots, T\}$. Under the variance-preserving (VP) noise schedule (Ho et al., 2020), the marginals satisfy $p_{Z_k|X}(\mathbf{z}_k|\mathbf{x}) = \mathcal{N}(\sqrt{\bar{\alpha}_k}\mathbf{x}, (1 - \bar{\alpha}_k)\mathbf{I})$, where $\beta_T \geq \cdots \geq \beta_0 = 0$ is the variance schedule, $\alpha_k = 1 - \beta_k$, and $\bar{\alpha}_k = \prod_{i=1}^{k} \alpha_i$. Note that $p_{Z_{k-1}|Z_k}(\mathbf{z}_{k-1}|\mathbf{z}_k)$ can be approximated by learning the score network $s_\theta(\mathbf{z}_k, k)$.

Meanwhile, there exists a deterministic counterpart to the SDE known as the *probability flow ODE (PF-ODE)* (Song et al., 2021), whose trajectories share the same path distribution as the SDE. By manipulating the Fokker-Planck equations (Maoutsa et al., 2020; Särkkä & Solin, 2019), the PF-ODE can be derived as:

$$d\overleftarrow{Z}_\tau = \left[-\frac{1}{2}\beta(\tau)\overleftarrow{Z}_\tau - \frac{1}{2}\beta(\tau)\nabla \log p_{Z_\tau}(\overleftarrow{Z}_\tau)\right]d\tau, \ \overleftarrow{Z}_{T_c} \sim p_{T_c}, \tag{5}$$

which can also be simulated using the same score-based model $s_\theta(\mathbf{z}_\tau, \tau)$.

## 2.3 DIFFC: LOSSY COMPRESSION WITH DIFFUSION MODELS

Towards flexible traversal of the RDP function, Theis et al. (2022) introduced *DiffC*, a novel compression algorithm based on diffusion models. The core idea is to transmit Gaussian-perturbed data and reconstruct it using a pretrained diffusion model. This framework has two key steps: first, the reverse channel coding (RCC) module produces a Gaussian-perturbed sample, distributed according to $p_{Z_t|X}(\cdot|\mathbf{x})$ at the decoder for a given index $t$, where $\mathbf{x}$ represents the data to be transmitted; second, a diffusion-based decoder (e.g., Eq. (5) starting from time index $t$ instead of $T_c$) reconstructs the data from this noisy input. Note that the reconstructions given by DiffC have the same distribution as the source, i.e., the algorithm achieves one extreme of the RDP function where $P = 0$.

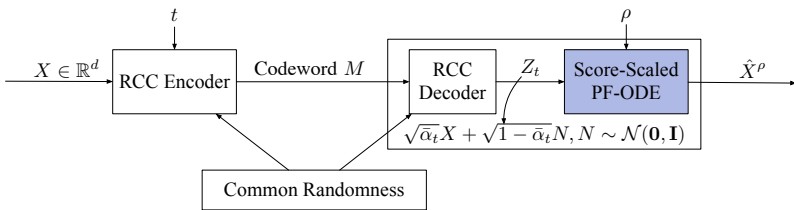

Figure 1: The proposed framework to traverse the RDP function using pre-trained diffusion models.

**Reverse Channel Coding:** One of the key component of the DiffC algorithm is the efficient transmission of a sample drawn from the conditional distribution $p_{Z|X}(\cdot|\mathbf{x})$ given data $\mathbf{x}$, using the shortest possible codeword. This task is known as reverse channel coding (RCC) or the channel simulation problem (Li, 2024). In practical DiffC implementations (Theis et al., 2022; Vonderfecht & Liu, 2025), they adopt the Poisson functional representation (PFR) algorithm from Li & Gamal (2018), which achieves the tightest coding length and can be implemented using CUDA acceleration.

The PFR algorithm enables the encoder to transmit an index $M$ that allows the decoder to draw a sample from the target conditional distribution $p_{Z|X}(\cdot|\mathbf{x})$, using a shared reference distribution $p_Z$, which is the $Z$-marginal of $p_X p_{Z|X}$. Li & Gamal (2018) proved that the entropy of the index satisfies $H(M) \leq I(X;Z) + \log(I(X;Z)+1) + 4$, and this upper bound is achievable via entropy coding $M$ using a Zipf distribution.

In the DiffC framework, the marginal distribution $p_{Z_k}$ is generally unknown. However, the conditional distribution $p_\theta(\mathbf{z}_k|\mathbf{z}_{k+1})$ is available via score matching. Therefore, we can use a *progressive version* of PFR, where an instance $\mathbf{z}_k$ sampled from $p_{Z_k|Z_{k+1},X}(\cdot|\mathbf{z}_{k+1},\mathbf{x})$ is transmitted using $p_\theta(\cdot|\mathbf{z}_{k+1})$ as the reference, for $k = T, T-1, \ldots, 1$. Due to the progressive nature of diffusion models, we can stop at any intermediate index $t \in \{1, 2, \ldots, T\}$ to obtain an instance $\mathbf{z}_t$ following $p_{Z_t|X}(\cdot|\mathbf{x}) = \mathcal{N}(\sqrt{\bar{\alpha}_t}\mathbf{x}, (1-\bar{\alpha}_t)\mathbf{I})$. Details will be presented in Algorithm 1 and 2 in Section 4.

**A Rate-Distortion Analysis of DiffC:** After receiving the noisy representation $Z_t = \sqrt{\bar{\alpha}_t}X + \sqrt{1-\bar{\alpha}_t}N$, where $N \sim \mathcal{N}(0,\mathbf{I})$, the decoder reconstructs the data by simulating the reverse SDE (4) or the PF-ODE (5), starting from time index $t$. Theoretically, the terminal state $Z_0$ has the same distribution as $X$, ensuring reconstructions with *perfect realism*.

Theis et al. (2022) conducted a rate-distortion analysis of the DiffC algorithm, characterizing its performance for Gaussian sources. They showed that under the *perfect realism* constraint, reconstruction given by PF-ODE can provably achieve lower mean squared error (MSE) than SDE when the source distribution satisfies certain regularity conditions.

## 3 DECODER DESIGN: SCORE-SCALED PROBABILITY FLOW ODE

Due to the monotonically increasing variance schedule in diffusion models, the DiffC framework naturally supports progressive coding, enabling flexible control over the compression rate. However, DiffC only offers reconstructions with perfect realism but relatively high distortion, i.e., a single extreme point on the DP curve. To fully traverse the ternary RDP function, we propose a novel decoder design based on a *score-scaled probability flow ODE* that allows flexible traversal along the DP axis for any compression level. The overall framework is illustrated in Figure 1, where the proposed score-scaled PF-ODE is highlighted in blue.

The RCC module generates a specific noisy observation $Z_t = \sqrt{\bar{\alpha}_t}X + \sqrt{1-\bar{\alpha}_t}N$, where $N \sim \mathcal{N}(0,\mathbf{I})$ and $t \in \{1, 2, \ldots, T\}$. Thus, the decoder is effectively tasked with a denoising problem. We show that the proposed method achieves the optimal DP tradeoff under AWGN degradation for multivariate Gaussian sources.

### 3.1 CONDITIONAL AND MARGINAL DISTRIBUTIONS OF SCORE-SCALED PF-ODE

At the perfect realism extreme, the PF-ODE in Eq. (5) yields the best perception with relatively low distortion when the input is corrupted by Gaussian noise (Theis et al., 2022). In the denoising problem, Xue et al. (2024) showed that the reverse mean propagation process by casting out the randomness in a conditional version of Eq. (4) converges to the minimum MSE (MMSE) estimation.

In Appendix B.1, we further show that the mean propagation process of the reverse SDE (4) starting from time index $t$ also converges to the MMSE estimation, which is equivalent to applying a scale on the score term in Eq. (5). Motivated by these observations, we propose the following *score-scaled PF-ODE* to bridge the two extremes:

$$d\overleftarrow{Z}_\tau = \Big[ -\frac{1}{2}\beta(\tau)\overleftarrow{Z}_\tau - \frac{1}{2}(2-\rho)\beta(\tau)\nabla \log p_{Z_t}(\overleftarrow{Z}_\tau)\Big]d\tau, \ \overleftarrow{Z}_t \sim \sqrt{\bar{\alpha}_t}X + \sqrt{1-\bar{\alpha}_t}N, \quad (6)$$

where $\rho \in [0, 1]$. When $\rho = 1$, the above score-scaled PF-ODE collapses to the original PF-ODE (5) and recovers $X$ with perfect realism. When $\rho = 0$, it corresponds to the mean propagation process and converges to the MMSE estimation. By Euler-Maruyama discretization (Särkkä & Solin, 2019) and similar approximations as in Song et al. (2021), we can simulate Eq. (6) by the following iterations

$$Z_k = \frac{1}{\sqrt{1-\beta_{k+1}}}\Big(Z_{k+1} + \frac{1}{2}(2-\rho)\beta_{k+1}\nabla \log p_{Z_{k+1}}(Z_{k+1})\Big), \ k \in \{0, 1, \cdots, t-1\}, \quad (7)$$

starting from an instance of $Z_t \sim \sqrt{\bar{\alpha}_t}X + \sqrt{1-\bar{\alpha}_t}N$. The details are provided in Appendix A.

To understand the distributional behavior of the score-scaled PF-ODE, we first analyze the conditional and marginal distributions at the endpoint of the proposed score-scaled PF-ODE for Gaussian sources. Then we will prove its optimality for the DP tradeoff in multivariate Gaussian in the next subsection.

**Lemma 1.** *Consider the multivariate Gaussian source $X \sim \mathcal{N}(\boldsymbol{\mu}_0, \boldsymbol{\Sigma}_0)$. Let $\boldsymbol{\mu}_k = \sqrt{\bar{\alpha}_k}\boldsymbol{\mu}_0$ and $\boldsymbol{\Sigma}_k = \bar{\alpha}_k\boldsymbol{\Sigma}_0 + (1-\bar{\alpha}_k)\mathbf{I}$ for $k \in \{1, \cdots, t\}$. Starting from $Z_t \sim \sqrt{\bar{\alpha}_t}X + \sqrt{1-\bar{\alpha}_t}N$, $N \sim \mathcal{N}(\mathbf{0}, \mathbf{I})$ and applying the score-scaled PF-ODE iterations in Eq. (7), the* conditional *reconstruction given $Z_t = \check{\mathbf{z}}_t$ is $Z_0(\check{\mathbf{z}}_t) = \mathbf{A}_t^\rho\check{\mathbf{z}}_t + \mathbf{B}_t^\rho\boldsymbol{\mu}_0$, where $\mathbf{A}_t^\rho := \sqrt{\bar{\alpha}_t}\boldsymbol{\Sigma}_0\prod_{i=0}^{t-1}\big(\mathbf{I} + \frac{1}{2}\rho\frac{\beta_{i+1}}{\alpha_{i+1}}\boldsymbol{\Sigma}_i^{-1}\big)\boldsymbol{\Sigma}_t^{-1}$ and $\mathbf{B}_t^\rho := \frac{1}{2}(2-\rho)\big(\sum_{i=2}^t \bar{\alpha}_{i-1}\beta_i\boldsymbol{\Sigma}_0\prod_{j=0}^{i-2}(\mathbf{I} + \frac{1}{2}\rho\frac{\beta_{j+1}}{\alpha_{j+1}}\boldsymbol{\Sigma}_j^{-1})\boldsymbol{\Sigma}_{i-1}^{-1}\boldsymbol{\Sigma}_i^{-1} + \beta_1\boldsymbol{\Sigma}_1^{-1}\big)$. In particular, the reconstruction $Z_0(\check{\mathbf{z}}_t) = \mathbf{A}_t^0\check{\mathbf{z}}_t + \mathbf{B}_t^0\boldsymbol{\mu}_0$ can be proved to be the MMSE point when $\rho = 0$. Then, the marginal distribution of $Z_0$ is given by*

$$Z_0 \sim \mathcal{N}\Big(\boldsymbol{\mu}_0, \ \boldsymbol{\Sigma}_0\prod_{i=0}^{t-1}\big(\rho\mathbf{I} + (1-\rho)\alpha_{i+1}\boldsymbol{\Sigma}_i\boldsymbol{\Sigma}_{i+1}^{-1}\big)\Big).$$

*In particular, when $\rho = 0$, the variance is $\bar{\alpha}_t\boldsymbol{\Sigma}_0^2\boldsymbol{\Sigma}_t^{-1}$. When $\rho = 1$, the marginal distribution of $Z_0$ is $\mathcal{N}(\boldsymbol{\mu}_0, \boldsymbol{\Sigma}_0)$, which coincides with that of the original source $X$.*

*Proof.* See the details in Appendix B. $\qquad\square$

From Lemma 1, we can observe that the reconstruction is deterministic for a fixed $Z_t = \mathbf{z}_t$, with behavior controlled by $\rho$ and $t$. For the marginal distribution, the mean remains fixed at $\boldsymbol{\mu}_0$, regardless of $\rho$ and $t$. When $\rho$ increases, the variance of $Z_0$ gradually approaches that of the original source $X$, leading to improved perception but increased distortion.

## 3.2 OPTIMAL DISTORTION-PERCEPTION TRADEOFF THROUGH AWGN CHANNEL

For each compression level $t$, the decoder receives $Z_t = \sqrt{\bar{\alpha}_t}X + \sqrt{1-\bar{\alpha}_t}N$, $N \sim \mathcal{N}(\mathbf{0}, \mathbf{I})$. Mathematically, the DP tradeoff with MSE distortion and Wasserstein-2 distance is defined as (Blau & Michaeli, 2018)

$$D = \min_{p_{\hat{X}|Z_t}} \mathbb{E}[\|X - \hat{X}\|_2^2], \quad \text{s.t.} \quad W_2^2(p_X, p_{\hat{X}}) \leq P, \quad (8)$$

where $W_2^2(\cdot, \cdot)$ denotes the squared Wasserstein-2 distance between two distributions. In Proposition 2, we first derive the optimal solution to Eq. (8) for the multivariate Gaussian case. Then, we show that our score-scaled PF-ODE achieves this optimal DP tradeoff for multivariate Gaussian sources under AWGN.

**Proposition 2** (Freirich et al. (2021)). *Consider the $d$-dimensional source $X \sim \mathcal{N}(\boldsymbol{\mu}_0, \boldsymbol{\Sigma}_0)$ and the additive white Gaussian noise (AWGN) observation $Z_t \sim \sqrt{\bar{\alpha}_t}X + \sqrt{1-\bar{\alpha}_t}N$, $N \sim \mathcal{N}(\mathbf{0}, \mathbf{I})$.*

*Assume $\mathbf{\Sigma}_0$ admits an eigen-decomposition $\mathbf{\Sigma}_0 = \mathbf{Q}\mathbf{\Lambda}_0\mathbf{Q}^\top$ with positive eigenvalues $\mathbf{\Lambda}_0 = \mathrm{diag}(\lambda_0, \cdots, \lambda_d)$. Then the optimal solution to Eq. (8) is*

$$D_t = \left( \sqrt{\sum_{i=1}^d \frac{\lambda_i}{\lambda_i^{(t)}} \left( \sqrt{\lambda_i^{(t)}} - \sqrt{\bar{\alpha}_t}\sqrt{\lambda_i} \right)^2} - \sqrt{P} \right)^2 + \sum_{\ell=1}^d \frac{(1-\bar{\alpha}_t)\lambda_\ell}{\lambda_\ell^{(t)}}, \tag{9}$$

*where $\lambda_\ell^{(t)} := \bar{\alpha}_t\lambda_\ell + (1-\bar{\alpha}_t)$.*

*Proof.* In Freirich et al. (2021), the authors derived the optimal DP tradeoff and the optimal estimators for multivariate Gaussian sources. Our results are equivalent to theirs in the case of AWGN degradation. The detailed closed-form in Eq. (9) and Lemma 7 in our proof offer insights into the achievability of our proposed method (Theorem 3). Therefore, we include the proof in Appendix C.2 for completeness. □

**Theorem 3.** *The optimal tradeoff (9) can be achieved by simulating the score-scaled PF-ODE iterations in Eq. (7), with a dedicated choice of $\rho$ for each dimension of $Z_t$.*

*Proof.* In the achievability proof, we use a per-dimension variant of Eq. (7), which applies independently to each component of $Z_t$. See details in Appendix C.3. □

**Remark 1.** *Although Freirich et al. (2021) have already derived the optimal DP tradeoff for the multivariate Gaussian case, they assumed the availability of two extreme point estimators (i.e., MMSE and perfect realism) and then constructed intermediate estimators via linear interpolation. In contrast, our achievability proof is constructive, as the score-scaled PF-ODE naturally provides a concrete scheme to achieve the two extremes of the DP tradeoff and offers flexible control over it.*

*Our main goal is to construct a flexible lossy compression scheme. Leveraging the use of RCC, which produces a special noisy observation $Z_t = \sqrt{\bar{\alpha}_t}X + \sqrt{1-\bar{\alpha}_t}N$, our score-scaled ODE focuses on a special case of the denoising problem across various noise levels $t$. The progressive structure of the proposed ODE and the introduction of $\rho$ successfully enable traversing the distortion-perception tradeoff for any noise level $t$ using a single pre-trained model.*

## 4 TRAVERSING RDP FUNCTION: OPTIMALITY AND GENERAL ALGORITHM

In this section, we jointly analyze the RDP performance when combining the RCC encoder with the proposed score-scaled PF-ODE. We prove that the proposed scheme is an optimal solution to the RDP function for scalar Gaussian sources. Then, we detail the practical algorithms applicable to general sources, enabling flexible traversal of the RDP surface using a pre-trained diffusion model.

Suppose we can transmit an instance of $Z_t \sim \sqrt{\bar{\alpha}_t}X + \sqrt{1-\bar{\alpha}_t}N$, $N \sim \mathcal{N}(\mathbf{0}, \mathbf{I})$ using the RCC algorithm (e.g., PFR) for different compression level $t$. Then the decoder can simulate the score-scaled PF-ODE iterations (7) with $\rho \in [0,1]$ to determine the balance between distortion and perception under each compression rate.

The following theorem shows that, despite its modularity and flexibility, our scheme achieves the information RDP function (1) in the scalar Gaussian case.

**Theorem 4.** *Consider the scalar Gaussian source $X \sim \mathcal{N}(\mu_0, \sigma_0^2)$. The information rate-distortion-perception function under MSE distortion and squared W2 distance is given by*

$$R(D, P) = \begin{cases} \frac{1}{2}\log \frac{\sigma_0^2(\sigma_0-\sqrt{P})^2}{\sigma_0^2(\sigma_0-\sqrt{P})^2-(\sigma_0^2+(\sigma_0-\sqrt{P})^2-D)^2/4} & \text{if } \sqrt{P} < \sigma_0 - \sqrt{|\sigma_0 - D|}, \\ \max\{\frac{1}{2}\log\frac{\sigma_0^2}{D}, 0\} & \text{if } \sqrt{P} \geq \sigma_0 - \sqrt{|\sigma_0 - D|}. \end{cases}$$

*Let $D_t^\rho$ and $P_t^\rho$ be the squared error distortion and squared W2 distance achieved by the score-scaled PF-ODE iterations (7) for $\rho \in [0,1]$ at compression level $t \in \{1, 2, \cdots, T\}$. Then we have*

$$R_t^1 \leq R(D_t^\rho, P_t^\rho) \leq R_t^1 + \log\left(R_t^1 + 1\right) + 4,$$
$$R_t^\infty = R(D_t^\rho, P_t^\rho),$$

*where $R_t^1$ and $R_t^\infty$ are the one-shot and asymptotic coding rate of the Poisson functional representation (Li & Gamal, 2018) to transmit samples of $Z_t \sim \sqrt{\bar{\alpha}_t}X + \sqrt{1-\bar{\alpha}_t}N$.*

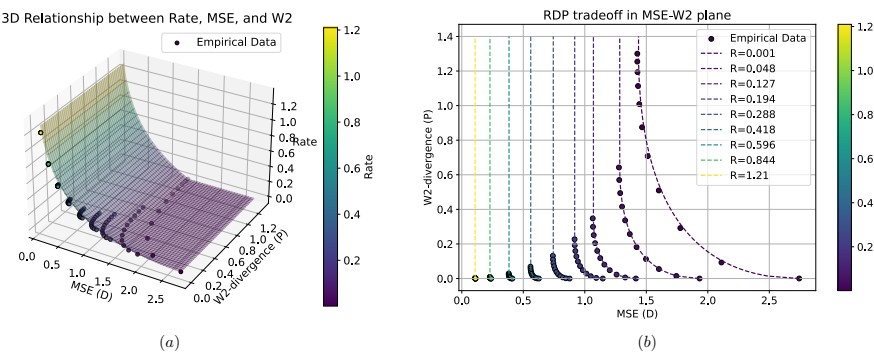

Figure 2: Information-theoretical RDP function for scalar Gaussian source (dashed line) and achieved rate, MSE, and W2 distance levels by our scheme (solid dots). (a) The RDP surface. (b) $R(D, P)$ function along DP planes. Different colors represent different rates.

*Proof.* The closed-form expression of the RDP function is derived in Zhang et al. (2021). In Appendix D, we show that the proposed score-scaled PF-ODE combined with the PFR encoder achieves $(R, D, P)$ triplets that asymptotically match the theoretical RDP function. □

Figure 2 visualizes Theorem 4 by plotting the theoretical RDP surface along with empirical $(R, D, P)$ points achieved by our method. For illustration, we omit entropy coding and directly compute the theoretical coding rate given by the RCC algorithm. We can observe that the empirical DP points generated by iteratively simulating Eq. (7) with different $\rho$ and $t$ align well with the theoretical RDP curves.

**Remark 2.** *Note that the proposed scheme inherits the "universal" property discussed in Zhang et al. (2021). Given all $(D, P)$ pairs associated with a fixed rate (e.g., one dashed line in Figure 2(b)), we can fix the time index $t$ of the encoder and adapt only the score-scaling parameter $\rho$ in the decoder to meet all these DP constraints.*

In Theorem 4, we prove the optimality of our scheme only for the scalar Gaussian case. For multivariate Gaussian sources, the information RDP function is formulated as an optimization problem and solved by a generalized reverse water-filling method (Qian et al., 2025). No explicit solution has been derived for more complicated sources. Advanced theoretical analysis of our scheme for such distributions remains future work. In Section 5, we will demonstrate the empirical effectiveness of our method on real-world datasets.

We now detail practical algorithms for applying our method to general high-dimensional sources using pre-trained diffusion models in Algorithm 1 and 2. As discussed in Section 2.3, directly transmitting $Z_t \sim \sqrt{\bar{\alpha}_t} X + \sqrt{1 - \bar{\alpha}_t} N$, $N \sim \mathcal{N}(\mathbf{0}, \mathbf{I})$ via RCC is challenging due to the inaccessibility of $p_{Z_t}$ for complicated source $X$. Instead, we approximate the conditional distributions $p_{Z_k | Z_{k+1}}$ using the pre-trained diffusion model and progressively transmit $Z_k$ conditioned on $Z_{k+1}$ and $X$.

| **Algorithm 1** Encoder | **Algorithm 2** Decoder |
|---|---|
| 1: **Input:** Pre-trained model $p_\theta(\mathbf{z}_k \mid \mathbf{z}_{k+1})$, target time index $t \in \{1, 2, \cdots, T\}$, source data $\mathbf{x}$ | 1: **Input:** Pre-trained model $p_\theta(\mathbf{z}_k \mid \mathbf{z}_{k+1})$, $t \in \{1, 2, \cdots, T\}$, ODE parameter $\rho$ |
| 2: $\mathbf{z}_T \leftarrow$ An instance following $p_{Z_T} = \mathcal{N}(\mathbf{0}, \mathbf{I})$ | 2: $\mathbf{z}_T \leftarrow$ An instance following $p_{Z_T} = \mathcal{N}(\mathbf{0}, \mathbf{I})$ ▷ Using shared seed |
| 3: **for** $k = T-1, \ldots, t-1, t$ **do** | 3: **for** $k = T-1, \ldots, t-1, t$ **do** |
| 4: ▷ Send $\mathbf{z}_k \sim p_{Z_k \mid Z_{k+1}, X}(\cdot \mid \mathbf{z}_{k+1}, \mathbf{x})$ using $p_\theta(\mathbf{z}_k \mid \mathbf{z}_{k+1})$. Here we use PFR as an illustration. | 4: $\bar{\mathbf{z}}_k^{(1)}, \bar{\mathbf{z}}_k^{(2)}, \cdots \sim p_\theta(\mathbf{z}_k \mid \mathbf{z}_{k+1})$ ▷ Using shared seed |
| 5: $\quad W_1, W_2, \cdots \text{Exp}(1)$, $S_n = \sum_{i=1}^n W_i$ | 5: $\quad \mathbf{z}_k = \bar{\mathbf{z}}_k^{(c_k)}$ after receiving $c_k$ |
| 6: $\quad \bar{\mathbf{z}}_k^{(1)}, \bar{\mathbf{z}}_k^{(2)}, \cdots \sim p_\theta(\mathbf{z}_k \mid \mathbf{z}_{k+1})$ | 6: **end for** |
| 7: $\quad c_k = \arg\min_{n \in \mathbb{N}_+} S_n \frac{p_\theta(\bar{\mathbf{z}}_k^{(n)} \mid \mathbf{z}_{k+1})}{p_{Z_k \mid Z_{k+1}, X}(\bar{\mathbf{z}}_k^{(n)} \mid \bar{\mathbf{z}}_{k+1}^{(n)}, \mathbf{x})}$ | 7: $\hat{\mathbf{x}}_t^\rho \leftarrow$ Simulate Eq. (7) given $Z_t = \mathbf{z}_t$ with chosen $\rho$. |
| 8: $\quad$ **Output:** Entropy code and send $c_k$ to Decoder. | 8: **Output:** $\hat{\mathbf{x}}_t^\rho$ |
| 9: **end for** | |

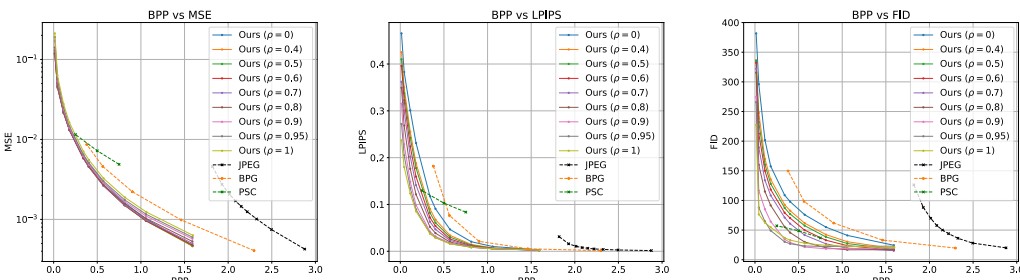

Figure 3: Effect of controlling $t$ and $\rho$ on different metrics for the CIFAR-10 dataset. Distortion is quantified by MSE, and perception is measured by LPIPS and FID.

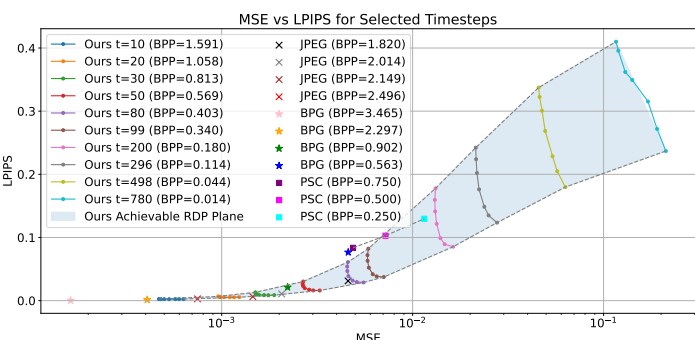

Figure 4: Rate-distortion-perception curves on the CIFAR-10 dataset. Distortion levels are quantified by MSE and perception levels are measured by LPIPS.

Note that we use the PFR algorithm as an illustration in Algorithms 1 and 2. Any other RCC algorithms (including sample-based methods and dithered quantization) can be employed, as long as samples following the distribution of $\sqrt{\bar{\alpha}_t}X + \sqrt{1 - \bar{\alpha}_t}N$ can be generated or approximated.

## 5 EXPERIMENTAL RESULTS

In this section, we demonstrate the flexibility and effectiveness of our proposed framework through experiments conducted on high-dimensional, real-world datasets.

### 5.1 CIFAR-10 DATASET

We begin with the CIFAR-10 dataset to validate our theoretical findings and illustrate the effectiveness of adjusting both the time index $t$ in the PFR algorithm and the score-scaling parameter $\rho$ in the proposed PF-ODE iterations (7). We directly employ the pre-trained diffusion model from a third-party repository[1], which is a PyTorch implementation following the details in Ho et al. (2020).

We benchmark our method against two traditional codecs, JPEG and BPG, and a posterior sampling-based diffusion compression method, PSC (Elata et al., 2025). PSC utilizes adaptive compressed sensing with a pre-trained diffusion model for flexible-rate compression. Notably, both our method and PSC share the same diffusion model backbone, ensuring a fair comparison.

Figure 3 illustrates the impact of varying $t$ and $\rho$ on distortion and perception metrics. As expected, for a fixed $t$, increasing $\rho$ improves perceptual metrics (lower LPIPS and FID) at the expense of higher distortion (increased MSE), aligning with our theoretical predictions. Meanwhile, decreasing $t$ yields a higher bit rate (BPP), leading to improvements in both distortion and perception metrics. Our scheme achieves lower distortion and superior perceptual quality compared to JPEG, BPG, and PSC at comparable bitrates.

---

[1] https://github.com/w86763777/pytorch-ddpm

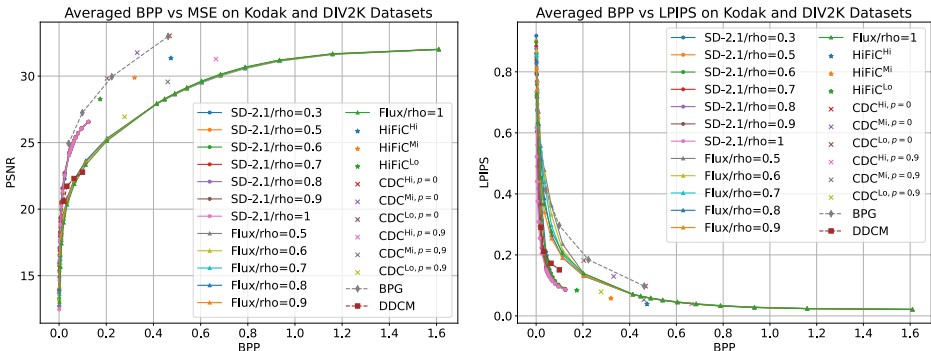

Figure 5: Effect of controlling $t$ and $\rho$ on different metrics for the Kodak and DIV2K datasets. Stable Diffusion 2.1 and Flux depict different rate-distortion (R-D) and rate-perception (R-P) curves.

We also plot the RDP curves in Figure 4 by varying $t$ and $\rho$. The RDP curves are constructed by connecting the points with the same $t$ but different $\rho$. The curves are convex and shift down-left as $t$ decreases, indicating improved tradeoffs at higher compression rates. These observations confirm our theoretical analysis. Our scheme provides *full control* over rate, distortion, and perception using a single pre-trained model, while PSC only supports adaptive rate control. Experimental details and additional results, as well as visual illustrations, are provided in Appendix E.1.

## 5.2 KODAK AND DIV2K DATASETS

In this subsection, we further evaluate our method on the Kodak and DIV2K datasets (Agustsson & Timofte, 2017). The Kodak dataset comprises 24 high-quality $768 \times 512$ images, while the DIV2K validation set contains 100 high-resolution images. We combine both datasets to form our test set.

We employ diverse open-source diffusion models, including multiple versions of Stable Diffusion (SD) (Rombach et al., 2022) and Flux (Black-Forest-Labs et al., 2025). Note that both Stable Diffusion and Flux are *latent* diffusion models, operating in the latent space of a pre-trained autoencoder. Although our theoretical results are derived for the original source space, experimental results demonstrate that, even in the latent space, our scheme can still effectively and flexibly traverse the RDP plane by controlling the time index $t$ and score-scaling parameter $\rho$ in the score-scaled PF-ODE (6). We adopt the CUDA-accelerated PFR implementation from Vonderfecht & Liu (2025).

We compare our method against several baselines. For HiFiC (Mentzer et al., 2020), we use three pre-trained models provided by their official repository: HiFiC$^{Lo}$, HiFiC$^{Mi}$, and HiFiC$^{Hi}$, targeting different compression rates. For CDC (Yang & Mandt, 2023), we include six models trained by the authors: three focused on distortion (CDC$^{Lo,p=0}$, CDC$^{Mi,p=0}$, CDC$^{Hi,p=0}$) and three on perception (CDC$^{Lo,p=0.9}$, CDC$^{Mi,p=0.9}$, CDC$^{Hi,p=0.9}$). DDCM (Ohayon et al., 2025) also use pre-trained diffusion models for image compression. We use their official implementation and the same Stable Diffusion 2.1 for comparison. We include BPG as a comparison with classical compression methods. *Note that HiFiC and CDC achieve only a single point on the RDP surface per model.*

Figure 5 shows the PSNR and LPIPS results across different $\rho$ and $t$ values for both SD 2.1 and Flux. While SD 2.1 performs better at low bitrates, Flux supports broader RDP traversal at higher bitrates. In both cases, increasing $\rho$ improves perception but leads to worse distortion, consistent with theoretical predictions. While HiFiC and CDC occasionally outperform in one metric due to their targeted training losses, their models may underperform on other metrics and lack flexibility.

Figure 6 visualizes the RDP tradeoff traversed by our proposed scheme on the Kodak and DIV2K datasets. As $t$ increases, the RDP curves shift downward and shrink, indicating that DP tradeoffs are more pronounced at lower bitrates. At higher bitrates, reconstructions approach the original images, achieving both low distortion and high perceptual quality. Note that, because the diffusion process is applied in the latent space, the supported $\rho$ ranges may differ from $[0, 1]$ in Theorem 3.

We include the detailed choices of $\rho$ in Appendix E.2.1 and more experimental results, including more metrics (e.g., MSE-FID tradeoff) and other pre-trained models (e.g., SD1.5 and SDXL), in Appendix E.2.2. We also report the model sizes and coding latencies in Appendix E.2.2.

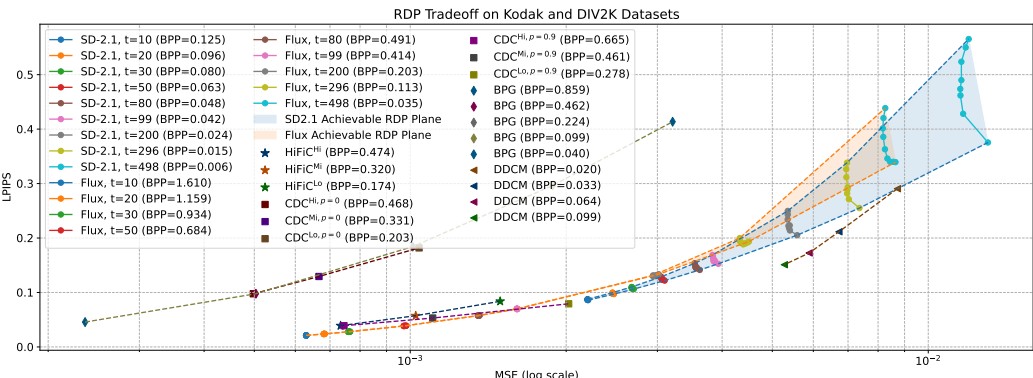

Figure 6: RDP tradeoff traversed by our proposed scheme on the Kodak and DIV2K datasets. We show the results obtained with Stable Diffusion (SD) 2.1 and the Flux model, respectively.

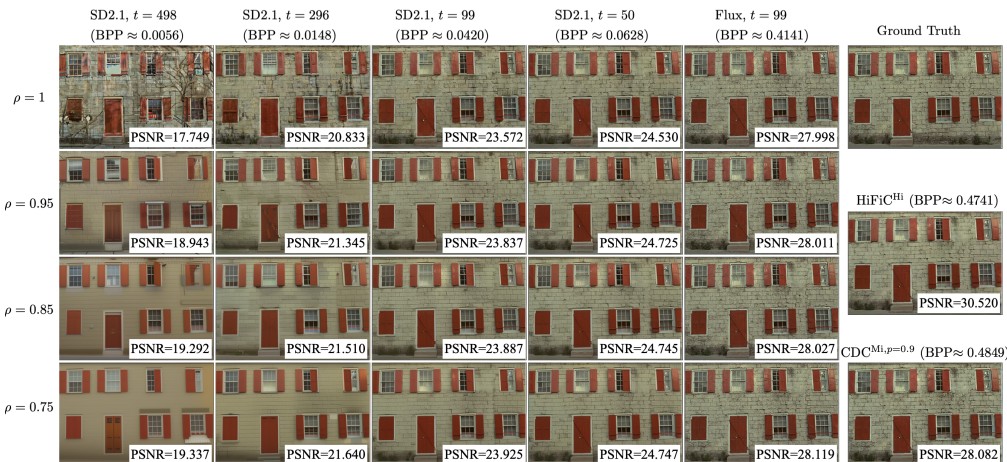

Figure 7: Samples from HiFiC, CDC, and our proposed schemes with different $\rho$ and $t$.

We also provide visual examples under various settings in Figure 7 to illustrate the RDP tradeoff. In the low BPP regime, a high $\rho$ yields perceptually pleasing images with vivid colors and sharp edges; however, the details may be unfaithful to the original source. Lowering $\rho$ suppresses hallucinated details, producing reconstructions that are blurrier but exhibit reduced distortion. *At higher bitrates*, reconstructions become both sharp and faithful across all $\rho$ values. More samples can be found in Appendix E.2.2. We also provide a demo website[2] to compare the details of *high-resolution images* for different $t$ and $\rho$. Our method provides an efficient way to control RDP tradeoff with one pre-trained model and two parameters $t$ and $\rho$. In practice, users can select a suitable bitrate according to resource restrictions and adjust the DP balance according to their specific needs without retraining.

## 6 CONCLUSIONS

In this paper, we introduced a training-free framework for traversing the complete RDP tradeoff in lossy compression, leveraging the RCC encoder and the proposed score-scaled PF-ODE decoder. Our theoretical analysis establishes the optimality of the decoder for DP tradeoffs under AWGN observation in multivariate Gaussian cases, and of the full framework for the RDP function in scalar Gaussian settings. Empirical evaluations on CIFAR-10, Kodak, and DIV2K datasets validate the method's ability to flexibly and effectively balance bitrate, distortion, and perceptual quality using pre-trained diffusion models. This work offers a practical and theoretically principled solution for adaptive, high-fidelity compression across the entire RDP surface, obviating the need for retraining.

---

[2]https://diffrdp.github.io/

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

# APPENDIX

## A   DISCRETIZATION OF SCORE-SCALED PF-ODE

Consider the following time-reversed SDE starting from a particular time index $t \in [0, T_c]$, which is the proposed score-scaled PF-ODE in Eq. (6) with an additional scaled Brownian motion:

$$d\overleftarrow{Z}_\tau = \Big[ -\frac{1}{2}\beta(\tau)\overleftarrow{Z}_\tau - \frac{1}{2}(2-\rho)\beta(t)\nabla \log p_{Z_\tau}(\overleftarrow{Z}_\tau) \Big] d\tau + \lambda dW_\tau, \quad \overleftarrow{Z}_t \sim \sqrt{\bar{\alpha}_t}X + \sqrt{1-\bar{\alpha}_t}N. \tag{10}$$

To analyze the probabilistic behavior of the proposed score-scaled PF-ODE in Eq. (6), we can study the above SDE at $\lambda \to 0$. By Euler-Maruyama discretization (Särkkä & Solin, 2019), we can divide the whole time interval $[0, T_c]$ into $T$ equal segments with step size $\Delta\tau = \frac{T_c}{T}, \tau = k\Delta\tau$. The noise schedule in the discrete time space is given by $\beta_k = \beta(k\Delta\tau)\Delta\tau$ for $k = 0, 1, \cdots, T$. Following the similar approximation with (Song et al., 2021, Appendix E), the discretized version of the above SDE can be written as

$$Z_k = (2 - \sqrt{1-\beta_{k+1}})Z_{k+1} + \frac{1}{2}(2-\rho)\beta_{k+1}\nabla \log p_{Z_{k+1}}(Z_{k+1}) + \lambda\sqrt{\beta_{k+1}}\epsilon$$

$$\overset{(a)}{\approx} \Big(2 - (1 - \frac{1}{2}\beta_{k+1})\Big)Z_{k+1} + \frac{1}{2}(2-\rho)\beta_{k+1}\nabla \log p_{Z_{k+1}}(Z_{k+1}) + \lambda\sqrt{\beta_{k+1}}\epsilon$$

$$\overset{(b)}{\approx} (1 + \frac{1}{2}\beta_{k+1})Z_{k+1} + \frac{1}{2}(2-\rho)\beta_{k+1}\nabla \log p_{Z_{k+1}}(Z_{k+1}) + \frac{1}{2}(2-\rho)\beta_{k+1}^2\nabla \log p_{Z_{k+1}}(Z_{k+1}) + \lambda\sqrt{\beta_{k+1}}\epsilon$$

$$= (1 + \frac{1}{2}\beta_{k+1})\Big(Z_{k+1} + \frac{1}{2}(2-\rho)\beta_{k+1}\nabla \log p_{Z_{k+1}}(Z_{k+1})\Big) + \lambda\sqrt{\beta_{k+1}}\epsilon$$

$$\overset{(c)}{\approx} \frac{1}{\sqrt{1-\beta_{k+1}}}\Big(Z_{k+1} + \frac{1}{2}(2-\rho)\beta_{k+1}\nabla \log p_{Z_{k+1}}(Z_{k+1})\Big) + \lambda\sqrt{\beta_{k+1}}\epsilon,$$

where $(a)$ and $(c)$ are due to the equivalent infinitesimal $\sqrt{1-\beta_k} \approx 1 - \frac{1}{2}\beta_k$ and $\frac{1}{\sqrt{1-\beta_k}} \approx 1 + \frac{1}{2}\beta_k$, and $(b)$ is given by eliminating the $\mathcal{O}(\beta_k)$ term when $\Delta\tau \to 0$. Here $\epsilon \sim \mathcal{N}(\mathbf{0}, \mathbf{I})$.

## B   PROOF OF LEMMA 1

Consider the source $X \sim \mathcal{N}(\boldsymbol{\mu}_0, \boldsymbol{\Sigma}_0)$, and the decoder receives $Z_t = \sqrt{\bar{\alpha}_t}X + \sqrt{1-\bar{\alpha}_t}N$, $N \sim \mathcal{N}(\mathbf{0}, \mathbf{I})$, which has distribution $p_{Z_t}(\mathbf{z}_t) = \mathcal{N}(\sqrt{\bar{\alpha}_t}\boldsymbol{\mu}_0, \bar{\alpha}_t\boldsymbol{\Sigma}_0 + (1-\bar{\alpha}_t)\mathbf{I})$. Denote the marginal distribution of $Z_k$ for $k = t, t-1, \cdots, 0$ as

$$p_{Z_k}(\mathbf{z}_k) = \mathcal{N}(\underbrace{\sqrt{\bar{\alpha}_k}\boldsymbol{\mu}_0}_{\boldsymbol{\mu}_k}, \underbrace{\bar{\alpha}_k\boldsymbol{\Sigma}_0 + (1-\bar{\alpha}_k)\mathbf{I}}_{\boldsymbol{\Sigma}_k}),$$

then $\nabla_{\mathbf{z}_k} \log p_{Z_k}(\mathbf{z}_k) = \boldsymbol{\Sigma}_k^{-1}(\boldsymbol{\mu}_k - \mathbf{z}_k)$. For $k = t-1, \ldots, 0$, the discretized score-scaled PF-ODE provides us with

$$\mathbf{z}_k = \frac{1}{\sqrt{1-\beta_{k+1}}}\Big(\mathbf{z}_{k+1} + \frac{1}{2}(2-\rho)\beta_{k+1}\nabla_{\mathbf{z}_{k+1}} \log p_{Z_{k+1}}(\mathbf{z}_{k+1})\Big) + \lambda\sqrt{\beta_{k+1}}\epsilon$$

$$= \frac{1}{\sqrt{1-\beta_{k+1}}}\Big(\mathbf{z}_{k+1} + \frac{1}{2}(2-\rho)\beta_{k+1}\boldsymbol{\Sigma}_{k+1}^{-1}(\boldsymbol{\mu}_{k+1} - \mathbf{z}_{k+1})\Big) + \lambda\sqrt{\beta_{k+1}}\epsilon$$

$$= \frac{1}{\sqrt{\alpha_{k+1}}}\Big(\boldsymbol{\Sigma}_{k+1} - \frac{1}{2}(2-\rho)\beta_{k+1}\mathbf{I}\Big)\boldsymbol{\Sigma}_{k+1}^{-1}\mathbf{z}_{k+1} + \frac{1}{\sqrt{\alpha_{k+1}}}\frac{1}{2}(2-\rho)\beta_{k+1}\boldsymbol{\Sigma}_{k+1}^{-1}\boldsymbol{\mu}_{k+1} + \lambda\sqrt{\beta_{k+1}}\epsilon$$

$$= \underbrace{\sqrt{\alpha_{k+1}}\Big(\boldsymbol{\Sigma}_k + \frac{1}{2}\rho\frac{\beta_{k+1}}{\alpha_{k+1}}\mathbf{I}\Big)\boldsymbol{\Sigma}_{k+1}^{-1}}_{\mathbf{U}_{k+1}^\rho}\mathbf{z}_{k+1} + \underbrace{\frac{1}{2}(2-\rho)\beta_{k+1}\sqrt{\bar{\alpha}_k}\boldsymbol{\Sigma}_{k+1}^{-1}}_{\mathbf{V}_{k+1}^\rho}\boldsymbol{\mu}_0 + \lambda\sqrt{\beta_{k+1}}\epsilon,$$

which provide us with the following relationship between $Z_k$ and $Z_{k+1}$:

$$p_{Z_k|Z_{k+1}}(\mathbf{z}_k|\mathbf{z}_{k+1}) = \mathcal{N}(\mathbf{U}_{k+1}^\rho \mathbf{z}_{k+1} + \mathbf{V}_{k+1}^\rho \boldsymbol{\mu}_0, \lambda\beta_{k+1}\mathbf{I}),$$

where

$$\mathbf{U}_{k+1}^\rho := \sqrt{\alpha_{k+1}}\Big(\mathbf{\Sigma}_k + \frac{1}{2}\rho\frac{\beta_{k+1}}{\alpha_{k+1}}\mathbf{I}\Big)\mathbf{\Sigma}_{k+1}^{-1},$$

$$\mathbf{V}_{k+1}^\rho := \frac{1}{2}(2-\rho)\beta_{k+1}\sqrt{\bar\alpha_k}\mathbf{\Sigma}_{k+1}^{-1},$$

for $k = t, t-1, \ldots, 0$.

### B.1 Conditional Distributions of $Z_0$ given $Z_t = \check{\mathbf{z}}_t$

**Lemma 5.** *([Bishop](), [2006](), Section 2.3.3) Given a marginal Gaussian distribution for $X$ and a conditional Gaussian distribution for $Y$ given $X$ in the form*

$$p_X(\mathbf{x}) = \mathcal{N}\left(\boldsymbol{\mu}, \mathbf{\Lambda}^{-1}\right),$$

$$p_Y(\mathbf{y} \mid \mathbf{x}) = \mathcal{N}\left(\mathbf{A}\mathbf{x} + \mathbf{b}, \mathbf{L}^{-1}\right),$$

*the marginal distribution of $Y$ and the conditional distribution of $X$ given $Y$ are given by*

$$p_Y(\mathbf{y}) = \mathcal{N}\left(\mathbf{A}\boldsymbol{\mu} + \mathbf{b}, \mathbf{L}^{-1} + \mathbf{A}\mathbf{\Lambda}^{-1}\mathbf{A}^\top\right)$$

$$p_{X|Y}(\mathbf{x} \mid \mathbf{y}) = \mathcal{N}\left(\mathbf{\Sigma}\left\{\mathbf{A}^\top\mathbf{L}(\mathbf{y} - \mathbf{b}) + \mathbf{\Lambda}\boldsymbol{\mu}\right\}, \mathbf{\Sigma}\right),$$

*where*

$$\mathbf{\Sigma} = \left(\mathbf{\Lambda} + \mathbf{A}^\top\mathbf{L}\mathbf{A}\right)^{-1}.$$

Starting from a sample $Z_t = \check{\mathbf{z}}_t$, we perform the reverse process and compute the conditional distribution of the reconstruction. Considering the above Lemma 5, together with $p_{Z_{t-1}|Z_t}(\mathbf{z}_{t-1}|\check{\mathbf{z}}_t) = \mathcal{N}(\mathbf{U}_t^\rho\check{\mathbf{z}}_t + \mathbf{V}_t^\rho\boldsymbol{\mu}_0, \lambda\beta_t\mathbf{I})$ and $p_{Z_{t-2}|Z_{t-1}}(\mathbf{z}_{t-2}|\mathbf{z}_{t-1}) = \mathcal{N}(\mathbf{U}_{t-1}^\rho\mathbf{z}_{t-1} + \mathbf{V}_{t-1}^\rho\boldsymbol{\mu}_0, \lambda\beta_{t-1}\mathbf{I})$, we have

$$p_{Z_{t-2}|Z_t}(\mathbf{z}_{t-2}|\check{\mathbf{z}}_t) = \mathcal{N}\Big(\mathbf{U}_{t-1}^\rho\big(\mathbf{U}_t^\rho\check{\mathbf{z}}_t + \mathbf{V}_t^\rho\boldsymbol{\mu}_0\big) + \mathbf{V}_{t-1}^\rho\boldsymbol{\mu}_0, \ \lambda\beta_{t-1}\mathbf{I} + \mathbf{U}_{t-1}^\rho\lambda\beta_t\mathbf{U}_{t-1}^{\rho\top}\Big)$$

$$= \mathcal{N}\Big(\mathbf{U}_{t-1}^\rho\mathbf{U}_t^\rho\check{\mathbf{z}}_t + \big(\mathbf{U}_{t-1}^\rho\mathbf{V}_t^\rho + \mathbf{V}_{t-1}^\rho\big)\boldsymbol{\mu}_0, \ \lambda\beta_{t-1}\mathbf{I} + \mathbf{U}_{t-1}^\rho\lambda\beta_t\mathbf{U}_{t-1}^{\rho\top}\Big),$$

where

$$\mathbf{U}_{t-1}^\rho\mathbf{U}_t^\rho = \sqrt{\alpha_{t-1}}\Big(\mathbf{\Sigma}_{t-2} + \frac{1}{2}\rho\frac{\beta_{t-1}}{\alpha_{t-1}}\mathbf{I}\Big)\mathbf{\Sigma}_{t-1}^{-1}\cdot\sqrt{\alpha_t}\Big(\mathbf{\Sigma}_{t-1} + \frac{1}{2}\rho\frac{\beta_t}{\alpha_t}\mathbf{I}\Big)\mathbf{\Sigma}_t^{-1}$$

$$= \sqrt{\alpha_{t-1}}\sqrt{\alpha_t}\Big(\mathbf{\Sigma}_{t-2} + \frac{1}{2}\rho\frac{\beta_{t-1}}{\alpha_{t-1}}\mathbf{I}\Big)\Big(\mathbf{I} + \frac{1}{2}\rho\frac{\beta_t}{\alpha_t}\mathbf{\Sigma}_{t-1}^{-1}\Big)\mathbf{\Sigma}_t^{-1},$$

and

$$\big(\mathbf{U}_{t-1}^\rho\mathbf{V}_t^\rho + \mathbf{V}_{t-1}^\rho\big) = \sqrt{\alpha_{t-1}}\Big(\mathbf{\Sigma}_{t-2} + \frac{1}{2}\rho\frac{\beta_{t-1}}{\alpha_{t-1}}\mathbf{I}\Big)\mathbf{\Sigma}_{t-1}^{-1}\cdot\frac{1}{2}(2-\rho)\beta_t\sqrt{\bar\alpha_{t-1}}\mathbf{\Sigma}_t^{-1} + \frac{1}{2}(2-\rho)\beta_{t-1}\sqrt{\bar\alpha_{t-2}}\mathbf{\Sigma}_{t-1}^{-1}$$

$$= \frac{1}{2}(2-\rho)\alpha_{t-1}\sqrt{\bar\alpha_{t-2}}\beta_t\Big(\mathbf{\Sigma}_{t-2} + \frac{1}{2}\rho\frac{\beta_{t-1}}{\alpha_{t-1}}\mathbf{I}\Big)\mathbf{\Sigma}_{t-1}^{-1}\mathbf{\Sigma}_t^{-1} + \frac{1}{2}(2-\rho)\beta_{t-1}\sqrt{\bar\alpha_{t-2}}\mathbf{\Sigma}_{t-1}^{-1}.$$

Now we prove the general case by induction. Suppose that in step $k$, $p_{Z_k|Z_t}(\mathbf{z}_k|\check{\mathbf{z}}_t)$ has a Gaussian distribution with mean

$$\mathbf{U}_{k+1}^\rho\mathbf{U}_{k+2}^\rho\cdots\mathbf{U}_t^\rho\check{\mathbf{z}}_t + \Big(\mathbf{U}_{k+1}^\rho\cdots\big(\mathbf{U}_{t-1}^\rho\mathbf{V}_t^\rho + \mathbf{V}_{t-1}^\rho\big)\cdots + \mathbf{V}_{k+1}^\rho\Big)\boldsymbol{\mu}_0,$$

and variance

$$\lambda\sum_{i=k+1}^t\beta_i\Big(\prod_{j=k+1}^{i-1}\mathbf{U}_j^\rho\Big)\Big(\prod_{j=k+1}^{i-1}\mathbf{U}_j^\rho\Big)^\top,$$

where

$$\mathbf{U}_{k+1}^\rho\mathbf{U}_{k+2}^\rho\cdots\mathbf{U}_t^\rho = \Big(\prod_{i=k+1}^t\sqrt{\alpha_i}\Big)\big(\mathbf{\Sigma}_k + \frac{1}{2}\rho\frac{\beta_{k+1}}{\alpha_{k+1}}\mathbf{I}\big)\prod_{i=k+1}^{t-1}\big(\mathbf{I} + \frac{1}{2}\rho\frac{\beta_{i+1}}{\alpha_{i+1}}\mathbf{\Sigma}_i^{-1}\big)\mathbf{\Sigma}_t^{-1},$$

and

$$\Big(\mathbf{U}_{k+1}^{\rho}\cdots(\mathbf{U}_{t-1}^{\rho}\mathbf{V}_t^{\rho}+\mathbf{V}_{t-1}^{\rho})\cdots+\mathbf{V}_{k+1}^{\rho}\Big)$$

$$=\frac{1}{2}(2-\rho)\sqrt{\bar{\alpha}_k}\Big(\sum_{i=k+2}^{t}(\prod_{j=k+1}^{i-1}\alpha_j)\beta_i(\mathbf{\Sigma}_k+\frac{1}{2}\rho\frac{\beta_{k+1}}{\alpha_{k+1}}\mathbf{I})(\prod_{j=k+1}^{i-2}(\mathbf{I}+\frac{1}{2}\rho\frac{\beta_{j+1}}{\alpha_{j+1}}\mathbf{\Sigma}_j^{-1}))\mathbf{\Sigma}_{i-1}^{-1}\mathbf{\Sigma}_i^{-1}+\beta_{k+1}\mathbf{\Sigma}_{k+1}^{-1}\Big).$$

With $p_{Z_{k-1}|Z_k}(\mathbf{z}_{k-1}|\mathbf{z}_k)=\mathcal{N}(\mathbf{U}_k^{\rho}\mathbf{z}_k+\mathbf{V}_k^{\rho}\boldsymbol{\mu}_0,\ \lambda\beta_k\mathbf{I})$, we can deduce that $p_{Z_{k-1}|Z_t}(\mathbf{z}_{k-1}|\check{\mathbf{z}}_t)$ has mean

$$\mathbf{U}_k^{\rho}\Big(\mathbf{U}_{k+1}^{\rho}\mathbf{U}_{k+2}^{\rho}\cdots\mathbf{U}_t^{\rho}\check{\mathbf{z}}_t+\Big(\mathbf{U}_{k+1}^{\rho}\cdots(\mathbf{U}_{t-1}^{\rho}\mathbf{V}_t^{\rho}+\mathbf{V}_{t-1}^{\rho})\cdots+\mathbf{V}_{k+1}^{\rho}\Big)\boldsymbol{\mu}_0\Big)+\mathbf{V}_k^{\rho}\boldsymbol{\mu}_0$$

$$=\mathbf{U}_k^{\rho}\mathbf{U}_{k+1}^{\rho}\mathbf{U}_{k+2}^{\rho}\cdots\mathbf{U}_t^{\rho}\check{\mathbf{z}}_t+\Big(\mathbf{U}_k^{\rho}\Big(\mathbf{U}_{k+1}^{\rho}\cdots(\mathbf{U}_{t-1}^{\rho}\mathbf{V}_t^{\rho}+\mathbf{V}_{t-1}^{\rho})\cdots+\mathbf{V}_{k+1}^{\rho}\Big)+\mathbf{V}_k^{\rho}\Big)\boldsymbol{\mu}_0,$$

and variance

$$\lambda\beta_k+\mathbf{U}_k^{\rho}\lambda\sum_{i=k+1}^{t}\beta_i\Big(\prod_{j=k+1}^{i-1}\mathbf{U}_j^{\rho}\Big)\Big(\prod_{j=k+1}^{i-1}\mathbf{U}_j^{\rho}\Big)^{\top}\mathbf{U}_k^{\rho\top}=\lambda\sum_{i=k}^{t}\beta_i\Big(\prod_{j=k}^{i-1}\mathbf{U}_j^{\rho}\Big)\Big(\prod_{j=k}^{i-1}\mathbf{U}_j^{\rho}\Big)^{\top},$$

where

$$\mathbf{U}_k^{\rho}\mathbf{U}_{k+1}^{\rho}\mathbf{U}_{k+2}^{\rho}\cdots\mathbf{U}_t^{\rho}$$

$$=\sqrt{\alpha_k}\Big(\mathbf{\Sigma}_{k-1}+\frac{1}{2}\rho\frac{\beta_k}{\alpha_k}\mathbf{I}\Big)\mathbf{\Sigma}_k^{-1}\cdot\Big(\prod_{i=k+1}^{t}\sqrt{\alpha_i}\Big)(\mathbf{\Sigma}_k+\frac{1}{2}\rho\frac{\beta_{k+1}}{\alpha_{k+1}}\mathbf{I})\prod_{i=k+1}^{t-1}(\mathbf{I}+\frac{1}{2}\rho\frac{\beta_{i+1}}{\alpha_{i+1}}\mathbf{\Sigma}_i^{-1})\mathbf{\Sigma}_t^{-1}$$

$$=\Big(\prod_{i=k}^{t}\sqrt{\alpha_i}\Big)\Big(\mathbf{\Sigma}_{k-1}+\frac{1}{2}\rho\frac{\beta_k}{\alpha_k}\mathbf{I}\Big)\Big(\mathbf{I}+\frac{1}{2}\rho\frac{\beta_{k+1}}{\alpha_{k+1}}\mathbf{\Sigma}_k^{-1}\Big)\prod_{i=k+1}^{t-1}(\mathbf{I}+\frac{1}{2}\rho\frac{\beta_{i+1}}{\alpha_{i+1}}\mathbf{\Sigma}_i^{-1})\mathbf{\Sigma}_t^{-1}$$

$$=\Big(\prod_{i=k}^{t}\sqrt{\alpha_i}\Big)\Big(\mathbf{\Sigma}_{k-1}+\frac{1}{2}\rho\frac{\beta_k}{\alpha_k}\mathbf{I}\Big)\prod_{i=k}^{t-1}(\mathbf{I}+\frac{1}{2}\rho\frac{\beta_{i+1}}{\alpha_{i+1}}\mathbf{\Sigma}_i^{-1})\mathbf{\Sigma}_t^{-1},$$

and

$$\mathbf{U}_k^{\rho}\Big(\mathbf{U}_{k+1}^{\rho}\cdots(\mathbf{U}_{t-1}^{\rho}\mathbf{V}_t^{\rho}+\mathbf{V}_{t-1}^{\rho})\cdots+\mathbf{V}_{k+1}^{\rho}\Big)+\mathbf{V}_k^{\rho}$$

$$=\frac{1}{2}(2-\rho)\sqrt{\bar{\alpha}_k}\Big(\sum_{i=k+1}^{t}\Big(\prod_{j=k}^{i-1}\alpha_j\Big)\beta_i\Big(\mathbf{\Sigma}_{k-1}+\frac{1}{2}\rho\frac{\beta_k}{\alpha_k}\mathbf{I}\Big)\Big(\prod_{j=k}^{i-2}(\mathbf{I}+\frac{1}{2}\rho\frac{\beta_{j+1}}{\alpha_{j+1}}\mathbf{\Sigma}_j^{-1})\Big)\mathbf{\Sigma}_{i-1}^{-1}\mathbf{\Sigma}_i^{-1}+\beta_k\mathbf{\Sigma}_k^{-1}\Big).$$

At the last step, the reconstruction $p_{Z_0|Z_t}(\mathbf{z}_0|\check{\mathbf{z}}_t)$ has Gaussian distribution with mean

$$\mathbf{U}_1^{\rho}\mathbf{U}_2^{\rho}\cdots\mathbf{U}_t^{\rho}\check{\mathbf{z}}_t+\Big(\mathbf{U}_1^{\rho}\Big(\mathbf{U}_2^{\rho}\cdots(\mathbf{U}_{t-1}^{\rho}\mathbf{V}_t^{\rho}+\mathbf{V}_{t-1}^{\rho})\cdots+\mathbf{V}_2^{\rho}\Big)+\mathbf{V}_1^{\rho}\Big)\boldsymbol{\mu}_0=\mathbf{A}_t^{\rho}\check{\mathbf{z}}_t+\mathbf{B}_t^{\rho}\boldsymbol{\mu}_0,$$

$$(11)$$

and variance

$$\lambda\mathbf{\Lambda}_t^{\rho}:=\lambda\sum_{i=1}^{t}\beta_i\Big(\prod_{j=1}^{i-1}\mathbf{U}_{t-1}^{\rho}\Big)\Big(\prod_{j=1}^{i-1}\mathbf{U}_{t-1}^{\rho}\Big)^{\top},$$

where

$$\mathbf{A}_t^{\rho}=\sqrt{\bar{\alpha}_t}\mathbf{\Sigma}_0\prod_{i=0}^{t-1}(\mathbf{I}+\frac{1}{2}\rho\frac{\beta_{i+1}}{\alpha_{i+1}}\mathbf{\Sigma}_i^{-1})\mathbf{\Sigma}_t^{-1},$$

and

$$\mathbf{B}_t^{\rho}=\frac{1}{2}(2-\rho)\Big(\sum_{i=2}^{t}\bar{\alpha}_{i-1}\beta_i\mathbf{\Sigma}_0\prod_{j=0}^{i-2}(\mathbf{I}+\frac{1}{2}\rho\frac{\beta_{j+1}}{\alpha_{j+1}}\mathbf{\Sigma}_j^{-1})\mathbf{\Sigma}_{i-1}^{-1}\mathbf{\Sigma}_i^{-1}+\beta_1\mathbf{\Sigma}_1^{-1}\Big).$$

In general, when $\lambda = 0$, the variance will be zero, regardless of the value of $\rho$, and the reconstruction is given by the mean in Eq. (11).

**At the point of** $\rho = 0$, $\mathbf{B}_t$ has a simplified expression,

$$\mathbf{B}_t = \sum_{i=k+1}^{t} \bar{\alpha}_{i-1}\beta_i \boldsymbol{\Sigma}_0 \boldsymbol{\Sigma}_{i-1}^{-1}\boldsymbol{\Sigma}_i^{-1} + \beta_1\boldsymbol{\Sigma}_1^{-1} = (1-\bar{\alpha}_t)\boldsymbol{\Sigma}_t^{-1},$$

which leads to the mean of $p_{Z_0|Z_t}(\mathbf{z}_0|\mathbf{z}_t)$ being

$$\hat{\boldsymbol{\mu}}_0(\check{\mathbf{z}}_t) = \sqrt{\bar{\alpha}_t}\boldsymbol{\Sigma}_0\boldsymbol{\Sigma}_t^{-1}\check{\mathbf{z}}_t + (1-\bar{\alpha}_t)\boldsymbol{\Sigma}_t^{-1}\boldsymbol{\mu}_0.$$

By Tweedie's formula (Robbins, 1956) and the relationship $Z_t = \sqrt{\bar{\alpha}_t}X + \sqrt{1-\bar{\alpha}_t}\epsilon$, we have

$$\mathbb{E}[X|Z_t = \check{\mathbf{z}}_t] = \frac{1}{\sqrt{\bar{\alpha}_t}}(\check{\mathbf{z}}_t + (1-\bar{\alpha}_t)\nabla_{\mathbf{z}_t}\log p_{Z_t}(\check{\mathbf{z}}_t))$$

$$= \frac{1}{\sqrt{\bar{\alpha}_t}}(\check{\mathbf{z}}_t + (1-\bar{\alpha}_t)\boldsymbol{\Sigma}_t^{-1}(\boldsymbol{\mu}_t - \check{\mathbf{z}}_t))$$

$$= \frac{1}{\sqrt{\bar{\alpha}_t}}(\boldsymbol{\Sigma}_t - (1-\bar{\alpha}_t)\mathbf{I})\boldsymbol{\Sigma}_t^{-1}\check{\mathbf{z}}_t + \frac{1}{\sqrt{\bar{\alpha}_t}}(1-\bar{\alpha}_t)\boldsymbol{\Sigma}_t^{-1}\boldsymbol{\mu}_t$$

$$= \sqrt{\bar{\alpha}_t}\boldsymbol{\Sigma}_0\boldsymbol{\Sigma}_t^{-1}\check{\mathbf{z}}_t + (1-\bar{\alpha}_t)\boldsymbol{\Sigma}_t^{-1}\boldsymbol{\mu}_0 = \hat{\boldsymbol{\mu}}_0(\check{\mathbf{z}}_t),$$

which implies that the reverse sampling starting with zero variance from step $\mathbf{z}_t$ can reach the MMSE $\mathbb{E}[X|Z_t]$.

## B.2 AN ALTERNATIVE EXPRESSION OF $\mathbf{A}_t^\rho$ AND $\mathbf{B}_t^\rho$

Since $\boldsymbol{\Sigma}_k$ is a covariance matrix, it is symmetric and positive semi-definite. Further suppose $\boldsymbol{\Sigma}_k$ is invertible. Then, For any $\boldsymbol{\Sigma}_i$ and $\boldsymbol{\Sigma}_j$, we have

$$\boldsymbol{\Sigma}_i\boldsymbol{\Sigma}_j = (\bar{\alpha}_i\boldsymbol{\Sigma}_0 + (1-\bar{\alpha}_i)\mathbf{I})(\bar{\alpha}_j\boldsymbol{\Sigma}_0 + (1-\bar{\alpha}_j)\mathbf{I})$$

$$= \bar{\alpha}_i\bar{\alpha}_j\boldsymbol{\Sigma}_0^2 + (\bar{\alpha}_i + \bar{\alpha}_j - 2\bar{\alpha}_i\bar{\alpha}_j)\boldsymbol{\Sigma}_0 + (1-\bar{\alpha}_i)(1-\bar{\alpha}_j)\mathbf{I},$$

which is symmetric. Thus, $\boldsymbol{\Sigma}_i\boldsymbol{\Sigma}_j = (\boldsymbol{\Sigma}_i\boldsymbol{\Sigma}_j)^\top = \boldsymbol{\Sigma}_j\boldsymbol{\Sigma}_i$, which implies that $\boldsymbol{\Sigma}_i$ and $\boldsymbol{\Sigma}_j$ commute with each other.

**Lemma 6.** *(Friedberg et al., 2003, Section 5.2) Two diagonalizable matrices $\mathbf{A}$ and $\mathbf{B}$ commute if and only if they can be simultaneously diagonalized, i.e., there exists a nonsingular matrix $\mathbf{P}$ such that both $\mathbf{PAP}^{-1}$ and $\mathbf{PBP}^{-1}$ are diagonal.*

First, we show an approximation which will be used frequently in the following proof. For $i \in \{1, \ldots, T\}$, we have

$$\left(\alpha_{i+1}\boldsymbol{\Sigma}_i + \frac{\rho}{2}\beta_{i+1}\mathbf{I}\right)^2$$

$$= \left(\alpha_{i+1}(\bar{\alpha}_i\boldsymbol{\Sigma}_0 + (1-\bar{\alpha}_i)\mathbf{I}) + \frac{\rho}{2}\beta_{i+1}\mathbf{I}\right)^2$$

$$= \bar{\alpha}_{i+1}\bar{\alpha}_{i+1}\boldsymbol{\Sigma}_0\boldsymbol{\Sigma}_0 + 2\bar{\alpha}_{i+1}(\alpha_{i+1} - \bar{\alpha}_{i+1} + \frac{\rho}{2}\beta_{i+1})\boldsymbol{\Sigma}_0 + (\alpha_{i+1} - \bar{\alpha}_{i+1} + \frac{\rho}{2}\beta_{i+1})^2\mathbf{I}$$

$$= \bar{\alpha}_{i+1}\bar{\alpha}_{i+1}\boldsymbol{\Sigma}_0\boldsymbol{\Sigma}_0 + \bar{\alpha}_{i+1}(2\alpha_{i+1} - 2\bar{\alpha}_{i+1} + \rho\beta_{i+1})\boldsymbol{\Sigma}_0 + (\alpha_{i+1} - \bar{\alpha}_{i+1} + \rho\beta_{i+1})(\alpha_{i+1} - \bar{\alpha}_{i+1})\mathbf{I} + \frac{\rho}{4}\beta_{i+1}^2\mathbf{I}$$

$$= \alpha_{i+1}\left(\bar{\alpha}_i\bar{\alpha}_{i+1}\boldsymbol{\Sigma}_0\boldsymbol{\Sigma}_0 + ((2-\rho)\bar{\alpha}_{i+1} + \rho\bar{\alpha}_i - 2\bar{\alpha}_{i+1}\bar{\alpha}_i)\boldsymbol{\Sigma}_0 + ((1-\rho)\alpha_{i+1} - \bar{\alpha}_{i+1} + \rho)(1-\bar{\alpha}_i)\mathbf{I}\right) + \mathcal{O}(\beta_{i+1}^2)$$

$$\overset{(a)}{\approx} \alpha_{i+1}\left(\bar{\alpha}_i\boldsymbol{\Sigma}_0 + (1-\bar{\alpha}_i)\mathbf{I}\right)\left(\bar{\alpha}_{i+1}\boldsymbol{\Sigma}_0 + (\rho + (1-\rho)\alpha_{i+1} - \bar{\alpha}_{i+1})\mathbf{I}\right)$$

$$= \alpha_{i+1}\boldsymbol{\Sigma}_i(\rho\boldsymbol{\Sigma}_{i+1} + (1-\rho)\alpha_{i+1}\boldsymbol{\Sigma}_i),$$

where $(a)$ is approximated by neglecting the $\mathcal{O}(\beta_{i+1}^2)$ term. Furthermore, since $\boldsymbol{\Sigma}_i$ and $\rho\boldsymbol{\Sigma}_{i+1} + (1-\rho)\alpha_{i+1}\boldsymbol{\Sigma}_i$ commute, together with Lemma 6, we have

$$\alpha_{i+1}\boldsymbol{\Sigma}_i + \frac{\rho}{2}\beta_{i+1}\mathbf{I} \approx \sqrt{\alpha_{i+1}}\boldsymbol{\Sigma}_i^{\frac{1}{2}}(\rho\boldsymbol{\Sigma}_{i+1} + (1-\rho)\alpha_{i+1}\boldsymbol{\Sigma}_i)^{\frac{1}{2}}. \tag{12}$$

In particular, $\alpha_{i+1}\boldsymbol{\Sigma}_i + \frac{1}{2}\beta_{i+1}\mathbf{I} \approx \sqrt{\alpha_{i+1}}\boldsymbol{\Sigma}_i^{\frac{1}{2}}\boldsymbol{\Sigma}_{i+1}^{\frac{1}{2}}$.

Then, we can derive the alternative expression of $\mathbf{A}_t^\rho$ and $\mathbf{B}_t^\rho$. For $t \in \{1, \ldots, T\}$, we have

$$\mathbf{A}_t^\rho = \sqrt{\bar{\alpha}_t}\boldsymbol{\Sigma}_0 \prod_{i=0}^{t-1}\big(\mathbf{I} + \frac{1}{2}\rho\frac{\beta_{i+1}}{\alpha_{i+1}}\boldsymbol{\Sigma}_i^{-1}\big)\boldsymbol{\Sigma}_t^{-1}$$

$$= \sqrt{\bar{\alpha}_t}\boldsymbol{\Sigma}_0 \prod_{i=0}^{t-1}\big((\alpha_{i+1}\boldsymbol{\Sigma}_i + \frac{\rho}{2}\beta_{i+1}\mathbf{I})\frac{1}{\alpha_{i+1}}\boldsymbol{\Sigma}_i^{-1}\big)\boldsymbol{\Sigma}_t^{-1}$$

$$\approx \sqrt{\bar{\alpha}_t}\boldsymbol{\Sigma}_0 \prod_{i=0}^{t-1}\big(\frac{\sqrt{\alpha_{i+1}}}{\alpha_{i+1}}\boldsymbol{\Sigma}_i^{\frac{1}{2}}(\rho\boldsymbol{\Sigma}_{i+1} + (1-\rho)\alpha_{i+1}\boldsymbol{\Sigma}_i)^{\frac{1}{2}}\boldsymbol{\Sigma}_i^{-1}\big)\boldsymbol{\Sigma}_t^{-1} \quad \text{[By Eq. (12)]}$$

$$= \sqrt{\bar{\alpha}_t}\boldsymbol{\Sigma}_0^{\frac{1}{2}}\frac{1}{\sqrt{\alpha_1}}\boldsymbol{\Sigma}_0^{-\frac{1}{2}}(\rho\boldsymbol{\Sigma}_1 + (1-\rho)\alpha_{i+1}\boldsymbol{\Sigma}_0)^{\frac{1}{2}}\cdots\frac{1}{\sqrt{\alpha_t}}\boldsymbol{\Sigma}_{t-1}^{-\frac{1}{2}}(\rho\boldsymbol{\Sigma}_1 + (1-\rho)\alpha_t\boldsymbol{\Sigma}_{t-1})^{\frac{1}{2}}\boldsymbol{\Sigma}_t^{-1}$$

$$= \boldsymbol{\Sigma}_0^{\frac{1}{2}}\prod_{i=0}^{t-1}\big(\rho\mathbf{I} + (1-\rho)\alpha_{i+1}\boldsymbol{\Sigma}_i\boldsymbol{\Sigma}_{i+1}^{-1}\big)^{\frac{1}{2}}\boldsymbol{\Sigma}_t^{-\frac{1}{2}}, \tag{13}$$

and

$$\mathbf{B}_t^\rho = \frac{1}{2}(2-\rho)\bigg(\sum_{i=2}^{t}\bar{\alpha}_{i-1}\beta_i\boldsymbol{\Sigma}_0\prod_{j=0}^{i-2}\big(\mathbf{I} + \frac{1}{2}\rho\frac{\beta_{j+1}}{\alpha_{j+1}}\boldsymbol{\Sigma}_j^{-1}\big)\boldsymbol{\Sigma}_{i-1}^{-1}\boldsymbol{\Sigma}_i^{-1} + \beta_1\boldsymbol{\Sigma}_1^{-1}\bigg)$$

$$\approx \frac{1}{2}(2-\rho)\bigg(\sum_{i=2}^{t}\bar{\alpha}_{i-1}\beta_i\frac{1}{\sqrt{\bar{\alpha}_{i-1}}}\boldsymbol{\Sigma}_0^{\frac{1}{2}}\prod_{j=0}^{i-2}\big(\rho\mathbf{I} + (1-\rho)\alpha_{j+1}\boldsymbol{\Sigma}_j\boldsymbol{\Sigma}_{j+1}^{-1}\big)^{\frac{1}{2}}\boldsymbol{\Sigma}_{i-1}^{-\frac{1}{2}}\boldsymbol{\Sigma}_i^{-1} + \beta_1\boldsymbol{\Sigma}_1^{-1}\bigg) \quad \text{[By Eq. (13)]}$$

$$\overset{(a)}{=} (2-\rho)\bigg(\mathbf{I} - \sqrt{\bar{\alpha}_t}\boldsymbol{\Sigma}_0^{\frac{1}{2}}\boldsymbol{\Sigma}_t^{-\frac{1}{2}} - \sum_{i=1}^{t-1}\frac{1}{2}\sqrt{\bar{\alpha}_i}\beta_{i+1}\boldsymbol{\Sigma}_0^{\frac{1}{2}}\big(\mathbf{I} - \prod_{j=0}^{i-1}\big(\rho\mathbf{I} + (1-\rho)\alpha_{j+1}\boldsymbol{\Sigma}_j\boldsymbol{\Sigma}_{j+1}^{-1}\big)^{\frac{1}{2}}\big)\boldsymbol{\Sigma}_{i+1}^{-1}\boldsymbol{\Sigma}_i^{-\frac{1}{2}}\bigg), \tag{14}$$

where (a) can be proved by induction: First,

$$\mathbf{B}_1^\rho = \frac{1}{2}(2-\rho)\beta_1\boldsymbol{\Sigma}_1^{-1} = (2-\rho)\big(\alpha_1\boldsymbol{\Sigma}_0 - \alpha\boldsymbol{\Sigma}_0 + (1-\alpha_1)\mathbf{I} - \frac{1}{2}\beta_1\mathbf{I}\big)\boldsymbol{\Sigma}_1^{-1}$$

$$= (2-\rho)\big(\boldsymbol{\Sigma}_1 - \sqrt{\alpha_1}\boldsymbol{\Sigma}_0^{\frac{1}{2}}\boldsymbol{\Sigma}_1^{\frac{1}{2}}\big)\boldsymbol{\Sigma}_1^{-1} = (2-\rho)\big(\mathbf{I} + \sqrt{\alpha_1}\boldsymbol{\Sigma}_0^{\frac{1}{2}}\boldsymbol{\Sigma}_1^{\frac{1}{2}}\big).$$

Suppose we have

$$\mathbf{B}_{t-1}^\rho = (2-\rho)\bigg(\mathbf{I} - \sqrt{\bar{\alpha}_{t-1}}\boldsymbol{\Sigma}_0^{\frac{1}{2}}\boldsymbol{\Sigma}_{t-1}^{-\frac{1}{2}} - \underbrace{\sum_{i=1}^{t-2}\frac{1}{2}\sqrt{\bar{\alpha}_i}\beta_{i+1}\boldsymbol{\Sigma}_0^{\frac{1}{2}}\big(\mathbf{I} - \prod_{j=0}^{i-1}(\rho\mathbf{I} + (1-\rho)\alpha_{j+1}\boldsymbol{\Sigma}_j\boldsymbol{\Sigma}_{j+1}^{-1})^{\frac{1}{2}}\big)\boldsymbol{\Sigma}_{i+1}^{-1}\boldsymbol{\Sigma}_i^{-\frac{1}{2}}}_{:=\mathbf{C}_i^\rho}\bigg).$$

Then, by induction, we have

$$\mathbf{B}_t^\rho = \mathbf{B}_{t-1}^\rho + \frac{1}{2}(2-\rho)\sqrt{\bar{\alpha}_{t-1}}\beta_t\boldsymbol{\Sigma}_0^{\frac{1}{2}}\prod_{j=0}^{t-2}\big(\rho\mathbf{I} + (1-\rho)\alpha_{j+1}\boldsymbol{\Sigma}_j\boldsymbol{\Sigma}_{j+1}^{-1}\big)^{\frac{1}{2}}\boldsymbol{\Sigma}_t^{-1}\boldsymbol{\Sigma}_{t-1}^{-\frac{1}{2}}$$

$$= (2-\rho)\bigg(\mathbf{I} - \sqrt{\bar{\alpha}_{t-1}}\boldsymbol{\Sigma}_0^{\frac{1}{2}}\boldsymbol{\Sigma}_{t-1}^{-\frac{1}{2}} - \sum_{i=1}^{t-2}\mathbf{C}_i^\rho + \frac{1}{2}\sqrt{\bar{\alpha}_{t-1}}\beta_t\boldsymbol{\Sigma}_0^{\frac{1}{2}}\prod_{j=0}^{t-2}\big(\rho\mathbf{I} + (1-\rho)\alpha_{j+1}\boldsymbol{\Sigma}_j\boldsymbol{\Sigma}_{j+1}^{-1}\big)^{\frac{1}{2}}\boldsymbol{\Sigma}_t^{-1}\boldsymbol{\Sigma}_{t-1}^{-\frac{1}{2}}\bigg)$$

$$= (2-\rho)\bigg(\mathbf{I} - \sqrt{\bar{\alpha}_{t-1}}\boldsymbol{\Sigma}_0^{\frac{1}{2}}\big(\boldsymbol{\Sigma}_t - \beta_t\frac{1}{2}\prod_{j=0}^{t-2}\big(\rho\mathbf{I} + (1-\rho)\alpha_{j+1}\boldsymbol{\Sigma}_j\boldsymbol{\Sigma}_{j+1}^{-1}\big)^{\frac{1}{2}}\big)\boldsymbol{\Sigma}_t^{-1}\boldsymbol{\Sigma}_{t-1}^{-\frac{1}{2}} - \sum_{i=1}^{t-2}\mathbf{C}_i^\rho\bigg)$$

$$\overset{(b)}{\approx} (2-\rho)\bigg(\mathbf{I} - \sqrt{\bar{\alpha}_{t-1}}\boldsymbol{\Sigma}_0^{\frac{1}{2}}\big(\sqrt{\alpha_t}\boldsymbol{\Sigma}_{t-1}^{\frac{1}{2}}\boldsymbol{\Sigma}_t^{\frac{1}{2}} + \frac{1}{2}\beta_t\big(\mathbf{I} - \prod_{j=0}^{t-2}\big(\rho\mathbf{I} + (1-\rho)\alpha_{j+1}\boldsymbol{\Sigma}_j\boldsymbol{\Sigma}_{j+1}^{-1}\big)^{\frac{1}{2}}\big)\big)\boldsymbol{\Sigma}_t^{-1}\boldsymbol{\Sigma}_{t-1}^{-\frac{1}{2}} - \sum_{i=1}^{t-2}\mathbf{C}_i^\rho\bigg)$$

$$= (2-\rho)\bigg(\mathbf{I} - \sqrt{\bar{\alpha}_t}\boldsymbol{\Sigma}_0^{\frac{1}{2}}\boldsymbol{\Sigma}_t^{-\frac{1}{2}} - \sum_{i=1}^{t-1}\frac{1}{2}\sqrt{\bar{\alpha}_i}\beta_{i+1}\boldsymbol{\Sigma}_0^{\frac{1}{2}}\big(\mathbf{I} - \prod_{j=0}^{i-1}(\rho\mathbf{I} + (1-\rho)\alpha_{j+1}\boldsymbol{\Sigma}_j\boldsymbol{\Sigma}_{j+1}^{-1})^{\frac{1}{2}}\big)\boldsymbol{\Sigma}_{i+1}^{-1}\boldsymbol{\Sigma}_i^{-\frac{1}{2}}\bigg),$$

where (b) follows from $\boldsymbol{\Sigma}_t = \alpha_t\boldsymbol{\Sigma}_{t-1} + \beta_t\mathbf{I}$ and Eq. (12).

## B.3 Unconditional Distributions of $p_{Z_0}(\cdot)$

With Lemma 5 and

$$p_{Z_t}(\mathbf{z}_t) = \mathcal{N}(\sqrt{\bar{\alpha}_t}\boldsymbol{\mu}_0, \ \bar{\alpha}_t\boldsymbol{\Sigma}_0 + (1 - \bar{\alpha}_t)\mathbf{I})$$

$$p_{Z_0|Z_t}(\mathbf{z}_0|\mathbf{z}_t) = \mathcal{N}(\mathbf{A}_t^\rho \check{\mathbf{z}}_t + \mathbf{B}_t^\rho \boldsymbol{\mu}_0, \ \lambda\boldsymbol{\Lambda}_t^\rho),$$

we have the mean of $p_{Z_0}(\mathbf{z}_0)$ is

$$\mathbf{A}_t^\rho \sqrt{\bar{\alpha}_t}\boldsymbol{\mu}_0 + \mathbf{B}_t^\rho \boldsymbol{\mu}_0 = (\mathbf{A}_t^\rho \sqrt{\bar{\alpha}_t} + \mathbf{B}_t^\rho)\boldsymbol{\mu}_0,$$

and the variance is

$$\lambda\boldsymbol{\Lambda}_t^\rho + \mathbf{A}_t^\rho \big(\bar{\alpha}_t\boldsymbol{\Sigma}_0 + (1 - \bar{\alpha}_t)\mathbf{I}\big)\mathbf{A}_t^{\rho\top}.$$

First, we show that the mean of $p_{Z_0}(\mathbf{z}_0)$ is $\boldsymbol{\mu}_0$ regardless of the value of $\rho$. For any $\rho \in [0, 1]$ and $t \in \{1, \ldots, T\}$, we have

$$\mathbf{A}_t^\rho \sqrt{\bar{\alpha}_t} + \mathbf{B}_t^\rho$$

$$=\bar{\alpha}_t\Big(\boldsymbol{\Sigma}_0 + \frac{1}{2}\rho\frac{\beta_1}{\alpha_1}\mathbf{I}\Big)\Big(\prod_{i=1}^{t-1}\big(\mathbf{I} + \frac{1}{2}\rho\frac{\beta_{i+1}}{\alpha_{i+1}}\boldsymbol{\Sigma}_i^{-1}\big)\Big)\boldsymbol{\Sigma}_t^{-1}$$

$$+ \frac{1}{2}(2 - \rho)\Big(\sum_{i=k+1}^{t}\bar{\alpha}_{i-1}\beta_i\Big(\boldsymbol{\Sigma}_0 + \frac{1}{2}\rho\frac{\beta_1}{\alpha_1}\mathbf{I}\Big)\Big(\prod_{j=1}^{i-2}\big(\mathbf{I} + \frac{1}{2}\rho\frac{\beta_{j+1}}{\alpha_{j+1}}\boldsymbol{\Sigma}_j^{-1}\big)\Big)\boldsymbol{\Sigma}_{i-1}^{-1}\boldsymbol{\Sigma}_i^{-1} + \beta_1\boldsymbol{\Sigma}_1^{-1}\Big)$$

$$=\bar{\alpha}_t\Big(\boldsymbol{\Sigma}_0 + \frac{1}{2}\rho\frac{\beta_1}{\alpha_1}\mathbf{I}\Big)\big(\mathbf{I} + \frac{1}{2}\rho\frac{\beta_2}{\alpha_2}\boldsymbol{\Sigma}_1^{-1}\big)\cdots\big(\mathbf{I} + \frac{1}{2}\rho\frac{\beta_t}{\alpha_t}\boldsymbol{\Sigma}_{t-1}^{-1}\big)\boldsymbol{\Sigma}_t^{-1} \qquad \text{①}$$

$$+ \frac{1}{2}(2 - \rho)\bar{\alpha}_{t-1}\beta_t\Big(\boldsymbol{\Sigma}_0 + \frac{1}{2}\rho\frac{\beta_1}{\alpha_1}\mathbf{I}\Big)\big(\mathbf{I} + \frac{1}{2}\rho\frac{\beta_2}{\alpha_2}\boldsymbol{\Sigma}_1^{-1}\big)\cdots\big(\mathbf{I} + \frac{1}{2}\rho\frac{\beta_{t-1}}{\alpha_{t-1}}\boldsymbol{\Sigma}_{t-2}^{-1}\big)\boldsymbol{\Sigma}_{t-1}^{-1}\boldsymbol{\Sigma}_t^{-1} \qquad \text{②}$$

$$+ \frac{1}{2}(2 - \rho)\bar{\alpha}_{t-2}\beta_{t-1}\Big(\boldsymbol{\Sigma}_0 + \frac{1}{2}\rho\frac{\beta_1}{\alpha_1}\mathbf{I}\Big)\big(\mathbf{I} + \frac{1}{2}\rho\frac{\beta_2}{\alpha_2}\boldsymbol{\Sigma}_1^{-1}\big)\cdots\big(\mathbf{I} + \frac{1}{2}\rho\frac{\beta_{t-2}}{\alpha_{t-2}}\boldsymbol{\Sigma}_{t-3}^{-1}\big)\boldsymbol{\Sigma}_{t-2}^{-1}\boldsymbol{\Sigma}_{t-1}^{-1} \qquad \text{③}$$

$$+ \cdots$$

$$+ \frac{1}{2}(2 - \rho)\bar{\alpha}_1\beta_2\Big(\boldsymbol{\Sigma}_0 + \frac{1}{2}\rho\frac{\beta_1}{\alpha_1}\mathbf{I}\Big)\boldsymbol{\Sigma}_1^{-1}\boldsymbol{\Sigma}_2^{-1} \qquad \text{ⓣ}$$

$$+ \frac{1}{2}(2 - \rho)\beta_1\boldsymbol{\Sigma}_1^{-1}.$$

By first considering ① + ②, we have

$$\text{①} + \text{②} = \bar{\alpha}_{t-1}\Big(\boldsymbol{\Sigma}_0 + \frac{1}{2}\rho\frac{\beta_1}{\alpha_1}\mathbf{I}\Big)\big(\mathbf{I} + \frac{1}{2}\rho\frac{\beta_2}{\alpha_2}\boldsymbol{\Sigma}_1^{-1}\big)\cdots\big(\mathbf{I} + \frac{1}{2}\rho\frac{\beta_{t-1}}{\alpha_{t-1}}\boldsymbol{\Sigma}_{t-2}^{-1}\big)\Big[\alpha_t\big(\mathbf{I} + \frac{1}{2}\rho\frac{\beta_t}{\alpha_t}\boldsymbol{\Sigma}_{t-1}^{-1}\big) + \frac{1}{2}(2 - \rho)\beta_t\boldsymbol{\Sigma}_{t-1}^{-1}\Big]\boldsymbol{\Sigma}_t^{-1}$$

$$= \bar{\alpha}_{t-1}\Big(\boldsymbol{\Sigma}_0 + \frac{1}{2}\rho\frac{\beta_1}{\alpha_1}\mathbf{I}\Big)\big(\mathbf{I} + \frac{1}{2}\rho\frac{\beta_2}{\alpha_2}\boldsymbol{\Sigma}_1^{-1}\big)\cdots\big(\mathbf{I} + \frac{1}{2}\rho\frac{\beta_{t-1}}{\alpha_{t-1}}\boldsymbol{\Sigma}_{t-2}^{-1}\big)\boldsymbol{\Sigma}_{t-1}^{-1}\Big[\alpha_t\boldsymbol{\Sigma}_{t-1} + \frac{1}{2}\rho\beta_t\mathbf{I} + \frac{1}{2}(2 - \rho)\beta_t\mathbf{I}\Big]\boldsymbol{\Sigma}_t^{-1}$$

$$= \bar{\alpha}_{t-1}\Big(\boldsymbol{\Sigma}_0 + \frac{1}{2}\rho\frac{\beta_1}{\alpha_1}\mathbf{I}\Big)\big(\mathbf{I} + \frac{1}{2}\rho\frac{\beta_2}{\alpha_2}\boldsymbol{\Sigma}_1^{-1}\big)\cdots\big(\mathbf{I} + \frac{1}{2}\rho\frac{\beta_{t-1}}{\alpha_{t-1}}\boldsymbol{\Sigma}_{t-2}^{-1}\big)\boldsymbol{\Sigma}_{t-1}^{-1}\Big[\bar{\alpha}_t\boldsymbol{\Sigma}_0 + (\alpha_t - \bar{\alpha}_t + \beta_t)\mathbf{I}\Big]\boldsymbol{\Sigma}_t^{-1}$$

$$= \bar{\alpha}_{t-1}\Big(\boldsymbol{\Sigma}_0 + \frac{1}{2}\rho\frac{\beta_1}{\alpha_1}\mathbf{I}\Big)\big(\mathbf{I} + \frac{1}{2}\rho\frac{\beta_2}{\alpha_2}\boldsymbol{\Sigma}_1^{-1}\big)\cdots\big(\mathbf{I} + \frac{1}{2}\rho\frac{\beta_{t-1}}{\alpha_{t-1}}\boldsymbol{\Sigma}_{t-2}^{-1}\big)\boldsymbol{\Sigma}_{t-1}^{-1}\boldsymbol{\Sigma}_t\boldsymbol{\Sigma}_t^{-1}$$

$$= \bar{\alpha}_{t-1}\Big(\boldsymbol{\Sigma}_0 + \frac{1}{2}\rho\frac{\beta_1}{\alpha_1}\mathbf{I}\Big)\big(\mathbf{I} + \frac{1}{2}\rho\frac{\beta_2}{\alpha_2}\boldsymbol{\Sigma}_1^{-1}\big)\cdots\big(\mathbf{I} + \frac{1}{2}\rho\frac{\beta_{t-1}}{\alpha_{t-1}}\boldsymbol{\Sigma}_{t-2}^{-1}\big)\boldsymbol{\Sigma}_{t-1}^{-1},$$

which has a similar structure to ① with $t$ replaced by $t-1$. Thus, when proceeding to the next line, we have a similar induction structure

$$\text{①} + \text{②} + \text{③}$$

$$=\bar{\alpha}_{t-2}\Big(\boldsymbol{\Sigma}_0 + \frac{1}{2}\rho\frac{\beta_1}{\alpha_1}\mathbf{I}\Big)\big(\mathbf{I} + \frac{1}{2}\rho\frac{\beta_2}{\alpha_2}\boldsymbol{\Sigma}_1^{-1}\big)\cdots\big(\mathbf{I} + \frac{1}{2}\rho\frac{\beta_{t-2}}{\alpha_{t-2}}\boldsymbol{\Sigma}_{t-3}^{-1}\big)\Big[\alpha_{t-1}\big(\mathbf{I} + \frac{1}{2}\rho\frac{\beta_{t-1}}{\alpha_{t-1}}\boldsymbol{\Sigma}_{t-2}^{-1}\big) + \frac{1}{2}(2 - \rho)\beta_{t-1}\boldsymbol{\Sigma}_{t-2}^{-1}\Big]\boldsymbol{\Sigma}_{t-1}^{-1}$$

$$=\bar{\alpha}_{t-2}\Big(\boldsymbol{\Sigma}_0 + \frac{1}{2}\rho\frac{\beta_1}{\alpha_1}\mathbf{I}\Big)\big(\mathbf{I} + \frac{1}{2}\rho\frac{\beta_2}{\alpha_2}\boldsymbol{\Sigma}_1^{-1}\big)\cdots\big(\mathbf{I} + \frac{1}{2}\rho\frac{\beta_{t-2}}{\alpha_{t-2}}\boldsymbol{\Sigma}_{t-3}^{-1}\big)\boldsymbol{\Sigma}_{t-2}^{-1}\Big[\alpha_{t-1}\boldsymbol{\Sigma}_{t-2} + \frac{1}{2}\rho\beta_{t-1}\mathbf{I} + \frac{1}{2}(2 - \rho)\beta_{t-1}\mathbf{I}\Big]\boldsymbol{\Sigma}_{t-1}^{-1}$$

$$=\bar{\alpha}_{t-2}\Big(\boldsymbol{\Sigma}_0 + \frac{1}{2}\rho\frac{\beta_1}{\alpha_1}\mathbf{I}\Big)\big(\mathbf{I} + \frac{1}{2}\rho\frac{\beta_2}{\alpha_2}\boldsymbol{\Sigma}_1^{-1}\big)\cdots\big(\mathbf{I} + \frac{1}{2}\rho\frac{\beta_{t-2}}{\alpha_{t-2}}\boldsymbol{\Sigma}_{t-3}^{-1}\big)\boldsymbol{\Sigma}_{t-2}^{-1}\boldsymbol{\Sigma}_{t-2}\boldsymbol{\Sigma}_{t-1}^{-1}$$

$$=\bar{\alpha}_{t-2}\Big(\boldsymbol{\Sigma}_0 + \frac{1}{2}\rho\frac{\beta_1}{\alpha_1}\mathbf{I}\Big)\big(\mathbf{I} + \frac{1}{2}\rho\frac{\beta_2}{\alpha_2}\boldsymbol{\Sigma}_1^{-1}\big)\cdots\big(\mathbf{I} + \frac{1}{2}\rho\frac{\beta_{t-2}}{\alpha_{t-2}}\boldsymbol{\Sigma}_{t-3}^{-1}\big)\boldsymbol{\Sigma}_{t-1}^{-1}.$$

By induction, the summation over $\boxed{1} \cdots \boxed{t}$ is

$$\boxed{1} + \cdots + \boxed{t}$$

$$= \bar{\alpha}_2 \Big( \boldsymbol{\Sigma}_0 + \frac{1}{2} \rho \frac{\beta_1}{\alpha_1} \mathbf{I} \Big) \Big( \mathbf{I} + \frac{1}{2} \rho \frac{\beta_2}{\alpha_2} \boldsymbol{\Sigma}_1^{-1} \Big) \boldsymbol{\Sigma}_2^{-1} + \frac{1}{2} (2 - \rho) \bar{\alpha}_1 \beta_2 \Big( \boldsymbol{\Sigma}_0 + \frac{1}{2} \rho \frac{\beta_1}{\alpha_1} \mathbf{I} \Big) \boldsymbol{\Sigma}_1^{-1} \boldsymbol{\Sigma}_2^{-1}$$

$$= \alpha_1 \Big( \boldsymbol{\Sigma}_0 + \frac{1}{2} \rho \frac{\beta_1}{\alpha_1} \mathbf{I} \Big) \Big[ \alpha_2 \Big( \mathbf{I} + \frac{1}{2} \rho \frac{\beta_2}{\alpha_2} \boldsymbol{\Sigma}_1^{-1} \Big) + \frac{1}{2} (1 - \rho) \beta_2 \boldsymbol{\Sigma}_1^{-1} \Big] \boldsymbol{\Sigma}_2^{-1}$$

$$= \alpha_1 \Big( \boldsymbol{\Sigma}_0 + \frac{1}{2} \rho \frac{\beta_1}{\alpha_1} \mathbf{I} \Big) \boldsymbol{\Sigma}_1^{-1} \Big[ \alpha_2 \boldsymbol{\Sigma}_1 + \frac{1}{2} \rho \beta_2 + \frac{1}{2} (1 - \rho) \beta_2 \Big] \boldsymbol{\Sigma}_2^{-1}$$

$$= \alpha_1 \Big( \boldsymbol{\Sigma}_0 + \frac{1}{2} \rho \frac{\beta_1}{\alpha_1} \mathbf{I} \Big) \boldsymbol{\Sigma}_1^{-1}.$$

Finally, we have the mean of $p_{Z_0}(\mathbf{z}_0)$ is

$$\Big( \mathbf{A}_t^\rho \sqrt{\bar{\alpha}_t} + \mathbf{B}_t^\rho \Big) \boldsymbol{\mu}_0 = \Big( \alpha_1 \Big( \boldsymbol{\Sigma}_0 + \frac{1}{2} \rho \frac{\beta_1}{\alpha_1} \mathbf{I} \Big) \boldsymbol{\Sigma}_1^{-1} + \frac{1}{2} (2 - \rho) \beta_1 \boldsymbol{\Sigma}_1^{-1} \Big) \boldsymbol{\mu}_0$$

$$= \Big[ \alpha_1 \boldsymbol{\Sigma}_0 + \frac{1}{2} \rho \beta_1 \mathbf{I} + \frac{1}{2} (2 - \rho) \beta_1 \mathbf{I} \Big] \boldsymbol{\Sigma}_1^{-1} \boldsymbol{\mu}_0$$

$$= \Big[ \alpha_1 \boldsymbol{\Sigma}_0 + (1 - \alpha_1) \mathbf{I} \Big] \boldsymbol{\Sigma}_1^{-1} \boldsymbol{\mu}_0$$

$$= \boldsymbol{\Sigma}_1 \boldsymbol{\Sigma}_1^{-1} \boldsymbol{\mu}_0 = \boldsymbol{\mu}_0.$$

Now, let's compute the variance of $p_{Z_0}(\mathbf{z}_0)$ with the help of the second expression of $\mathbf{A}_t^\rho$ in Eq. (13). For $\lambda = 0$, the variance of $p_{Z_0}(\mathbf{z}_0)$ is

$$\lambda \boldsymbol{\Lambda}_t^\rho + \mathbf{A}_t^\rho \big( \bar{\alpha}_t \boldsymbol{\Sigma}_0 + (1 - \bar{\alpha}_t) \mathbf{I} \big) \mathbf{A}_t^{\rho\top}$$

$$= \boldsymbol{\Sigma}_0^{\frac{1}{2}} \prod_{i=0}^{t-1} \Big( \rho \mathbf{I} + (1 - \rho) \alpha_{i+1} \boldsymbol{\Sigma}_i \boldsymbol{\Sigma}_{i+1}^{-1} \Big)^{\frac{1}{2}} \boldsymbol{\Sigma}_t^{-\frac{1}{2}} \boldsymbol{\Sigma}_t \Big( \boldsymbol{\Sigma}_0^{\frac{1}{2}} \prod_{i=0}^{t-1} \Big( \rho \mathbf{I} + (1 - \rho) \alpha_{i+1} \boldsymbol{\Sigma}_i \boldsymbol{\Sigma}_{i+1}^{-1} \Big)^{\frac{1}{2}} \boldsymbol{\Sigma}_t^{-\frac{1}{2}} \Big)^\top$$

$$= \boldsymbol{\Sigma}_0 \prod_{i=0}^{t-1} \Big( \rho \mathbf{I} + (1 - \rho) \alpha_{i+1} \boldsymbol{\Sigma}_i \boldsymbol{\Sigma}_{i+1}^{-1} \Big). \tag{15}$$

When $\rho = 0$, the variance is

$$\boldsymbol{\Sigma}_0 \prod_{i=0}^{t-1} \Big( \alpha_{i+1} \boldsymbol{\Sigma}_i \boldsymbol{\Sigma}_{i+1}^{-1} \Big) = \bar{\alpha}_t \boldsymbol{\Sigma}_0 \boldsymbol{\Sigma}_0 \boldsymbol{\Sigma}_1^{-1} \cdots \boldsymbol{\Sigma}_{t-1} \boldsymbol{\Sigma}_t^{-1} = \bar{\alpha}_t \boldsymbol{\Sigma}_0^2 \boldsymbol{\Sigma}_t^{-1},$$

and the marginal distribution of the reconstruction is $p_{Z_0}(\mathbf{z}_0) = \mathcal{N}(\boldsymbol{\mu}_0, \bar{\alpha}_t \boldsymbol{\Sigma}_0^2 \boldsymbol{\Sigma}_t^{-1})$.

When $\rho = 1$, the variance is $\boldsymbol{\Sigma}_0$, and the marginal distribution is $p_{Z_0}(\mathbf{z}_0) = \mathcal{N}(\boldsymbol{\mu}_0, \boldsymbol{\Sigma}_0)$, which match the marginal distribution of $p_X(\mathbf{x})$.

## C    PROOF OF THEOREM 3

### C.1    SCALAR GAUSSIAN CASE

Let's first prove the scalar Gaussian case, which will be part of the basis for the achievability proof of the vector Gaussian case.

#### C.1.1    CONVERSE PROOF

Suppose we have the source $X \sim \mathcal{N}(\mu_0, \sigma_0^2)$, and the decoder receives $Z_t = \sqrt{\bar{\alpha}_t} X + \sqrt{1 - \bar{\alpha}_t} N$, $N \sim \mathcal{N}(0, 1)$, which has distribution

$$p_{Z_t}(z_t) = \mathcal{N}(\underbrace{\sqrt{\bar{\alpha}_t} \mu_0}_{\mu_t}, \underbrace{\bar{\alpha}_t \sigma_0^2 + 1 - \bar{\alpha}_t}_{\sigma_t^2}).$$

Consider the following optimization problem:

$$D = \min_{p_{\hat{X}|Z}(\hat{x}|z)} \mathbb{E}[(X - \hat{X})^2]$$

$$\text{s.t. } W_2^2(p_X, p_{\hat{X}}) \le P.$$

To be proved in Lemma 7 in the multivariate case, we can restrict the reconstruction to have the form of $p_{\hat{X}|Z}(\hat{x}|z) = \mathcal{N}(az + b, c^2)$. Then, the marginal distribution can be expressed as $p_{\hat{X}}(\hat{x}) = \mathcal{N}(a\sqrt{\bar{\alpha}_t}\mu_0 + b, c^2 + a^2\sigma_t^2)$, and the covariance of $X$ and $\hat{X}$ is $\text{Cov}[X, \hat{X}] = a\sqrt{\bar{\alpha}_t}\sigma_0^2$. Define $\hat{\mu}_0 := a\sqrt{\bar{\alpha}_t}\mu_0 + b$.

Then, we can express the distortion and perception term as

$$\mathbb{E}[(X - \hat{X})^2] = \mathbb{E}[X^2] - 2\mathbb{E}[X\hat{X}] + \mathbb{E}[\hat{X}^2]$$
$$= \sigma_0^2 + \mu_0^2 - 2(a\sqrt{\bar{\alpha}_t}\sigma_0^2 + \mu_0\hat{\mu}_0) + \hat{\mu}_0^2 + (c^2 + a^2\sigma_t^2)$$
$$= (\mu_0 - \hat{\mu}_0)^2 + \sigma_0^2 + (c^2 + a^2\sigma_t^2) - 2a\sqrt{\bar{\alpha}_t}\sigma_0^2,$$

and

$$W_2^2(p_X, p_{\hat{X}}) = (\mu_0 - \hat{\mu}_0)^2 + (\sigma_0 - \sqrt{c^2 + a^2\sigma_t^2})^2.$$

Without loss of optimality, we can set $\hat{\mu}_0 = \mu_0$, which leads to $b = \mu_0 - a\sqrt{\bar{\alpha}_t}\mu_0$. Then, the optimization problem can be simplified as

$$D = \min_{a,c} \sigma_0^2 + (c^2 + a^2\sigma_t^2) - 2a\sqrt{\bar{\alpha}_t}\sigma_0^2 \tag{16}$$

$$\text{s.t. } (\sigma_0 - \sqrt{c^2 + a^2\sigma_t^2})^2 \le P.$$

Denoting $\hat{\sigma}_0^2 := c^2 + a^2\sigma_t^2$, the problem is equivalent to

$$D = \min_{a,\hat{\sigma}_0} \sigma_0^2 + \hat{\sigma}_0^2 - 2a\sqrt{\bar{\alpha}_t}\sigma_0^2$$

$$\text{s.t. } (\sigma_0 - \hat{\sigma}_0)^2 \le P$$
$$a^2\sigma_t^2 \le \hat{\sigma}_0^2.$$

For any fixed $\hat{\sigma}_0$ (which leads to fixed value of $(\sigma_0 - \hat{\sigma}_0)^2$), we can find the optimal $a$ that minimizes the distortion level to $\frac{\hat{\sigma}_0}{\sigma_t}$. Then, the problem is just to minimize a quadratic function $\hat{\sigma}_0^2 - 2\frac{\hat{\sigma}_0}{\sigma_t}\sqrt{\bar{\alpha}_t}\sigma_0^2 + \sigma_0^2 = (\hat{\sigma}_0 - \frac{\sqrt{\bar{\alpha}_t}\sigma_0^2}{\sigma_t})^2 + (1 - \bar{\alpha}_t)\frac{\sigma_0^2}{\sigma_t^2}$, with respect to $\sigma_0 - \sqrt{P} \le \hat{\sigma}_0 \le \sigma_0 + \sqrt{P}$.

$1^o$ When $\sqrt{P} < \sigma_0 - \frac{\sqrt{\bar{\alpha}_t}\sigma_0^2}{\sigma_t}$, the optimal $\hat{\sigma}_0$ is $\sigma_0 - \sqrt{P}$, and the optimal distortion is $\sigma_0^2 + (\sigma_0 - \sqrt{P})^2 - 2\frac{\sigma_0 - \sqrt{P}}{\sigma_t}\sqrt{\bar{\alpha}_t}\sigma_0^2 = \frac{(1-\bar{\alpha}_t)\sigma_0^2}{\sigma_t^2} + (\sigma_0 - \sqrt{P} - \frac{\sigma_0^2\sqrt{\bar{\alpha}_t}}{\sigma_t})^2$. The corresponding reconstruction distribution is

$$p_{\hat{X}|Z}(\hat{x}|z) = \mathcal{N}\Big(\frac{\sigma_0 - \sqrt{P}}{\sigma_t}z + \Big(1 - \frac{(\sigma_0 - \sqrt{P})\sqrt{\bar{\alpha}_t}}{\sigma_t}\Big)\mu_0, \ 0\Big).$$

$2^o$ When $\sqrt{P} \ge \sigma_0 - \frac{\sqrt{\bar{\alpha}_t}\sigma_0^2}{\sigma_t}$, the optimal $\hat{\sigma}_0$ is $\frac{\sqrt{\bar{\alpha}_t}\sigma_0^2}{\sigma_t}$, and the optimal distortion is $\frac{(1-\bar{\alpha}_t)\sigma_0^2}{\sigma_t^2}$. The corresponding reconstruction distribution is

$$p_{\hat{X}|Z}(\hat{x}|z) = \mathcal{N}\Big(\frac{\sqrt{\bar{\alpha}_t}\sigma_0^2}{\sigma_t^2}z + \frac{1 - \bar{\alpha}_t}{\sigma_t^2}\mu_0, \ 0\Big).$$

The resulting DP tradeoff is

$$D_t(P) = \begin{cases} \frac{(1-\bar{\alpha}_t)\sigma_0^2}{\sigma_t^2} + \Big(\sigma_0 - \sqrt{P} - \frac{\sigma_0^2\sqrt{\bar{\alpha}_t}}{\sigma_t}\Big)^2 & \sqrt{P} < \sigma_0 - \frac{\sqrt{\bar{\alpha}_t}\sigma_0^2}{\sigma_t}, \\ \frac{(1-\bar{\alpha}_t)\sigma_0^2}{\sigma_t^2} & \sqrt{P} \ge \sigma_0 - \frac{\sqrt{\bar{\alpha}_t}\sigma_0^2}{\sigma_t}. \end{cases}$$

**Remark 3.** *We can observe that $c^2$, which is the variance of the reconstruction, is always zero. This coincides with the fact that the reconstruction is a deterministic function of z.*

### C.1.2 ACHIEVABILITY PROOF

As shown in Eq. (11)-Eq. (14), the reconstruction $\hat{X}^\rho$ given by our proposed score-scaled PF-ODE is

$$\hat{X}^\rho := Z_0 = a_t^\rho Z_t + b_t^\rho \mu_0,$$

where

$$a_t^\rho = \sqrt{\bar{\alpha}_t} \frac{\sigma_0^2}{\sigma_t^2} \prod_{i=0}^{t-1} \Big(1 + \frac{1}{2}\rho\frac{\beta_{i+1}}{\alpha_{i+1}}\frac{1}{\sigma_i^2}\Big) = \frac{\sigma_0}{\sigma_t} \prod_{i=0}^{t-1} \Big(\rho + (1-\rho)\alpha_{i+1}\frac{\sigma_i^2}{\sigma_{i+1}^2}\Big)^{\frac{1}{2}},$$

$$b_t^\rho = \frac{1}{2}(2-\rho)\Big(\sum_{i=2}^{t}\bar{\alpha}_{i-1}\beta_i\frac{\sigma_0^2}{\sigma_{i-1}^2\sigma_i^2}\prod_{j=0}^{i-2}\Big(1 + \frac{1}{2}\rho\frac{\beta_{j+1}}{\alpha_{j+1}}\frac{1}{\sigma_j}\Big) + \beta_1\frac{1}{\sigma_1^2}\Big)$$

$$= (2-\rho)\Big(1 - \sqrt{\bar{\alpha}_t}\frac{\sigma_0}{\sigma_t} - \sum_{i=1}^{t-1}\frac{1}{2}\sqrt{\bar{\alpha}_i}\beta_{i+1}\frac{\sigma_0}{\sigma_{i+1}^2\sigma_i}\Big(\mathbf{I} - \prod_{j=0}^{i-1}\big(\rho\mathbf{I} + (1-\rho)\alpha_{j+1}\mathbf{\Sigma}_j\mathbf{\Sigma}_{j+1}^{-1}\big)^{\frac{1}{2}}\Big)\Big).$$

According to Eq. (15), the marginal distribution of the reconstruction is

$$p_{\hat{X}^\rho}(\hat{x}) = \mathcal{N}(\mu_0, \hat{\sigma}_0^2),$$

where

$$\hat{\sigma}_0^2 = \bar{\alpha}_t\frac{\sigma_0^4}{\sigma_t^2}\prod_{i=0}^{t-1}\Big(1 + \frac{1}{2}\rho\frac{\beta_{i+1}}{\alpha_{i+1}}\frac{1}{\sigma_i^2}\Big)^2 = \sigma_0^2\prod_{i=0}^{t-1}\Big(\rho + (1-\rho)\alpha_{i+1}\frac{\sigma_i^2}{\sigma_{i+1}^2}\Big).$$

The covariance between $\hat{X}^\rho$ and $X$ is

$$\mathrm{Cov}[X, \hat{X}^\rho] = \mathrm{Cov}[X, a_t^\rho Z_t + b_t^\rho \mu_0] = \mathrm{Cov}[X, a_t^\rho(\sqrt{\bar{\alpha}_t}X + \sqrt{1-\bar{\alpha}_t}N)] = a_t^\rho\sqrt{\bar{\alpha}_t}\sigma_0^2.$$

We can compute the achievable distortion and perception levels of our proposed ODE as functions of $\rho$

$$D_t^\rho = \mathbb{E}[(X - \hat{X}^\rho)] = \sigma_0^2 + \sigma_0^2\prod_{i=0}^{t-1}\Big(\rho + (1-\rho)\alpha_{i+1}\frac{\sigma_i^2}{\sigma_{i+1}^2}\Big) - 2\frac{\sigma_0}{\sigma_t}\prod_{i=0}^{t-1}\Big(\rho + (1-\rho)\alpha_{i+1}\frac{\sigma_i^2}{\sigma_{i+1}^2}\Big)^{\frac{1}{2}}\sqrt{\bar{\alpha}_t}\sigma_0^2$$

$$= \sigma_0^2\Big(\prod_{i=0}^{t-1}\Big(\rho + (1-\rho)\alpha_{i+1}\frac{\sigma_i^2}{\sigma_{i+1}^2}\Big)^{\frac{1}{2}} - \frac{\sqrt{\bar{\alpha}_t}\sigma_0}{\sigma_t}\Big)^2 + \sigma_0^2 - \frac{\bar{\alpha}_t\sigma_0^4}{\sigma_t^2}, \tag{17}$$

$$P_t^\rho = W_2^2(p_X, p_{\hat{X}^\rho}) = \Big(\sigma_0 - \sigma_0\prod_{i=0}^{t-1}\Big(\rho + (1-\rho)\alpha_{i+1}\frac{\sigma_i^2}{\sigma_{i+1}^2}\Big)^{\frac{1}{2}}\Big)^2. \tag{18}$$

From Eq. (18) together with the fact that $\rho + (1-\rho)\alpha_{i+1}\frac{\sigma_i^2}{\sigma_{i+1}^2} \leq 1, \forall i$, we can obtain $\prod_{i=0}^{t-1}\big(\rho + (1-\rho)\alpha_{i+1}\frac{\sigma_i^2}{\sigma_{i+1}^2}\big)^{\frac{1}{2}} = \frac{\sigma_0 - P_t^\rho}{\sigma_0}$. Since $0 \leq \rho \leq 1$, we have $\prod_{i=0}^{t-1}\big(\rho + (1-\rho)\alpha_{i+1}\frac{\sigma_i^2}{\sigma_{i+1}^2}\big)^{\frac{1}{2}} \in \big[\frac{\sqrt{\bar{\alpha}_t}\sigma_0}{\sigma_t}, 1\big]$, thus $P_t^\rho \in [0, (\sigma_0 - \frac{\sqrt{\bar{\alpha}_t}\sigma_0}{\sigma_t})^2]$. Plugging into Eq. (17) we can get the achievable DP tradeoff is

$$D_t^\rho(P_t^\rho) = \sigma_0^2\Big(\frac{\sigma_0 - \sqrt{P_t^\rho}}{\sigma_0} - \frac{\sqrt{\bar{\alpha}_t}\sigma_0}{\sigma_t}\Big)^2 + \frac{\sigma_0^2(\sigma_t^2 - \bar{\alpha}_t\sigma_0^2)}{\sigma_t^2}$$

$$= \Big(\sigma_0 - \sqrt{P_t^\rho} - \frac{\sqrt{\bar{\alpha}_t}\sigma_0^2}{\sigma_t}\Big)^2 + \frac{(1-\bar{\alpha}_t)\sigma_0^2}{\sigma_t^2}, \text{ for } 0 \leq P \leq \big(\sigma_0 - \frac{\sqrt{\bar{\alpha}_t}\sigma_0}{\sigma_t}\big)^2,$$

which matches the optimal DP tradeoff derived in the last section.

### C.2 CONVERSE PROOF FOR MULTIVARIATE GAUSSIAN CASE

For a $d$-dimensional source $X = (X_1, \cdots, X_d) \sim \mathcal{N}(\boldsymbol{\mu}_0, \boldsymbol{\Sigma}_0)$, consider the eigen-decomposition of the covariance matrix

$$\boldsymbol{\Sigma}_0 = \mathbf{Q}\boldsymbol{\Lambda}_0\mathbf{Q}^\top,$$

where $\mathbf{Q}$ is orthogonal and $\boldsymbol{\Lambda}_0$ is a diagonal matrix with positive eigenvalues $\boldsymbol{\Lambda}_0 = \mathrm{diag}(\lambda_0, \cdots, \lambda_d)$. We then define

$$Y = \mathbf{Q}^\top X,$$

which implies $Y = (Y_1, \cdots, Y_d) \sim \mathcal{N}(\mathbf{Q}^\top \boldsymbol{\mu}_0, \boldsymbol{\Lambda}_0)$. The components of $Y$ are mutually independent.

Given the received $Z_t = \sqrt{\bar{\alpha}_t}X + \sqrt{1 - \bar{\alpha}_t}N$, $N \sim \mathcal{N}(\mathbf{0}, \mathbf{I})$, consider the following optimization problem

$$D = \min_{p_{\hat{X}|Z_t}(\hat{\mathbf{x}}|\mathbf{z}_t)} \mathbb{E}[||X - \hat{X}||_2^2] \tag{19}$$

$$\text{s.t. } W_2^2(p_X, p_{\hat{X}}) \leq P.$$

**Lemma 7.** *Without loss of optimality, for the optimization problem in Eq. (19), we can restrict the conditional distribution $p_{\hat{X}|Z_t}(\hat{\mathbf{x}}|\mathbf{z}_t)$ as the following form: Let $\tilde{Z}_t = \mathbf{Q}^\top Z_t = \sqrt{\bar{\alpha}_t}Y + \sqrt{1 - \bar{\alpha}_t}N_1$ and $\hat{Y} = \tilde{\mathbf{A}}\tilde{Z}_t + \tilde{\mathbf{b}} + \tilde{\mathbf{C}}N_2$, where $N_1, N_2 \overset{i.i.d.}{\sim} \mathcal{N}(\mathbf{0}, \mathbf{I})$, $\tilde{\mathbf{A}} = \mathrm{diag}(\tilde{a}_1, \ldots, \tilde{a}_d)$ and $\tilde{\mathbf{C}} = \mathrm{diag}(\tilde{c}_1, \ldots, \tilde{c}_d)$ are diagonal matrices with $\tilde{c}_\ell \geq 0$ for $1 \leq \ell \leq d$. Then $\hat{X} = \mathbf{Q}\hat{Y}$.*

*Proof.* For any $Y = \mathbf{Q}^\top X$ and $\hat{X} = \mathbf{Q}Y$, we have

$$\mathbb{E}[||X - \hat{X}||_2^2] \overset{(a)}{=} \mathbb{E}[||Y - \hat{Y}||_2^2] = \sum_{\ell=1}^d \mathbb{E}[(Y_\ell - \hat{Y}_\ell)_2^2], \tag{20}$$

where $(a)$ follows from the invariance of Euclidean distance under orthogonal matrix. Meanwhile,

$$W_2^2(p_X, p_{\hat{X}}) \overset{(b)}{=} W_2^2(p_Y, p_{\hat{Y}}) \overset{(c)}{\geq} \sum_{i=1}^d W_2^2(p_{Y_\ell}, p_{\hat{Y}_\ell}), \tag{21}$$

where $(b)$ follows from the invariance of Wasserstein-2 under unitary transformations; $(c)$ follows from the tensorization property of Wasserstein-2 distance and the equality holds if $(Y_i, \hat{Y}_i)$ and $(Y_j, \hat{Y}_j)$ are independent for any $i \neq j$ (Panaretos & Zemel, 2020). Thus, we can optimize on $p_{\hat{Y}|\tilde{Z}_t}(\hat{\mathbf{y}}|\tilde{\mathbf{z}}_t)$ instead of $p_{\hat{X}|Z_t}(\hat{\mathbf{x}}|\mathbf{z}_t)$ and assume that $\hat{Y} = (\hat{Y}_1, \ldots, \hat{Y}_d)$ has independent components without loss of optimality.

Furthermore, as discussed in Wang et al. (2025) and Qian et al. (2025), the optimal $\hat{Y}$ must be jointly Gaussian with $Y$, which implies that $\tilde{Z}_t$ and $\hat{Y}$ are jointly Gaussian. Together with the independence of components within each $\tilde{Z}_t$ and $\hat{Y}$, we have $(\tilde{Z}_{t,\ell}, Y_\ell)$ are jointly Gaussian for each $\ell$ and $\{(\tilde{Z}_{t,\ell}, Y_\ell)\}_{\ell=1}^d$ are mutually independent with each other.

For any bivariate Gaussian $U \sim \mathcal{N}(\mu_u, \sigma_u^2)$, $V \sim \mathcal{N}(\mu_v, \sigma_v^2)$, and given covariance $\mathrm{Cov}[U, V] = \theta$, one can express $U$ and $V$ as

$$U \sim \mathcal{N}(\mu_u, \sigma_u^2)$$

$$V = \frac{\theta}{\sigma_u^2}U - \frac{\sigma_v}{\sigma_u}\mu_u + \mu_v + \sqrt{\sigma_v^2 - \frac{\theta^2}{\sigma_u^2}}N, \; N \sim \mathcal{N}(0, 1).$$

Thus, for any possible first and second moments induced by a conditional distribution $p_{\hat{Y}_\ell|Z_{t,\ell}}$ with $(\tilde{Z}_{t,\ell}, \hat{Y}_\ell)$ being jointly Gaussian, we can express the conditional relationship with a linear expression $\hat{Y}_\ell = \tilde{a}_\ell \tilde{Z}_{t,\ell} + \tilde{b}_\ell + \tilde{c}_\ell N$, $N \sim \mathcal{N}(0, 1)$ with adjustable $\tilde{a}_\ell$, $\tilde{b}_\ell$, and $\tilde{c}_\ell$.

In summary, it is sufficient to consider the reconstruction of $\hat{Y}$ as $\hat{Y} = \tilde{\mathbf{A}}\tilde{Z}_t + \tilde{\mathbf{b}} + \tilde{\mathbf{C}}N$, where $N \sim \mathcal{N}(\mathbf{0}, \mathbf{I})$, $\tilde{\mathbf{A}} = \mathrm{diag}(\tilde{a}_1, \cdots, \tilde{a}_d)$ and $\tilde{\mathbf{C}} = \mathrm{diag}(\tilde{c}_1, \cdots, \tilde{c}_d)$ are diagonal matrices with $\tilde{c}_\ell \geq 0$ for $1 \leq \ell \leq d$. Note that $\tilde{Z}_t = \mathbf{Q}^\top Z_t = \sqrt{\bar{\alpha}_t}\mathbf{Q}^\top X + \sqrt{1 - \bar{\alpha}_t}\mathbf{Q}^\top N = \sqrt{\bar{\alpha}_t}Y + \sqrt{1 - \bar{\alpha}_t}N'$, $N' \sim \mathcal{N}(\mathbf{0}, \mathbf{I})$.

$\square$

Recall that $Y = (Y_1, \cdots, Y_d) \sim \mathcal{N}(\mathbf{Q}^\top \boldsymbol{\mu}_0, \boldsymbol{\Lambda}_0)$, where $\boldsymbol{\Lambda}_0 = \text{diag}(\lambda_0, \cdots, \lambda_d)$. Denote

- $\lambda_\ell^{(t)} := \bar{\alpha}_t \lambda_\ell + (1 - \bar{\alpha}_t)$ representing the variance of $\tilde{Z}_{t,\ell}$;
- $\mu_{y,\ell}$ as the mean of $Y_\ell$;
- $\hat{\mu}_{y,\ell} := \tilde{a}_i \sqrt{\bar{\alpha}_t} \mu_{y,\ell} + \tilde{b}_\ell$ as the mean of $\hat{Y}_\ell$.

Now, the optimization problem can be written as

$$D = \min_{\tilde{\mathbf{A}}, \tilde{\mathbf{b}}, \tilde{\mathbf{C}}} \sum_{\ell=1}^d \mathbb{E}\big[(Y_\ell - \hat{Y}_\ell)_2^2\big] = \sum_{\ell=1}^d \big(\mu_{y,\ell} - \hat{\mu}_{y,\ell}\big)^2 + \lambda_\ell + (\tilde{c}_\ell^2 + \tilde{a}_\ell^2 \lambda_\ell^{(t)}) - 2\tilde{a}\sqrt{\bar{\alpha}_t}\lambda_\ell$$

$$\text{s.t.} \sum_{\ell=1}^d W_2^2(p_{Y_\ell}, p_{\hat{Y}_\ell}) = \sum_{\ell=1}^d \big(\mu_{y,\ell} - \hat{\mu}_{y,\ell}\big)^2 + \Big(\sqrt{\lambda_\ell} - \sqrt{\tilde{c}_\ell^2 + \tilde{a}_\ell^2 \lambda_\ell^{(t)}}\Big)^2 \leq P.$$

We can set $\mu_{y,\ell} = \hat{\mu}_{y,\ell}$ (i.e., $\tilde{b}_\ell = \mu_{y,\ell} - \tilde{a}_\ell \sqrt{\bar{\alpha}_t} \mu_{y,\ell}$) and $\tilde{c}_\ell = 0$ without loss of optimality. Let $f_\ell = \sqrt{\frac{\tilde{a}_\ell^2 \lambda_\ell^{(t)}}{\lambda_\ell}}$. The optimization problem (19) now becomes

$$D_t = \min_{\{f_\ell\}_{\ell=1}^d} \sum_{\ell=0}^d \lambda_\ell \left(f_\ell - \sqrt{\frac{\bar{\alpha}_t \lambda_\ell}{\lambda_\ell^{(t)}}}\right)^2 + \frac{(1 - \bar{\alpha}_t)\lambda_\ell}{\lambda_\ell^{(t)}}$$

$$\text{s.t.} \sum_{\ell=1}^d \lambda_\ell \big(1 - f_\ell\big)^2 \leq P$$

$$f_\ell \geq 0.$$

Consider the following KKT conditions:

$$\frac{\partial}{\partial f_\ell} \left[ \sum_{\ell=0}^d \lambda_\ell \left(f_\ell - \sqrt{\frac{\bar{\alpha}_t \lambda_\ell}{\lambda_\ell^{(t)}}}\right)^2 + \frac{(1 - \bar{\alpha}_t)\lambda_\ell}{\lambda_\ell^{(t)}} + \nu_0 \Big(\sum_{\ell=1}^d \lambda_\ell \big(1 - f_\ell\big)^2 - P\Big) - \sum_{\ell=0}^d \nu_\ell f_\ell \right]$$

$$= 2\lambda_\ell \left(f_\ell - \sqrt{\frac{\bar{\alpha}_t \lambda_\ell}{\lambda_\ell^{(t)}}}\right) + 2\nu_0 \lambda_\ell (f_\ell - 1) - v_\ell = 0, \quad \text{for } \ell = 1, \ldots, d, \tag{22}$$

$$\nu_\ell \geq 0, \quad \text{for } \ell = 0, 1, \ldots, d, \tag{23}$$

$$\nu_0 \Big( \sum_{\ell=1}^d \lambda_\ell \big(1 - f_\ell\big)^2 - P \Big) = 0 \tag{24}$$

$$\nu_\ell f_\ell = 0, \quad \text{for } \ell = 1, \ldots, d. \tag{25}$$

From Eq. (22), we can solve

$$f_\ell = \frac{\sqrt{\bar{\alpha}_t}\sqrt{\lambda_\ell} + \nu_0 \sqrt{\lambda_\ell^{(t)}}}{(1 + \nu_0)\sqrt{\lambda_\ell^{(t)}}} + \frac{\nu_\ell}{2\lambda(1 + \nu_0)}.$$

Since $\lambda_\ell > 0$, by Eq. (23) and Eq. (25), we have $\nu_\ell$ must be zero for $\ell = 1, \cdots, d$. Thus, $f_\ell = \frac{\sqrt{\bar{\alpha}_t}\sqrt{\lambda_\ell} + \nu_0 \sqrt{\lambda_\ell^{(t)}}}{(1 + \nu_0)\sqrt{\lambda_\ell^{(t)}}} > 0$. Plugging the value of $f_\ell$ into Eq. (24), we have

$$\nu_0 \Big( \sum_{\ell=0}^d \lambda_\ell \Big(1 - \frac{\sqrt{\bar{\alpha}_t}\sqrt{\lambda_\ell} + \nu_0 \sqrt{\lambda_\ell^{(t)}}}{(1 + \nu_0)\sqrt{\lambda_\ell^{(t)}}}\Big)^2 - P \Big) = \nu_0 \Big( \frac{1}{(1 + \nu_0)^2} \sum_{\ell=0}^d \frac{\lambda_\ell}{\lambda_\ell^{(t)}} \big(\sqrt{\lambda_\ell^{(t)}} - \sqrt{\bar{\alpha}_t}\sqrt{\lambda_\ell}\big)^2 - P \Big) = 0.$$

If $\nu_0 = 0$, we have $f_\ell = \sqrt{\frac{\bar{\alpha}_t \lambda_\ell}{\lambda_\ell^{(t)}}}$, and $P$ should be larger than $\sum_{\ell=1}^d \frac{\lambda_\ell}{\lambda_\ell^{(t)}} \big(\sqrt{\lambda_\ell^{(t)}} - \sqrt{\bar{\alpha}_t}\sqrt{\lambda_\ell}\big)^2$ to satisfy primal feasibility. Then, the distortion level is

$$D_t = \sum_{\ell=1}^d \frac{(1 - \bar{\alpha}_t)\lambda_\ell}{\lambda_\ell^{(t)}}.$$

Here $\tilde{a}_\ell = \frac{\sqrt{\bar{\alpha}_t}\lambda_\ell}{\lambda_\ell^{(t)}}, \tilde{b}_\ell = \frac{1-\bar{\alpha}_t}{\bar{\alpha}_t+(1-\bar{\alpha}_t)}\mu_{y,\ell}, \tilde{c}_\ell = 0$, and the distribution of $\tilde{Y}_\ell$ is $\mathcal{N}(\mu_{y,\ell}, \bar{\alpha}_t\lambda_\ell)$.

If $\nu_0 > 0$, we have $\frac{1}{(1+\nu_0)^2}\sum_{\ell=0}^d \frac{\lambda_\ell}{\lambda_\ell^{(t)}}(\sqrt{\lambda_\ell^{(t)}} - \sqrt{\bar{\alpha}_t}\sqrt{\lambda_\ell})^2 = P$, which gives us

$$\nu_0 = \sqrt{\frac{1}{P}\sum_{\ell=0}^d \frac{\lambda_\ell}{\lambda_\ell^{(t)}}(\sqrt{\lambda_\ell^{(t)}} - \sqrt{\bar{\alpha}_t}\sqrt{\lambda_\ell})^2} - 1. \tag{26}$$

In this case, $P < \sum_{\ell=1}^d \frac{\lambda_\ell}{\lambda_\ell^{(t)}}(\sqrt{\lambda_\ell^{(t)}} - \sqrt{\bar{\alpha}_t}\sqrt{\lambda_\ell})^2$ to ensure $\nu > 0$. With Eq. (26), we can obtain the value of $f_\ell$ as

$$f_\ell = 1 - \frac{\left(1 - \sqrt{\frac{\bar{\alpha}_t\lambda_\ell}{\lambda_\ell^{(t)}}}\right)\cdot\sqrt{P}}{\sqrt{\sum_{i=1}^d \frac{\lambda_i}{\lambda_i^{(t)}}(\sqrt{\lambda_i^{(t)}} - \sqrt{\bar{\alpha}_t}\sqrt{\lambda_i})^2}}. \tag{27}$$

Then, plugging Eq. (27) into the expression of $D_t$, we can get the optimal DP curve for multivariate Gaussian case:

$$\begin{aligned}
D_t &= \sum_{\ell=1}^d \lambda_\ell\left(1 + \frac{\left(\sqrt{\frac{\alpha_t\lambda_\ell}{\lambda_\ell^{(t)}}} - 1\right)\cdot\sqrt{P}}{\sqrt{\sum_{i=1}^d \frac{\lambda_i}{\lambda_i^{(t)}}(\sqrt{\lambda_i^{(t)}} - \sqrt{\bar{\alpha}_t}\sqrt{\lambda_i})^2}} - \sqrt{\frac{\bar{\alpha}_t\lambda_\ell}{\lambda_\ell^{(t)}}}\right)^2 + \sum_{\ell=1}^d \frac{(1-\bar{\alpha}_t)\lambda_\ell}{\lambda_\ell^{(t)}} \\
&= \sum_{\ell=1}^d \lambda_\ell\frac{\left(1 - \sqrt{\frac{\bar{\alpha}_t\lambda_\ell}{\lambda_\ell^{(t)}}}\right)^2\cdot\left(\sqrt{\sum_{i=1}^d \frac{\lambda_i}{\lambda_i^{(t)}}(\sqrt{\lambda_i^{(t)}} - \sqrt{\bar{\alpha}_t}\sqrt{\lambda_i})^2} - \sqrt{P}\right)^2}{\sum_{i=1}^d \frac{\lambda_i}{\lambda_i^{(t)}}(\sqrt{\lambda_i^{(t)}} - \sqrt{\bar{\alpha}_t}\sqrt{\lambda_i})^2} + \sum_{\ell=1}^d \frac{(1-\bar{\alpha}_t)\lambda_\ell}{\lambda_\ell^{(t)}} \\
&= \frac{\left(\sqrt{\sum_{i=1}^d \frac{\lambda_i}{\lambda_i^{(t)}}(\sqrt{\lambda_i^{(t)}} - \sqrt{\bar{\alpha}_t}\sqrt{\lambda_i})^2} - \sqrt{P}\right)^2}{\sum_{i=1}^d \frac{\lambda_i}{\lambda_i^{(t)}}(\sqrt{\lambda_i^{(t)}} - \sqrt{\bar{\alpha}_t}\sqrt{\lambda_i})^2}\sum_{\ell=1}^d \frac{\lambda_\ell}{\lambda_\ell^{(t)}}(\sqrt{\lambda_\ell^{(t)}} - \sqrt{\bar{\alpha}_t\lambda_\ell})^2 + \sum_{\ell=1}^d \frac{(1-\bar{\alpha}_t)\lambda_\ell}{\lambda_\ell^{(t)}} \\
&= \left(\sqrt{\sum_{i=1}^d \frac{\lambda_i}{\lambda_i^{(t)}}(\sqrt{\lambda_i^{(t)}} - \sqrt{\bar{\alpha}_t}\sqrt{\lambda_i})^2} - \sqrt{P}\right)^2 + \sum_{\ell=1}^d \frac{(1-\bar{\alpha}_t)\lambda_\ell}{\lambda_\ell^{(t)}}. \tag{28}
\end{aligned}$$

## C.3 Achievability

For $Y \sim \mathcal{N}(\boldsymbol{\mu}_y, \boldsymbol{\Lambda}_0)$ and $\tilde{Z}_t = \sqrt{\bar{\alpha}_t}Y + \sqrt{1-\bar{\alpha}_t}N', N' \sim \mathcal{N}(\mathbf{0}, \mathbf{I})$, let

$$\boldsymbol{\Lambda}_k = \bar{\alpha}_k\boldsymbol{\Lambda}_0 + (1-\bar{\alpha}_k)\mathbf{I} = \mathrm{diag}(\lambda_1^{(k)}, \ldots, \lambda_d^{(k)}), \quad \text{for } k \in \{0, \ldots, t\},$$

where $\lambda_\ell^{(k)} = \bar{\alpha}_k\lambda_\ell + (1-\bar{\alpha}_k)$.

Similar to (Theis et al., 2022), we can decompose the score-scaled PF-ODE into $d$ separate ODEs, each of which is given by

$$d\overleftarrow{Z}_{\tau,\ell} = \left[-\frac{1}{2}\beta(\tau)\overleftarrow{Z}_{\tau,\ell} - \frac{1}{2}(2-\rho_\ell)\beta(t)\nabla_{z_{\tau,\ell}}\log p_t(\overleftarrow{Z}_{\tau,\ell})\right]d\tau, \quad \overleftarrow{Z}_{t,\ell} = \tilde{Z}_{t,\ell} \sim \sqrt{\bar{\alpha}_t}Y_\ell + \sqrt{1-\bar{\alpha}_t}N.$$

For each $\ell \in \{1, \ldots, d\}$, we can always find a $\rho_\ell \in [0, 1]$ such that

$$\prod_{i=0}^{t-1}\sqrt{\rho_\ell + (1-\rho_\ell)\alpha_{i+1}\frac{\lambda_\ell^{(i)}}{\lambda_\ell^{(i+1)}}} = 1 - \frac{\left(1 - \sqrt{\frac{\bar{\alpha}_t\lambda_\ell}{\lambda_\ell^{(i)}}}\right)\cdot\sqrt{P}}{\sqrt{\sum_{i=1}^d \lambda_i\left(1 - \sqrt{\frac{\bar{\alpha}_t\lambda_i}{\lambda_i^{(t)}}}\right)^2}}. \tag{29}$$

Then, according to Eq. (17) and Eq. (18), the achievable distortion and Wasserstein levels in each dimension $\ell$ given by the per-dimensional ODE is

$$D_\ell = \lambda_\ell \left(1 - \frac{\left(1 - \sqrt{\frac{\bar{\alpha}_t \lambda_\ell}{\lambda_\ell^{(t)}}}\right) \cdot \sqrt{P}}{\sqrt{\sum_{i=1}^d \lambda_i \left(1 - \sqrt{\frac{\bar{\alpha}_t \lambda_i}{\lambda_i^{(t)}}}\right)^2}} - \sqrt{\frac{\bar{\alpha}_t \lambda_\ell}{\lambda_\ell^{(t)}}}\right)^2 + \sum_{\ell=1}^d \frac{(1-\bar{\alpha}_t)\lambda_\ell}{\lambda_\ell^{(t)}},$$

$$P_\ell = \frac{\lambda_\ell \left(1 - \sqrt{\frac{\bar{\alpha}_t \lambda_\ell}{\lambda_\ell^{(t)}}}\right) \cdot \sqrt{P}}{\sqrt{\sum_{i=1}^d \lambda_i \left(1 - \sqrt{\frac{\bar{\alpha}_t \lambda_i}{\lambda_i^{(t)}}}\right)^2}}.$$

Denote the reconstruction in dimension $\ell$ as $\hat{Y}_\ell$. Overall, we can obtain that the achievable distortion and Wasserstein levels is

$$\mathbb{E}[||X - \hat{X}||_2^2] \overset{(a)}{=} \mathbb{E}[||Y - \hat{Y}||_2^2] = \sum_{\ell=1}^d \mathbb{E}[(Y_\ell - \hat{Y}_\ell)^2] = \sum_{\ell=1}^d D_\ell$$

$$= \left(\sqrt{\sum_{i=1}^d \frac{\lambda_i}{\lambda_i^{(t)}} \left(\sqrt{\lambda_i^{(t)}} - \sqrt{\bar{\alpha}_t}\sqrt{\lambda_i}\right)^2} - \sqrt{P}\right)^2 + \sum_{\ell=1}^d \frac{(1-\bar{\alpha}_t)\lambda_\ell}{\lambda_\ell^{(t)}},$$

$$W_2^2(p_X, p_{\hat{X}}) \overset{(b)}{=} W_2^2(p_Y, p_{\hat{Y}}) \overset{(c)}{=} \sum_{i=1}^d W_2^2(p_{Y_\ell}, p_{\hat{Y}_\ell}) = \sum_{\ell=1}^d P_\ell = P.$$

Thus, the achievable DP tradeoff is

$$D_t^\rho = \left(\sqrt{\sum_{i=1}^d \frac{\lambda_i}{\lambda_i^{(t)}} \left(\sqrt{\lambda_i^{(t)}} - \sqrt{\bar{\alpha}_t}\sqrt{\lambda_i}\right)^2} - \sqrt{P_t^\rho}\right)^2 + \sum_{\ell=1}^d \frac{(1-\bar{\alpha}_t)\lambda_\ell}{\lambda_\ell^{(t)}},$$

which coincide with the optimal DP tradeoff derived in Eq. (28). The optimal DP tradeoff can be achieved by component-wise reconstruction with delicate design of $\rho_\ell$ as in Eq. (29).

## D  PROOF OF THEOREM 4

Let $I_t = I(X; \sqrt{\bar{\alpha}_t}X + \sqrt{1 - \bar{\alpha}_t}N)$. We have

$$\begin{aligned}
I_t &= I(X; \sqrt{\bar{\alpha}_t}X + \sqrt{1 - \bar{\alpha}_t}N) \\
&= h(\sqrt{\bar{\alpha}_t}X + \sqrt{1 - \bar{\alpha}_t}N) - h(\sqrt{\bar{\alpha}_t}X + \sqrt{1 - \bar{\alpha}_t}N|X) \\
&= \frac{1}{2}\log\left(2\pi(\bar{\alpha}_t\sigma_0^2 + 1 - \bar{\alpha}_t)\right) - \frac{1}{2}\log\left(2\pi(1 - \bar{\alpha}_t)\right) \\
&= \frac{1}{2}\log\left(\frac{\bar{\alpha}_t}{1 - \bar{\alpha}_t}\sigma_0^2 + 1\right),
\end{aligned}$$

where $h(\cdot)$ denotes the differential entropy for continuous random variables. Given a noising level $t$, the RCC encoder transmits the codeword $M$, and the RCC decoder produces $Z_t = \sqrt{\bar{\alpha}_t}X + \sqrt{1 - \bar{\alpha}_t}N$. Subsequently, the score-scaled PF-ODE reconstructs $\hat{X}^\rho$ for a chosen $\rho$ from $Z_t$.

According to the strong functional representation lemma Li & Gamal (2018), the one-shot achievable rate $R_t^1$ is bounded by the cross-entropy between the distribution of $M$ and the Zipf distribution $\text{Zipf}(1 + 1/(I(X; Z_t) + 1))$, i.e.,

$$H(M) \leq I(X; Z_t) + \log_2(I(X; Z_t) + 1) + 4 \text{ bits.}$$

Thus, the one-shot and asymptotic *achievable* rates (denoted as $R_t^1$ and $R_t^\infty$) provided by the PFR algorithm are (Li & Gamal, 2018)

$$\begin{aligned}
I_t \leq &R_t^1 \leq I_t + \log(I_t + 1) + 4, \\
&R_t^\infty = I_t.
\end{aligned}$$

According to Eq. (17) and Eq. (18), the *achievable* distortion and perception levels by adjusting compression parameter $t$ and score-scaling parameter $\rho$ are

$$D_t^\rho = \sigma_0^2 + \sigma_0^2 \prod_{i=0}^{t-1} \left( \rho + (1-\rho)\alpha_{i+1} \frac{\sigma_i^2}{\sigma_{i+1}^2} \right) - 2\frac{\sigma_0}{\sigma_t} \prod_{i=0}^{t-1} \left( \rho + (1-\rho)\alpha_{i+1} \frac{\sigma_i^2}{\sigma_{i+1}^2} \right)^{\frac{1}{2}} \sqrt{\bar{\alpha}_t} \sigma_0^2$$

$$= \sigma_0^2 \Big( \prod_{i=0}^{t-1} \left( \rho + (1-\rho)\alpha_{i+1} \frac{\sigma_i^2}{\sigma_{i+1}^2} \right)^{\frac{1}{2}} - \frac{\sqrt{\bar{\alpha}_t}\sigma_0}{\sigma_t} \Big)^2 + \sigma_0^2 - \frac{\bar{\alpha}_t \sigma_0^4}{\sigma_t^2} \tag{17}$$

$$= \sigma_0^2 \Big( f_t^\rho - \frac{\sqrt{\bar{\alpha}_t}\sigma_0}{\sigma_t} \Big)^2 + \sigma_0^2 - \frac{\bar{\alpha}_t \sigma_0^4}{\sigma_t^2}, \tag{30}$$

$$P_t^\rho = \Big( \sigma_0 - \sigma_0 \prod_{i=0}^{t-1} \left( \rho + (1-\rho)\alpha_{i+1} \frac{\sigma_i^2}{\sigma_{i+1}^2} \right)^{\frac{1}{2}} \Big)^2 \tag{18}$$

$$= \sigma_0^2 \big( 1 - f_t^\rho \big)^2, \tag{31}$$

where $f_t^\rho := \prod_{i=0}^{t-1} \left( \rho + (1-\rho)\alpha_{i+1} \frac{\sigma_i^2}{\sigma_{i+1}^2} \right)^{\frac{1}{2}}$.

From Zhang et al. (2021), the optimal RDP tradeoff for the scalar Gaussian source $X \sim \mathcal{N}(\mu_0, \sigma_0)$ is

$$R(D, P) = \begin{cases} \frac{1}{2} \log \frac{\sigma_0^2(\sigma_0 - \sqrt{P})^2}{\sigma_0^2(\sigma_0 - \sqrt{P})^2 - (\sigma_0^2 + (\sigma_0 - \sqrt{P})^2 - D)^2/4} & \text{if } \sqrt{P} < \sigma_0 - \sqrt{|\sigma_0 - D|}, \\ \max\{\frac{1}{2} \log \frac{\sigma_0^2}{D}, 0\} & \text{if } \sqrt{P} \geq \sigma_0 - \sqrt{|\sigma_0 - D|}. \end{cases}$$

First, for $0 < \rho \leq 1$, we have $f_t^\rho := \prod_{i=0}^{t-1} \left( \rho + (1-\rho)\alpha_{i+1} \frac{\sigma_i^2}{\sigma_{i+1}^2} \right)^{\frac{1}{2}}$ falls in $(\frac{\sqrt{\bar{\alpha}_t}\sigma_0}{\sigma_t}, 1]$, which implies that $\sqrt{P_t^\rho} < \sigma_0 - \sqrt{|\sigma_0 - D_t^\rho|}$. Plugging Eq. (30) and Eq. (31) into the first case of $R(D, P)$ function, we have

$$R(D_t^\rho, P_t^\rho) = \frac{1}{2} \log \left( \frac{\sigma_0^2 \cdot \sigma_0^2 (f_t^\rho)^2}{\sigma_0^2 \cdot \sigma_0^2 (f_t^\rho)^2 - \big( \sigma_0^2 + \sigma_0^2 (f_t^\rho)^2 - \sigma_0^2 (f_t^\rho - \frac{\sqrt{\bar{\alpha}_t}\sigma_0}{\sigma_t})^2 - \frac{1-\bar{\alpha}_t}{\sigma_0} \big)^2/4} \right)$$

$$= \frac{1}{2} \log \left( \frac{4\sigma_0^4 (f_t^\rho)^2}{4\sigma_0^4 (f_t^\rho)^2 - 4\frac{\bar{\alpha}_t \sigma_0^2}{\sigma_t^2} (f_t^\rho)^2 \sigma_0^4} \right)$$

$$= \frac{1}{2} \log \left( \frac{\bar{\alpha}_t}{1 - \bar{\alpha}_t} \sigma_0^2 + 1 \right).$$

For the second case, when $\rho = 0$, $\sqrt{P_t^\rho} = \sigma_0 - \sqrt{|\sigma_0 - D_t^\rho|}$, the distortion now become the MMSE value

$$D_t^0 = \sigma_0^2 - \frac{\bar{\alpha}_t \sigma_0^4}{\sigma_t^2},$$

and the optimal rate is

$$R(D_t^0, P_t^0) = \frac{1}{2} \log \frac{\sigma_0^2}{D_t^0} = \frac{1}{2} \log \left( \frac{\bar{\alpha}_t}{1 - \bar{\alpha}_t} \sigma_0^2 + 1 \right).$$

We can observe that in both cases, the optimal rate $R(D, P)$ given distortion level $D_t^\rho$ and perception level $P_t^\rho$ is $I_t = \frac{1}{2} \log \left( \frac{\bar{\alpha}_t}{1 - \bar{\alpha}_t} \sigma_0^2 + 1 \right)$, which coincides with the asymptotical achievable rate $I_t$ provided by PFR when transmitting $Z_t \sim \sqrt{\bar{\alpha}_t} X + \sqrt{1 - \bar{\alpha}_t} N$.

# E    MORE EXPERIMENTAL RESULTS

## E.1    CIFAR-10 DATASET

**Experimental details:** We use a pre-trained diffusion model provided by a third-party repository[3], which closely follows the original DDPM setup Ho et al. (2020). The training details can be found

---

[3] https://github.com/w86763777/pytorch-ddpm

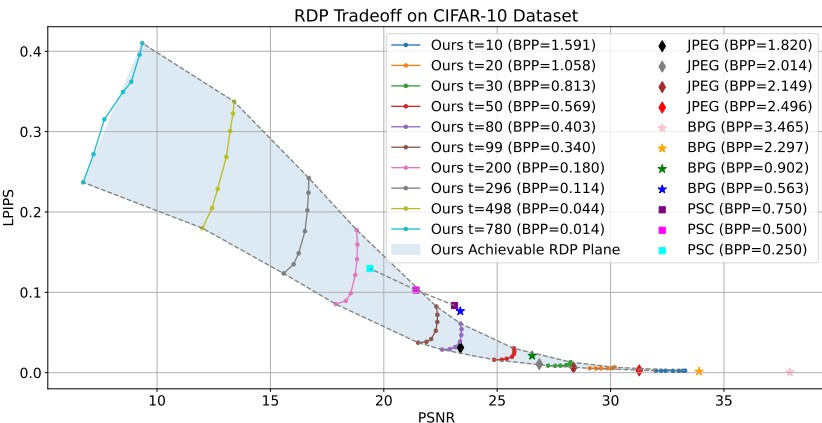

Figure 8: RDP curves on CIFAR-10 using PSNR vs. LPIPS.

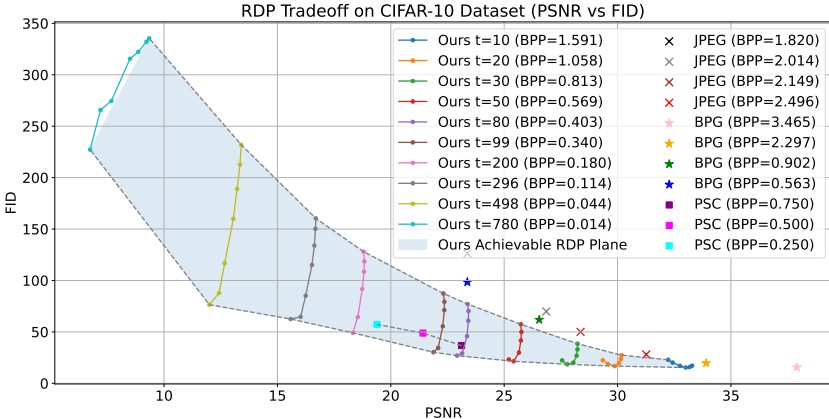

Figure 9: RDP curves on CIFAR-10 using PSNR vs. FID.

in the original repository. For FID computation, we compress and reconstruct 2,000 samples, extract features using the pre-trained Inception network, and compute the Fréchet distance using the empirical means and covariances of real and generated features. To generate the RDP curves in Figure 4, we vary the score-scaling parameter $\rho \in \{0.5, 0.6, 0.7, 0.8, 0.9, 0.95, 1\}$. All experiments are conducted on a single NVIDIA A100 GPU.

**More results:** In Figures 8 and 9, we present additional RDP results using PSNR as the distortion metric, and LPIPS and FID as the perception metrics. The tradeoff behavior remains consistent with our theoretical analysis: a higher $\rho$ leads to better perceptual quality but lower PSNR. We also provide qualitative examples in Figure 10 showcasing reconstructions under different $t$ and $\rho$. These results further demonstrate the smooth and controllable tradeoff enabled by our method.

### E.2 KODAK AND DIV2K DATASETS

#### E.2.1 EXPERIMENTAL DETAILS

We evaluate our method using two high-resolution datasets: Kodak (24 images with size of $768 \times 512 \times 3$) and DIV2K validation (100 images at 2K resolution). We use pre-trained latent diffusion models, including Stable Diffusion (versions 1.5, 2.1, and SDXL)(Rombach et al., 2022) and Flux (Black-Forest-Labs et al., 2025). For DDCM (Ohayon et al., 2025), we fol-

$t = 500 \quad t = 300 \quad t = 200 \quad t = 100 \quad t = 50 \quad t = 10$

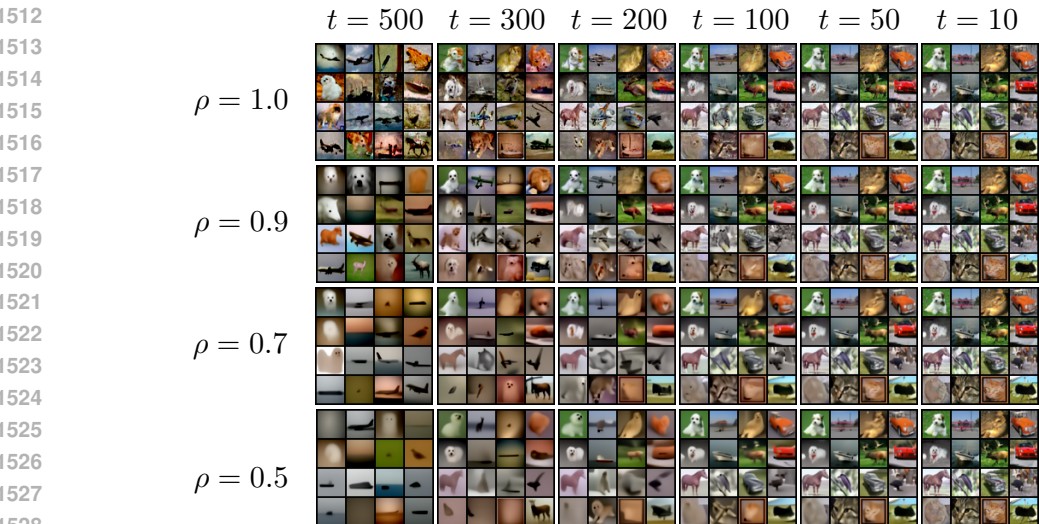

Figure 10: Sample reconstructions on CIFAR-10 under varying $t$ and $\rho$. Higher $\rho$ produces more vivid but less faithful images.

low their official implementation and configurations on the rate control. Specifically, we set $(K, M, C) = (256, 1, 1), (8192, 1, 1), (2048, 2, 3), (2048, 3, 3)$ according to their paper to obtain the reported points.

In Figure 6, we choose different timesteps $t \in \{498, 296, 200, 99, 80, 50, 30, 20, 10\}$. Note that other choices of $t$ along the process of reverse diffusion sampling are also possible. The score-scaling parameter $\rho$ is varied on the latent space to explore the distortion-perception tradeoff under different compression levels $t$. Specifically, we use the following $\rho$ values for each model:

- Stable Diffusion 2.1: $\rho \in \{0.75, 0.83, 0.85, 0.88, 0.9, 0.92, 0.93, 0.95, 1\}$,
- Flux: $\rho \in \{0.7, 0.75, 0.8, 0.85, 0.88, 0.9, 0.92, 0.95\}$.

All experiments are run on a single NVIDIA A100 GPU. The CUDA-accelerated implementation of the PFR algorithm from Vonderfecht & Liu (2025) is applied as the RCC encoder.

### E.2.2   MORE RESULTS

**Latency and model size:** We report the model sizes and encoding/decoding latencies of different methods in Table 2. The encoding/decoding time is measured on Kodak dataset, and all experiments are run on a single NVIDIA A100 GPU. When the bitrate increases, the encoding time increases while the decoding time decreases due to the fewer steps required in the ODE reverse sampling process. The overall running time is approximately 2.31-9.47 seconds per image. While this is slower than some more lightweight models like HiFiC, it remains acceptable. Meanwhile, our scheme is compatible with any RCC coding method. Thus, the encoding time can be reduced by using more efficient RCC coding methods. The decoding time can also be reduced by employing improved diffusion model sampling methods.

Furthermore, our framework is training-free. With a single pre-trained model, we can cover a wide range of RDP tradeoffs, thereby saving significant training time and model storage costs. For example, to cover 10 different bitrates and 5 different distortion-perception tradeoffs, HiFiC or CDC would need to train and store 50 distinct models, resulting in a storage cost five times larger than that of our method.

**More metrics:** We provide additional RDP results using PSNR-LPIPS curves for SD2.1 and Flux across both datasets in Figure 11. We also report the FID on the Kodak dataset and plot the rate-MSE-FID tradeoff in Figure 12. Similar trends are observed as in MSE-LPIPS results. Note that the computation of FID here follows Mentzer et al. (2020); Ohayon et al. (2025), wherein $64 \times 64$ patches are extracted from the high resolution images to compute the FID scores on these patches. We also provide tables of numerical results for each metric in Table 3 and Table 4.

Table 2: Comparison of Different Methods

| Method | # of Parameters | Encoding (s) | Decoding (s) | Overall Time (s) |
|---|---|---|---|---|
| HiFiC | 181M | 0.67 | 1.53 | 2.20 |
| CDC | 53.8M | 0.07 | 3.25 | 3.32 |
| Ours (SD2.1) | 950M | 0.22-9.14 | 0.33-2.10 | 2.31-9.47 |
| DDCM (SD2.1) | 950M | 37.76 | 37.91 | 75.67 |

Table 3: PSNR and LPIPS values across different $\rho$ and $t$ values on Kodak and Div2k datasets (SD 2.1)

| (PSNR/LPIPS) BPP | $t=10$ 0.125 | $t=30$ 0.080 | $t=50$ 0.063 | $t=80$ 0.048 | $t=99$ 0.042 | $t=200$ 0.024 | $t=296$ 0.015 | $t=498$ 0.006 |
|---|---|---|---|---|---|---|---|---|
| $\rho=0.83$ | 26.578 0.088 | 25.726 0.110 | 25.165 0.129 | 24.509 0.154 | 24.165 0.169 | 22.719 0.250 | 21.578 0.338 | 19.219 0.565 |
| $\rho=0.85$ | 26.578 0.087 | 25.725 0.110 | 25.156 0.128 | 24.505 0.151 | 24.157 0.166 | 22.713 0.243 | 21.583 0.326 | 19.278 0.550 |
| $\rho=0.88$ | 26.578 0.087 | 25.725 0.109 | 25.153 0.127 | 24.500 0.150 | 24.152 0.163 | 22.707 0.234 | 21.581 0.312 | 19.383 0.524 |
| $\rho=0.9$ | 26.577 0.087 | 25.722 0.108 | 25.147 0.124 | 24.490 0.146 | 24.141 0.159 | 22.675 0.223 | 21.555 0.293 | 19.360 0.490 |
| $\rho=0.92$ | 26.576 0.087 | 25.720 0.108 | 25.146 0.125 | 24.494 0.146 | 24.137 0.159 | 22.669 0.222 | 21.561 0.289 | 19.382 0.474 |
| $\rho=0.93$ | 26.576 0.086 | 25.715 0.107 | 25.142 0.124 | 24.487 0.145 | 24.132 0.157 | 22.687 0.218 | 21.560 0.282 | 19.378 0.462 |
| $\rho=0.95$ | 26.577 0.086 | 25.716 0.107 | 25.138 0.123 | 24.476 0.144 | 24.126 0.156 | 22.661 0.214 | 21.533 0.271 | 19.327 0.428 |
| $\rho=1$ | 26.566 0.086 | 25.684 0.106 | 25.089 0.122 | 24.411 0.141 | 24.053 0.153 | 22.531 0.205 | 21.328 0.255 | 18.848 0.375 |

Table 4: MSE and FID values across different $\rho$ and $t$ values on Kodak dataset (SD 2.1)

| MSE ($\times 10^{-3}$) /FID | $t$ values | | | | | | |
|---|---|---|---|---|---|---|---|
| | 10 | 30 | 50 | 99 | 200 | 296 | 498 |
| $\rho=0.75$ | 3.242 24.665 | 3.784 29.253 | 4.211 33.367 | 5.043 43.615 | 6.733 73.500 | 8.584 106.041 | 13.414 171.234 |
| $\rho=0.83$ | 3.239 24.158 | 3.782 27.854 | 4.209 31.040 | 5.038 38.579 | 6.718 59.002 | 8.539 89.957 | 13.458 147.872 |
| $\rho=0.85$ | 3.238 24.026 | 3.783 27.529 | 4.210 30.425 | 5.040 37.295 | 6.725 55.286 | 8.549 84.503 | 13.509 141.376 |
| $\rho=0.88$ | 3.238 23.823 | 3.784 27.026 | 4.213 29.601 | 5.046 35.397 | 6.744 49.927 | 8.586 75.094 | 13.619 130.549 |
| $\rho=0.90$ | 3.237 23.692 | 3.786 26.734 | 4.216 29.067 | 5.054 34.241 | 6.766 46.616 | 8.628 68.504 | 13.732 120.375 |
| $\rho=0.92$ | 3.237 23.544 | 3.788 26.405 | 4.219 28.509 | 5.062 33.085 | 6.795 43.259 | 8.688 61.814 | 13.921 110.608 |
| $\rho=0.93$ | 3.238 23.477 | 3.789 26.250 | 4.222 28.266 | 5.068 32.538 | 6.813 41.707 | 8.727 58.462 | 14.038 104.204 |
| $\rho=0.95$ | 3.237 23.358 | 3.791 25.933 | 4.226 27.784 | 5.080 31.470 | 6.855 38.907 | 8.815 52.233 | 14.345 88.912 |

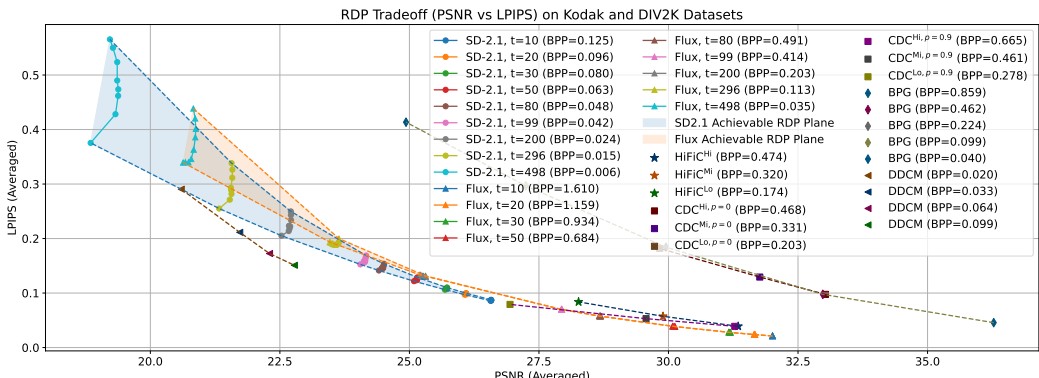

Figure 11: RDP curves for Kodak and DIV2K using PSNR vs. LPIPS under SD2.1 and Flux.

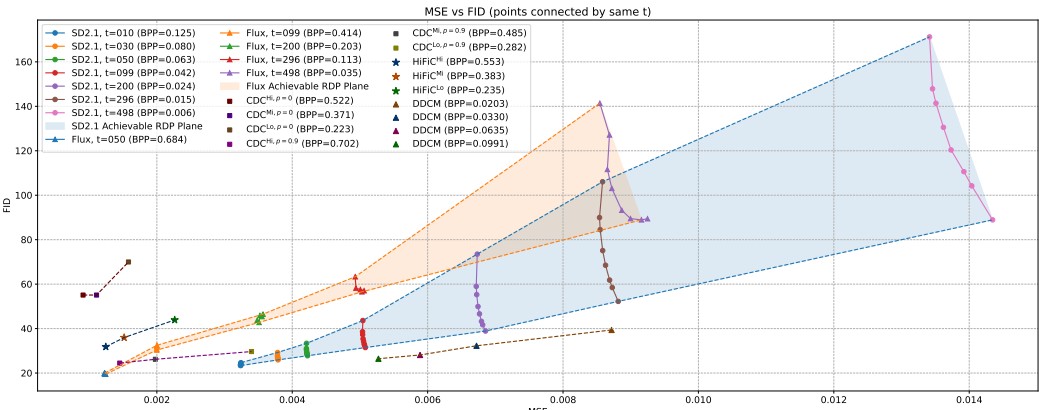

Figure 12: RDP curves for Kodak dataset using MSE vs. FID under SD2.1 and Flux.

**More models:** We also include the distrotion and perception performance results for SD1.5 and SDXL under $\rho = 1$ in Figure 13. We can observe that the R-D and R-P performances of SD1.5 and SDXL are inferior to SD2.1. Thus, we choose SD2.1 in our experiments when comparing with the benchmark.

**More samples:** Figures 14 and 15 presents more sample reconstructions on Kodak and DIV2K datasets with diverse $\rho$ selections and bitrate levels against baselines. Figure 16 depicts the visual changes in reconstructions provided by Flux under different $t$ and $\rho$. Figures 17 and 18 samples with high resolution details. Note that we also provide an interactive demo online [4] to help the reader compare the details of images for different values of $t$ and $\rho$. These qualitative results further illustrate the smooth and controllable RDP tradeoff achieved by our method.

---

[4]https://diffrdp.github.io/

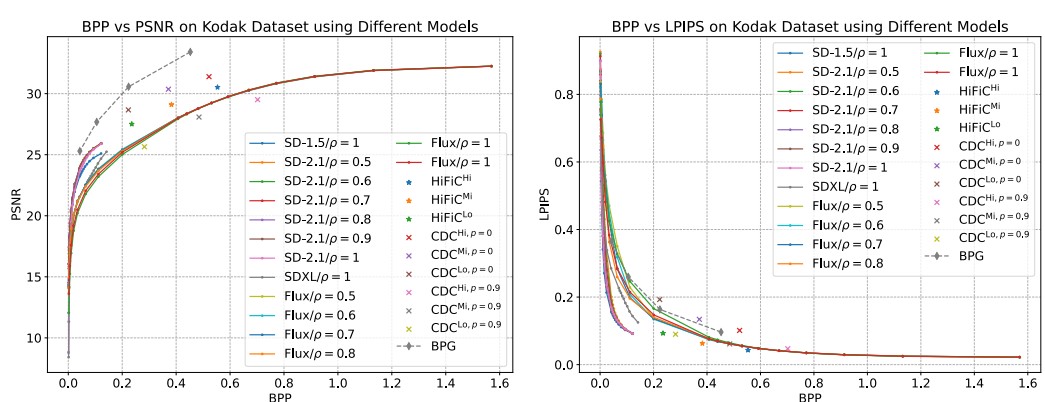

Figure 13: RDP metrics under $\rho = 1$ for SD1.5 and SDXL.

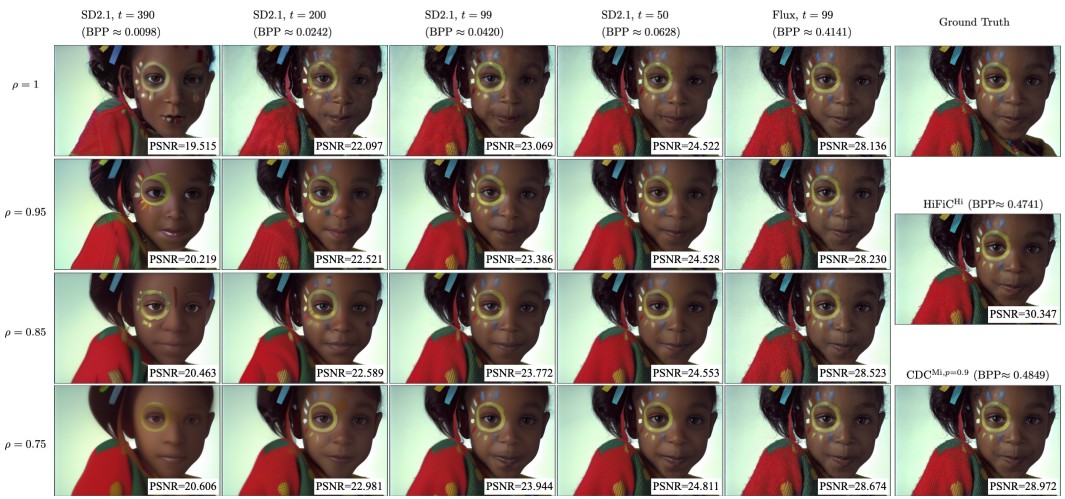

Figure 14: Sample reconstructions on Kodak dataset under different $t$ and $\rho$.

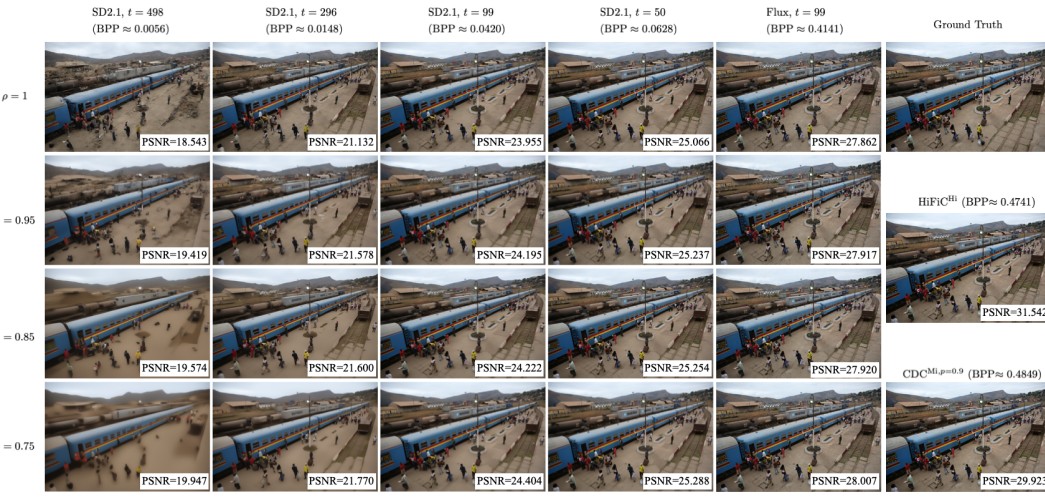

Figure 15: Sample reconstructions on DIV2K dataset under different $t$ and $\rho$.

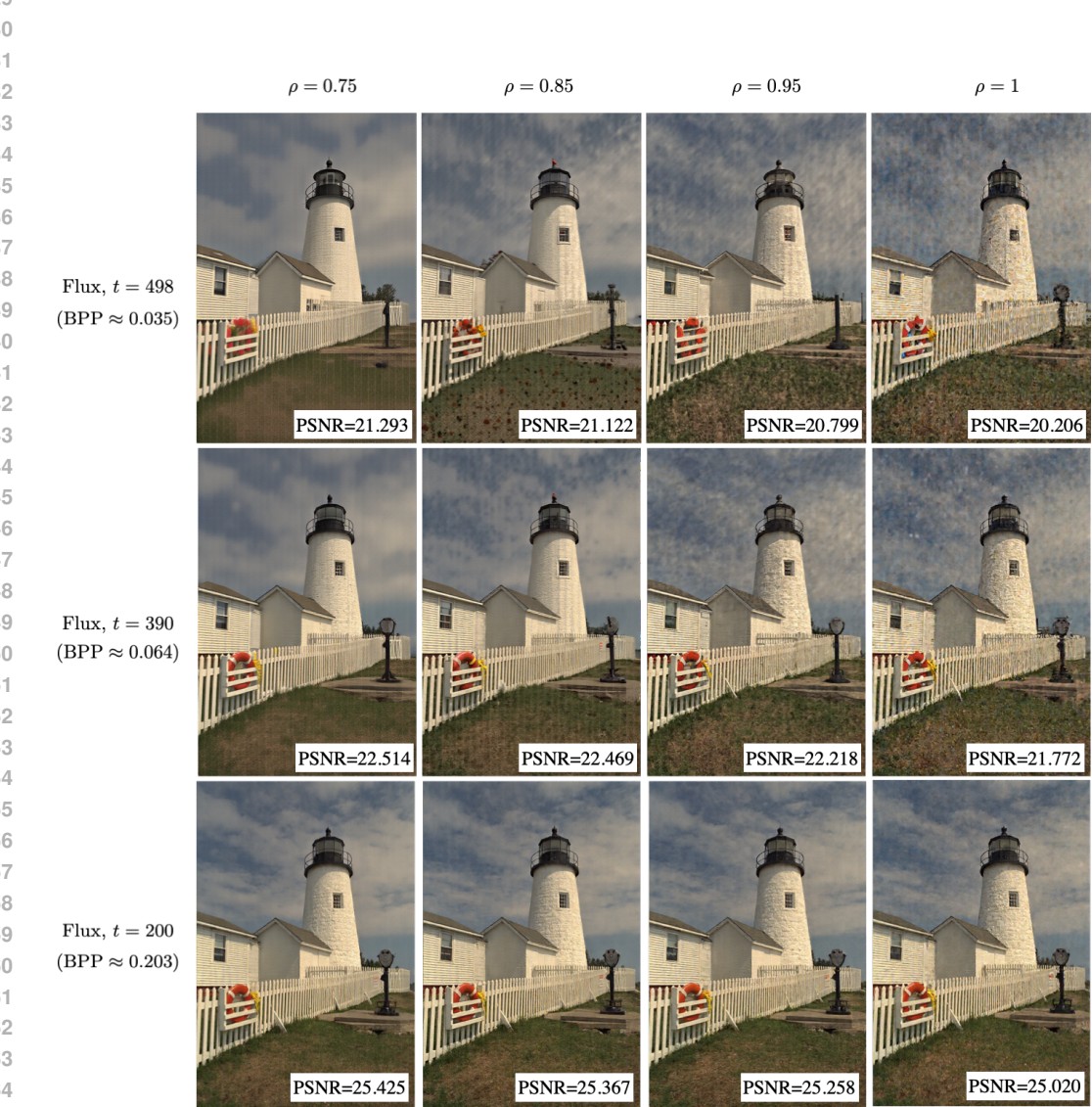

Figure 16: Sample reconstructions provided by Flux under different $t$ and $\rho$.

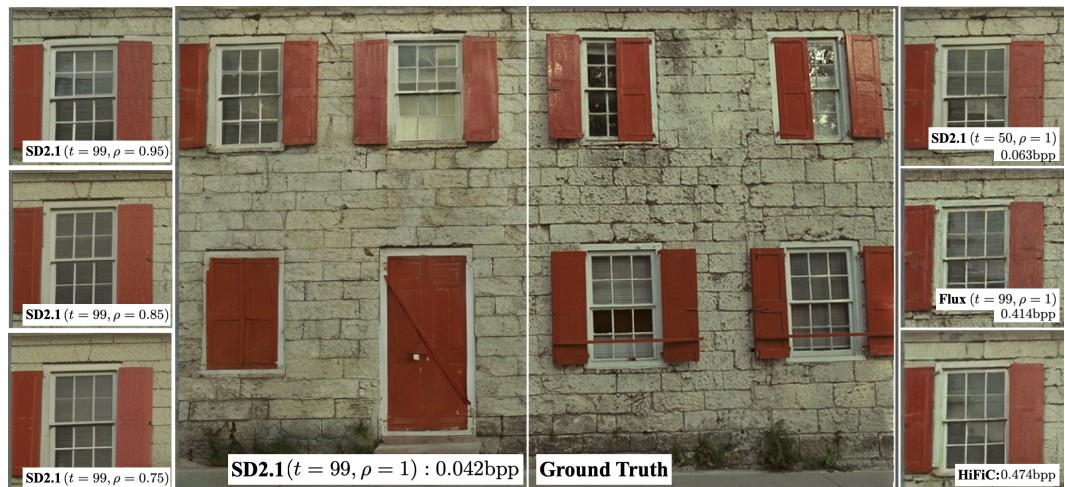

Figure 17: Sample reconstructions with high resolution details under different $t$ and $\rho$.

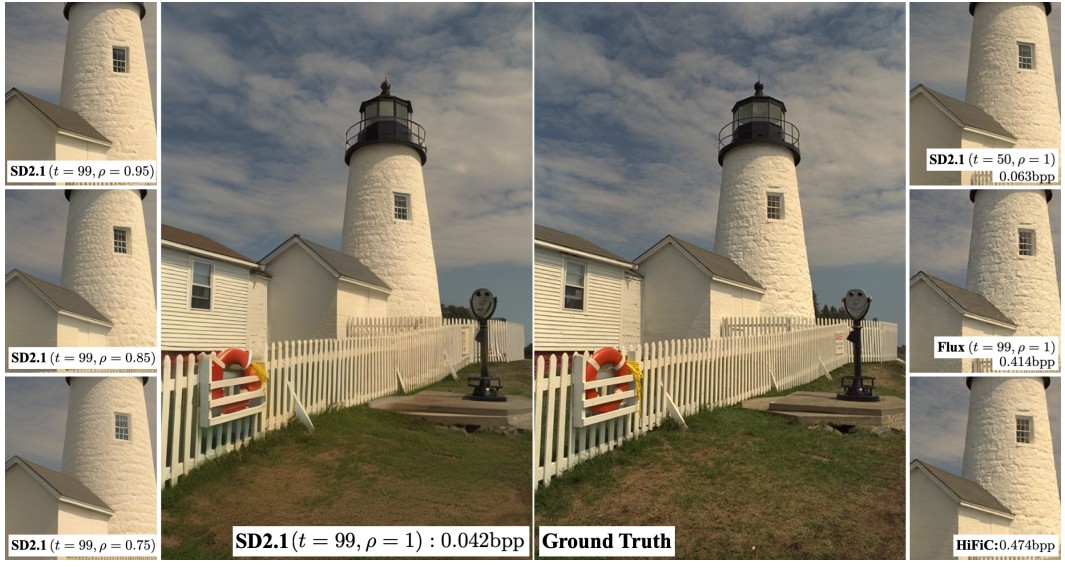

Figure 18: Sample reconstructions with high resolution details under different $t$ and $\rho$.

