# OpenReview forum: "Training-Free Rate-Distortion-Perception Traversal With Diffusion"
_ICLR.cc/2026/Conference — Submitted to ICLR 2026_

### Official Review · Reviewer_Z2xu · 2025-10-31

**Soundness:** 3
**Presentation:** 4
**Contribution:** 2
**Rating:** 4
**Confidence:** 4

**Summary:**

This paper propose a lossy image compression scheme based on the DiffC algorithm to traverse the rate-distortion-perception frontier. While DiffC achieves perfect realism for reconstructions, the current work modifies the decoder by simulating a score-scaled PF-ODE (eq. 5). The authors further prove optimality under Wasserstein-2 perception measure of the proposed scheme for scalar Gaussian sources. In practice, the conditional distributions are approximated by a pre-trained diffusion model. Shared seed is required for encoder and decoder to implement the algorithm. Experiments are implemented on CIFAR-10, Kodak, and Div2k under LPIPS perception measure.

**Strengths:**

The paper provides a practical image lossy compression scheme using pre-trained diffusion model to traverse the RDP frontier. The main theoretical contribution is Eq. (5), which traverses two extremes: reverse SDE in Eq. (3) and PF-ODE in Eq. (4). For Gaussian cases, the proposed scheme is shown to achieve optimality.

The proposed scheme is training-free, and practical algorithms are given and implemented on CIFAR-10, Kodak, and Div2k datasets to demonstrate effectiveness.

**Weaknesses:**

I am most concerned with the motivation for traversing the RDP frontier. As is mentioned in the paper, a GAN-based scheme is available in Zhang et al. (2021). Comparisons between the proposed scheme and the GAN-based scheme regarding practical constraints such as latency, compute, and memory usage are not provided.

**Questions:**

1. In Eq. (5), when $\rho=0$, should $Z_t$ follow $p_{T_c}$? What are the conditions for Eq. (5) to hold at $\rho=0$?

2. Do the optimality results and the results in Lemma 1 hold for Gaussian cases because of the coincident $p_{T_c}=W_\tau$ therein? Could you provide some insights into more complex sources such as mixture Gaussian?

3. Does the shared seed need to be entropy coded in Alg 1 and Alg 2?

4. In line 320, should $z_k = \bar{z}_k^{(c_k)}$ be $z_t =...$?

5. In the experiment, LPIPS is used as the perception measure, while in section 3 and section 4, W2 is used. Why?

6. Could you compare with the GAN-based scheme (Zhang et al. 2021) in terms of latency, compute, and memory?

---

> ### Author Response · Authors · 2025-11-21
> **Response to Reviewer 5 (Part 1)**
>
> **On latency and memory usage:** We thank the reviewer for the constructive comments. We have conducted experiments to compare the latency and memory usage of our method with other baselines, specifically focusing on the cost to store model parameters. As Zhang et al. (2021) primarily conducted experiments on small datasets like MNIST and SVHN, we instead use the state-of-the-art GAN-based method HiFiC (Mentzer et al., 2020) as a baseline for high-resolution images.
>
> The detailed results are shown in the table below.
>
> | Method       | # of Parameters | Encoding (s) | Decoding (s) | Overall Time (s) |
> | :----------- | --------------: | -----------: | -----------: | ---------------: |
> | HiFiC        |            181M |         0.67 |         1.53 |             2.20 |
> | CDC          |           53.8M |         0.07 |         3.25 |             3.32 |
> | Ours (SD2.1) |            950M |    0.22-9.14 |    0.33-2.10 |        2.31-9.47 |
> | DDCM (SD2.1) |            950M |        37.76 |        37.91 |            75.67 |
>
> As the bitrate increases, the encoding time of our scheme increases, while the decoding time decreases due to fewer steps in the ODE reverse sampling. The overall running time, ranging from 2.31 to 9.47 seconds per image, is slower than lighter models like HiFiC but remains acceptable. Furthermore, our scheme is compatible with any RCC coding method, implying that the encoding time can be further reduced by employing more efficient RCC coding methods. Similarly, the decoding time can be reduced through improved sampling methods for diffusion models. For instance, compared to DDCM, which adopts the original DDPM sampling method with 1000 steps, our framework achieves more than an 18x speedup in decoding time.
>
> In practice, the training-free nature of our framework significantly enhances its efficiency regarding the memory cost for storing model parameters. With a single pre-trained model, we can cover a wide range of RDP tradeoffs, thereby saving both training time and model storage costs. For example, to cover 10 different bitrates and 5 different distortion-perception tradeoffs, HiFiC or CDC would need to train and store 50 distinct models (approximately 2.5-9GB total), resulting in a storage cost 2-9 times larger than our method.
>
> **Response to Question 1:** In our scheme, $Z_t$ is produced by the RCC encoder, which guarantees that $Z_t \sim \sqrt{\bar{\alpha}_t}X + \sqrt{1-\bar{\alpha}_t}N$. The ODE decoder then reconstructs $Z_0$ from $Z_t$, where $\rho$ controls the DP tradeoff. Therefore, the initial condition $Z_t$ is irrelevant to $\rho$.
>
> **Response to Question 2:** In the proof of Proposition 2 and Theorem 3, we show that in the Gaussian case, the optimization over all possible conditional distributions $p_{\hat{X}|Z_t}$ can be restricted to be a linear function of $Z_t$ without loss of optimality. Thus, we can derive a closed-form expression for the DP function in the multivariate Gaussian case and prove its optimality. In general cases, it is difficult to derive a closed-form expression due to the complex structure of the source distribution.
>
> Intuitively, we can observe that the variance of the reconstruction distribution depends on $\rho$. When $\rho$ is small, the variance is small, leading to better distortion. When $\rho$ is large, the variance is large, leading to better perception. This explains why we can traverse the DP tradeoff by adjusting $\rho$. We illustrate this phenomenon using a mixture of Gaussian sources where $p_X = w_1*\mathcal{N}(u_1, \sigma_1^2) + w_2*\mathcal{N}(u_2, \sigma_2^2)$ and $Y=\sqrt{\bar{\alpha}_t}X + \sqrt{1-\bar{\alpha}_t}N$ with $N\sim\mathcal{N}(\mathbf 0, \mathbf I)$. The effect of $t$ and $\rho$ on the sampling trajectories of the ODE on the mixture Gaussian source is shown in Figure R1 in this link https://mixturegaussianfig.github.io/.
>
> **Response to Question 3:** Common randomness is assumed to be available at both sides in advance. In practice, one can first entropy encode the random seed once to ensure the encoder and decoder are synchronized before the algorithm starts.

---

> ### Author Response · Authors · 2025-11-21
> **Response to Reviewer 5 (Part 2)**
>
> **Response to Question 4:** No. We update $\mathbf z_k$ progressively in each step, and output $\mathbf z_t$ only at the very end.
>
> **Response to Question 5:** In our theoretical analysis, we use W2 as the perception measure due to its concise mathematical expression and good theoretical properties. LPIPS is a learned perceptual metric that computes the distance between deep features extracted from a pre-trained neural network. It does not have a closed-form expression, making it difficult to use in theoretical analysis.
>
> In practice, W2 is very hard to compute for high-dimensional data like images, while LPIPS is widely used as a perceptual metric that aligns well with human perception. Thus, we use LPIPS in our experiments to evaluate the perceptual quality of the reconstructed images.
>
> **Response to Question 6:** As discussed in the response to the weakness, we have conducted experiments to compare the latency and memory cost to store the model parameters of our method with other baselines. Zhang et al. (2021) only conducted experiments on small datasets like MNIST and SVHN. Here, we use the state-of-the-art GAN-based method HiFiC (Mentzer et al., 2020) as a baseline for high-resolution images.
>
> The detailed results are recalled below.
>
> | Method       | # of Parameters | Encoding (s) | Decoding (s) | Overall Time (s) |
> | :----------- | --------------: | -----------: | -----------: | ---------------: |
> | HiFiC        |            181M |         0.67 |         1.53 |             2.20 |
> | CDC          |           53.8M |         0.07 |         3.25 |             3.32 |
> | Ours (SD2.1) |            950M |    0.22-9.14 |    0.33-2.10 |        2.31-9.47 |
> | DDCM (SD2.1) |            950M |        37.76 |        37.91 |            75.67 |
>
> >References:
> >
> > George Zhang, et al., “Universal rate-distortion-perception representations for lossy compression.” NeurIPS, 2021.
> >
> > Fabian Mentzer, et al., “High-fidelity generative image compression.” NeurIPS,2020.

---

> ### Comment · Reviewer_Z2xu · 2025-11-27
>
> Thank you for the comprehensive reply. After reading the new experiments, I still do not see convincing evidence that the proposed method outperforms HiFiC in either model size or latency, so the practical motivation remains weak. In addition, the theory is hard to follow and is largely confined to the Gaussian case with the W2 distance. I also remain unconvinced by the claim that “W2 is very hard to compute for high-dimensional data such as images.” For these reasons, I will maintain my original score.

---

> > ### Author Response · Authors · 2025-12-02
> > **Further Response to Reviewer 5**
> >
> > We thank the reviewer for the prompt response and additional comments. We highlight the following points.
> >
> > **On efficiency of model size and latency:** As discussed in the revised paper and our previous response, the training-free nature of our framework offers the advantage of significantly reducing the memory cost for storing model parameters compared to HiFiC and CDC. For instance, to cover 10 different bitrates and 5 distinct distortion-perception tradeoffs, HiFiC or CDC would necessitate training and storing 50 individual models (approximately 2.5-9GB total), **leading to a storage cost 2-9 times larger than our method**. The total model size required to cover 10 different bitrates and 5 distinct distortion-perception tradeoffs is summarized in the following table:
> >
> > > Table: Total Number of Parameters Required to Cover 10 Different Bitrates and 5 Distinct DP Tradeoffs
> >
> > | Method       | Total # of Parameters |
> > | :----------- | --------------------: |
> > | HiFiC        |         181x50=9,050M |
> > | CDC          |        53.8x50=2,690M |
> > | Ours (SD2.1) |                  950M |
> > | DDCM (SD2.1) |                  950M |
> >
> > **On presentation of the theoretical results:** We have revised the main result sections to more clearly state the distinct results on DP and RDP. We also revised the Appendices, specifically:
> >
> > *   Appendix A was split to provide a clearer explanation of the ODE discretization and the proof of Lemma 1. The discretization of the score-scaled PF-ODE part was moved to a separate appendix section (now becomes Appendix A). The proof of Lemma 1 is now discussed separately in Appendix B.
> > *   A discussion of two extreme cases ($\rho=0$ and $\rho=1$) has been included within Lemma 1 and Appendix B.3 of the revised manuscript to offer further insights into the ODE reconstruction.
> > *   The Proof of Theorem 4 has been revised to include additional details regarding the one-shot coding results.
> >
> > In general, it is challenging to derive closed-form expressions due to the complex structure of the source distribution. Intuitively, the variance of the reconstruction distribution depends on $\rho$. When $\rho$ is small, the variance is small, leading to better distortion. When $\rho$ is large, the variance is large, leading to better perception. This phenomenon is illustrated using a mixture of Gaussian sources where $p_X = w_1\*\mathcal{N}(u_1, \sigma_1^2) + w_2\*\mathcal{N}(u_2, \sigma_2^2)$ and $Y=\sqrt{\bar{\alpha}_t}X + \sqrt{1-\bar{\alpha}_t}N$ with $N\sim\mathcal{N}(\mathbf 0, \mathbf I)$. The effect of $t$ and $\rho$ on the sampling trajectories of the ODE for a mixture Gaussian source is shown in Figure R1, available at this link https://mixturegaussianfig.github.io/.
> >
> > **On W2 distance:** Direct computation of Wasserstein-2 distance involves characterizing the marginal distributions of the original image set and the generated image set, which are generally inaccessible. We could also consider leveraging Kantorovich-Rubinstein duality to compute Wasserstein-1 distance by training a GAN; however, this approach is still computationally costly. Consequently, in most papers across various domains—including denoising problems (Park et al., 2023; Zhu et al., 2024; Freirich et al., 2021), distortion-perception (DP) tradeoff (Wang et al., 2025; Zhu et al., 2025), and compression problems (Yang and Mandt, 2024)—**LPIPS or FID** are commonly employed as perceptual metrics, even for Freirich et al., (2021) and Wang et al., (2025) where W2 distance is used in their theoretical analysis.
> >
> >
> >
> > > Dror Freirich, et al., “A theory of the distortion-perception tradeoff in wasserstein space.” NeurIPS, 2021.
> > >
> > > S. H. Park, et al., "Perception-Oriented Single Image Super-Resolution using Optimal Objective Estimation.” CVPR, 2023.
> > >
> > > Qiwen Zhu, et al., “Perceptual-Distortion Balanced Image Super-Resolution is a Multi-Objective Optimization Problem.” ACMMM, 2024.
> > >
> > > Yuhan Wang, et al., “Traversing Distortion-Perception Tradeoff using a Single Score-Based Generative Model.”, CVPR, 2025.
> > >
> > > Yuanzhi Zhu, et al., “OFTSR: One-Step Flow for Image Super-Resolution with Tunable Fidelity-Realism Trade-offs.”, arXiv, 2025.
> > >
> > > Ruihan Yang and Stephan Mandt, “Lossy Image Compression with Conditional Diffusion Models.” NeurIPS, 2023.

---

### Official Review · Reviewer_D1im · 2025-11-01

**Soundness:** 3
**Presentation:** 4
**Contribution:** 3
**Rating:** 6
**Confidence:** 3

**Summary:**

This paper proposes a training-free compression framework that allows traversal of the Rate-Distortion-Perception tradeoff using pre-trained diffusion models. It uses a reverser-channel-coding encoder and a probability-flow ODE based decoder. The decoder allows for he distortion-perception tradeoff whereas the encoder can vary the compression rate. Along with theoretical proofs the paper also shows empirical results on multiple datasets using a pre-trained diffusion model.

**Strengths:**

The paper is well-written and easy to follow.

1. The idea to re-purpose a trained diffusion model for adaptive compression is well-motivated and allows to overcome the expensive retraining of decoders.

2. The factors controlling the RDP tradeoff - sampling step and scaling are intuitive and easy to follow.

Overall, the paper is exploring a useful idea of training-free RDP traversal. Both the theoretical and empirical justifications look promising.

**Weaknesses:**

1. Although the method described in the paper is training-free, it appears to be expensive during inference-time as it needs multiple ODE iterations to traverse the RDP  curve and the pre-trained model itself might be much more expensive than other GAN/hybrid codecs.

2. An ablation study disentangling the effect of RCC encoder and the PF Decoder could be useful here.  That is, how much contribution towards rate-distortion comes from RCC encoder vs the PF-ODE decoder. For example, fixing one parameter (say the scale) and varying the encoder parameter, would the distortion perception curve move around much?

Looking forward to authors' responses and will adjust my score accordingly.

**Questions:**

Same as weaknesses.

---

> ### Author Response · Authors · 2025-11-21
> **Response to Reviewer 4**
>
> **1. On latency and computational complexity:** We thank the reviewer for the constructive comments. We have conducted experiments to compare the latency of our method with other baselines. The detailed results are shown in the table below.
>
> | Method       | # of Parameters | Encoding (s) | Decoding (s) | Overall Time (s) |
> | :----------- | --------------: | -----------: | -----------: | ---------------: |
> | HiFiC        |            181M |         0.67 |         1.53 |             2.20 |
> | CDC          |           53.8M |         0.07 |         3.25 |             3.32 |
> | Ours (SD2.1) |            950M |    0.22-9.14 |    0.33-2.10 |        2.31-9.47 |
> | DDCM (SD2.1) |            950M |        37.76 |        37.91 |            75.67 |
>
>  When the bitrate increases, the encoding time increases while the decoding time decreases due to fewer steps in the ODE reverse sampling. The overall running time for our method is approximately 2.31-9.47s per image. While this is slightly slower than some lightweight models like HiFiC, it remains acceptable. We would like to point out that our scheme is compatible with any RCC coding method, meaning the encoding time can be further reduced by utilizing more efficient RCC coding methods. Similarly, the decoding time can be reduced by using an improved sampling method for the diffusion model. For instance, compared to DDCM, which adopts the original DDPM sampling method with 1000 steps, our framework achieves more than an 18x speedup in decoding time.
>
>  Furthermore, our framework is training-free. With a single pre-trained model, we can cover a wide range of RDP tradeoffs, thereby saving significant training time and model storage costs. For example, to cover 10 different bitrates and 5 different distortion-perception tradeoffs, HiFiC or CDC would need to train and store 50 distinct models, resulting in a storage cost five times larger than that of our method.
>
> **2. On the effect of $\rho$ and $t$:** We thank the reviewer for the insightful comments. In our framework, the parameter $t$ primarily governs the coding rate, while $\rho$ controls the distortion-perception tradeoff.
>
> To elaborate:
>
> *   Varying $\rho$ with fixed $t$: When we fix the coding rate (by setting a constant $t$) and adjust $\rho$, we can traverse the distortion-perception curve. Each curve presented in Figure 6 in the revised paper corresponds to a fixed $t$ and varying $\rho$.
> *   Varying $t$ with fixed $\rho$: On the other hand, if we fix the scaling parameter (by setting a constant $\rho$) and vary the encoder parameter $t$, we can trace out a rate-distortion (and correspondingly, a rate-perception) curve. Figure R2 in this link (https://mixturegaussianfig.github.io/) demonstrates this, where we connect all points with the same $\rho$. Note that increasing $\rho$ shifts the overall curve on the DP plane towards the bottom-right, which signifies a stronger emphasis on perceptual quality, at the expense of distortion metrics.

---

### Official Review · Reviewer_63wi · 2025-11-04

**Soundness:** 2
**Presentation:** 3
**Contribution:** 2
**Rating:** 4
**Confidence:** 4

**Summary:**

This paper introduces a training-free framework for navigating the rate-distortion-perception (RDP) tradeoff in lossy compression using pre-trained diffusion models. The method combines a reverse channel coding (RCC) encoder with a novel score-scaled probability flow ODE decoder, controlled by two parameters: the noise level t (rate) and the score-scaling factor ρ (distortion-perception tradeoff). Theoretically grounded and empirically validated, this approach enables flexible traversal of the entire RDP surface using a single model, achieving a versatile balance between bitrate, reconstruction fidelity, and perceptual quality without any retraining.

**Strengths:**

Training-Free and Generalizable: It directly leverages pre-trained diffusion models without requiring retraining for different rates or trade-offs. A single model, controlled by the two knobs t and ρ, covers a wide R-D-P region, enabling a perception-distortion trade-off.
Rigorous Theoretical Proofs: The paper provides extensive theoretical proofs, significantly enhancing its readability and credibility.

**Weaknesses:**

1. Perceived Lack of Innovation: Overall, the main innovation arguably lies in introducing the parameter ρ on top of the DiffC[1] framework to control the perception-distortion trade-off. While it successfully achieves this trade-off, the core concept may not be considered highly novel, as it does not address the fundamental limitations of this class of methods.
2. High Computational Complexity and Slow Inference: Similar to DiffC[1], this method inherently suffers from significant encoding and decoding latency. This delay increases further at higher bitrates. Compared to some existing, more efficient image compression methods[2,3,4], the advantages of this approach are less pronounced.
3. Engineering Burden of RCC/PFR: While RCC (PFR) is information-theoretically elegant, it requires shared randomness and careful index coding. The paper provides the one-shot vs. asymptotic rate bounds but does not deeply quantify the runtime or bpp overhead resulting from practical PFR implementation aspects like truncation or batching on large images.
[1] Vonderfecht J, Liu F. Lossy Compression with Pretrained Diffusion Models[C]//The Thirteenth International Conference on Learning Representations.
[2]Jia Z, Li J, Li B, et al. Generative latent coding for ultra-low bitrate image compression[C]//Proceedings of the IEEE/CVF Conference on Computer Vision and Pattern Recognition. 2024: 26088-26098.
[3] A. Ke, X. Zhang, T. Chen, M. Lu, C. Zhou, J. Gu, and Z. Ma, “Ultra Lowrate Image Compression with Semantic Residual Coding and Compression-aware Diffusion,” in Proc. Int. Conf. Mach. Learn. (ICML), 2025.
[4] Zhang, Tianyu, et al. "StableCodec: Taming One-Step Diffusion for Extreme Image Compression." arXiv preprint arXiv:2506.21977 (2025).

**Questions:**

1. On the performance of generative models: In your article, I see that both Flux and SDXL demonstrate better performance than SD 2.1. Why would stronger generative models lead to worse performance metrics?
2. Regarding the issue of latency: I'm very curious about potential feasible directions for optimization. Otherwise, it's difficult to assess whether this type of method can be applied in practice.

---

> ### Author Response · Authors · 2025-11-21
> **Response to Reviewer 3**
>
> **1. On the Novelty of Our Method:** We thank the reviewer for the valuable comments. Our scheme is simple but effective. With a single pretrained diffusion model, our scheme provides an efficient way to seamlessly control the compression rate and the distortion-perception tradeoff with two parameters. We would like to highlight the following points regarding the novelty of our method:
>
>  *   From a theoretical perspective, we prove the optimality of our score-scaled ODE and the overall compression framework in controlling DP and RDP tradeoffs for the Gaussian case. To the best of our knowledge, this is the first work that provides a theoretical guarantee on the optimality of traversing the ternary RDP tradeoff with a single pre-trained model.
>  *   From a practical perspective, we demonstrate the effectiveness of our scheme in high-resolution images when applying existing pre-trained models (e.g., Stable Diffusion and Flux). Not only can we flexibly control the RDP tradeoff, but we also achieve superior or comparable performance to existing methods in terms of distortion and perception metrics.
>
> **2. On Computational Complexity and Latency:** We thank the reviewer for the constructive comments. We have conducted experiments to compare the latency of our method with other baselines. The detailed results are shown in the table below.
>
> | Method       | # of Parameters | Encoding (s) | Decoding (s) | Overall Time (s) |
> | :----------- | --------------: | -----------: | -----------: | ---------------: |
> | HiFiC        |            181M |         0.67 |         1.53 |             2.20 |
> | CDC          |           53.8M |         0.07 |         3.25 |             3.32 |
> | Ours (SD2.1) |            950M |    0.22-9.14 |    0.33-2.10 |        2.31-9.47 |
> | DDCM (SD2.1) |            950M |        37.76 |        37.91 |            75.67 |
>
>  When the bitrate increases, the encoding time increases while the decoding time decreases due to the *fewer* steps required in the ODE reverse sampling process. The overall running time is approximately 2.31-9.47 seconds per image. While this is slower than some *more lightweight* models like HiFiC, it remains acceptable. Meanwhile, we would like to point out that our scheme is compatible with any RCC coding method. Thus, the encoding time can be reduced by using more efficient RCC coding methods. The decoding time can also be reduced by employing improved diffusion model sampling methods. For example, compared to DDCM, which adopts the original DDPM sampling method with 1000 steps, our framework is more than 18x faster in decoding time.
>
> **3. On Engineering Burden of RCC/PFR:** We thank the reviewer for the insightful feedback. Regarding shared randomness, in practice, we can agree upon a random seed before the algorithm starts. There are also RCC methods available that do not require common randomness (Li, 2025, Section 4). Our scheme is compatible with any RCC method, provided that samples of $Z_t=\sqrt{\bar{\alpha}_t}X+\sqrt{1-\bar{\alpha}_t}N$ can be transmitted. As for runtime overhead, the encoding phase is indeed a bottleneck in our scheme, as shown in the table in our previous response. However, it is possible to improve by using more advanced RCC methods. We would like to point out that our framework possesses a training-free property. With a single pre-trained model, we can cover a wide range of RDP tradeoffs, thereby saving on training time and model storage costs. For example, to cover 10 different bitrates and 5 different distortion-perception tradeoffs, HiFiC or CDC would need to train and store 50 different models, resulting in a storage cost five times larger than that of our method.
>
> **Response to Question 1:** On the one hand, the coding rate and performance depend on the noise schedule. Flux performs worse than SD2.1 at low bpp but achieves a larger range of the DP curve. On the other hand, a stronger generative model leads to better perception, but not necessarily better distortion due to its strong generative ability.
>
> **Response to Question 2:** As discussed in the previous response, the latency can be reduced from two aspects: The encoding time can be reduced by using more efficient RCC coding methods. The decoding time can be reduced by using an improved sampling method for the diffusion model.
>
> >Reference:
> >
> > Cheuk Ting Li. Channel simulation: Theory and applications to lossy compression and differential privacy. Found. Trends Commun. Inf. Theory, 21(6):847–1106, December 2024. ISSN 1567-2190.

---

### Official Review · Reviewer_y2u7 · 2025-11-06

**Soundness:** 3
**Presentation:** 3
**Contribution:** 2
**Rating:** 4
**Confidence:** 4

**Summary:**

This work proposes a training-free framework for traversing the rate-distortion-perception (RDP) tradeoff using pre-trained diffusion models. The authors largely build on DiffC - a work that proposed to transmit the (diffusion) noisy sample at some timestep $t$, $x_t$, via reverse-channel-coding efficiently, and then showed that it's better on the decoder side to continue to the reconstruction $x_0$ via an ODE (DiffC-F) instead of via ancestral sampling (the SDE, termed DiffC-A).
The key idea in the current paper is to modify the decoder side of DiffC-F by introducing a "score-scaled probability flow ODE (PF-ODE)",
which is a combination of the original DiffC-F and another method which gives the MMSE solution.
Specifically, the PF-ODE includes a parameter $\rho$ which balances between perception (closer to DiffC) and distortion (closer to the MMSE solution), thereby enabling traversal of the distortion-perception tradeoff for a given bitrate. Compression rate is controlled on the encoder side, as in DiffC, by selecting which timestep is transmitted, yielding full traversal of the RDP surface.

The authors provide theoretical results establishing DPR optimality for the scalar Gaussian case of their proposed PF-ODE, though parts of this analysis build on directions already explored in related literature.
Experimentally, they test their method in pixel space on CIFAR-10, and on latent space with SD2.1 and Flux on Kodak and DIV2K demonstrating traversal, against traditional codecs and relatively recent neural methods.

**Strengths:**

- The paper motivation of traversing the RDP tradeoff[1] is clear and, in my opinion, important.
- This paper continues the recent trend (DiffC[2], PSC[3], DDCM[4]) of using a single, pre-trained model to support multiple bitrates in a flexible manner - which is also very important in this reviewer's opinion. Other methods such as HiFiC that need differently trained models for every new bitrate make neural compression less practical in the long run for edge devices.
- The idea of basically combining 2 solutions, one adapted for distortion and one for perception, has not been done recently in neural compression, which is good. *That said, the novelty in the context of RDP is limited, as mentioned in the weaknesses section.*
- The paper gives theoretical guaranties for the full algorithm's RDP optimality in the scalar Gaussian case (Thm 4), and for multivariate sources under an AWGN channel (Thm 3). Theoretical guarantees, although simplified, of algorithms in such a practical field as compression are good and paper should strive more towards that. However, *the novelty of the proofs is again limited, as mentioned in the weaknesses section.*
- Relatively detailed proofs are provided for all claims in the paper.
- The provided demo helps with visualization.
- The paper was generally well-written, with intuition into the core idea.

[1] Yochai Blau and Tomer Michaeli. "Rethinking lossy compression: The rate-distortion-perception
tradeoff". ICML 2019.

[2] Lucas Theis, Tim Salimans, Matthew D. Hoffman, and Fabian Mentzer. "Lossy compression with gaussian diffusion". ArXiv preprint, 2022.

[3] Noam Elata, Tomer Michaeli, and Michael Elad. "PSC: Posterior sampling-based compression". TMLR 2025.

[4] Guy Ohayon, Hila Manor, Tomer Michaeli, and Michael Elad. "Compressed Image Generation with Denoising Diffusion Codebook Models." ICML 2025.

**Weaknesses:**

- The key concept - traversing the RDP curve by using a combined solution of the MMSE (distortion-optimal) and perfect-perception reconstructions - is the main idea in [5]. That work formalized the interpolation mechanism and proved its optimality. While here the combination is of the score (and not the estimator), this manuscript does not acknowledge them at all. Specifically, as the optimality in this paper is concerned with the scalar Gaussian case - are their results (and therefore the optimality) not transferable to your case by a linear transformation? In any case, this should be explicitly discussed to accurately position the actual contribution of this paper (which is more technical in that regard).
- LPIPS is used throughout the paper as a “perception” metric, but LPIPS is a feature-space distortion measure. This is well-documented in the literature, including the original PD tradeoff paper [6] (it's before LPIPS' time, so it appears as MSE between VGG features in Secs. 5-6), and in the RPD paper [1] (See e.g. Sec. 4.2, they even cite LPIPS there directly). Therefore, Figs 3,4,5 and in the App 7,10,11 all show a **distortion-vs-distortion** curve, and **not the RDP curve** as is claimed. The only FID evidence is for CIFAR-10 in Figs 2+8, where the resolution is too low to visually assess realism. Specifically, Fig 8 - the only actual RDP curve - does not clearly demonstrate the canonical tradeoff shape (e.g., at high BPPs the same FID is achieved for both high and low $\rho$ values). This, in total, amounts to a misrepresentation of the tradeoff.
- Proposition 2 (Eq. 8) appears redundant and insufficiently attributed. The tradeoff in this setting, and even in more general cases, was already derived in [5] (e.g., Sec. 3.3; see their Eq. 9). This should be cited accordingly and properly discussed. Does plugging in your assumptions on the inputs into the general case aligns with the results? As written, it looks like re-deriving a special case without acknowledging the general theory.
- The writing occasionally feels overclaiming. For example, the statement "PF-ODE yields the best perception with relatively low distortion" is somewhat misleading. What is *relatively low*? The algorithm of Theis et al. in the general case is not optimal. Specifically, in the scalar Gaussian case, they calculate the distortion for DiffC-F as $2(1-\sqrt{1-\sigma^2}$ (their eq11), which they did not recognize to be optimal (and you did). However, in the multivariate case and the more general case this is not optimal. This distinction on the conditions for each optimality are not clear throughout the paper, and imply more than what is actually proven. Please rephrase to avoid implying global MSE optimality and to clearly separate cases.
- Several important baselines are missing. Beyond not discussing [5], the experiments omit recent diffusion-based compression works such as [4,7]. In particular, I think [4] is especially theoretically relevant, as it is similar to DiffC but positions itself as predetemined-rate alternative. Including these baselines is crucial for a fair performance evaluation.
- High-resolution images for higher bitrates should be included in the appendix in large figures to allow careful visual inspection. For example, in Fig. 6 the entire right-most column looks identical at the available zoom, even though theoretically it shouldn't.
- Samples figures (e.g. 6, 12, 13) should include PSNR values per image, to allow for evaluation of distortion.
- Flux visual samples are only reported for a single BPP of 0.41. This doesn't really allow examining the **R** in the RDP tradeoff. Moreover, the provided zoom shows visually identical results across different $\rho$ values, making the DP traversal unconvincing for Flux

Overall, the paper would benefit from a more measured positioning of its novelty and a clearer acknowledgment of how it extends, rather than restates, previous results. It is important for me to mention that under the current phrasing, I would give the paper **a score of 2**, with the main reason being the missing discussion of [5]. Still, I am currently giving the paper a score of 4 under the assumption that this is easily fixable during the rebuttal phase, leaving the rest of the weaknesses as the reason for the score of 4.

Minor:
- The proof appendices are sometimes difficult to follow. For example, in Appendix A the discretization proof and the proof of Lemma 1 are not well separated. Please split the appendix into more subsections to improve readability. Also, A.4 introduces results without clear cross-references in the main text, leaving some proofs "hanging"
- For pixel-space experiments, I suggest ImageNet would be more appropriate than CIFAR. Several works (e.g., [3,4]) have already used ImageNet, and this would allow for more meaningful visual comparisons
- Please provide in the appendix full tables with numerical results for the metrics per BPP to allow for better reproducibility.
- Generally speaking, theoretical guarantees are limited to very special cases: Gaussian data, and scalar Gaussian data for full RDP optimality. The authors acknowledge this as out of scope, but it still limits the generality of the claims.
- The authors cite Tweedie's formula for "Herbert E. Robbins. An Empirical Bayes Approach to Statistics, 1992". As far as I'm aware, the proceedings are from 1956 (Proceedings of the Third Berkeley Symposium on Mathematical Statistics and Probability).

[5] Freirich, Dror, Tomer Michaeli, and Ron Meir. "A theory of the distortion-perception tradeoff in Wasserstein space." NeurIPS 2021.

[6] Blau, Yochai, and Tomer Michaeli. "The perception-distortion tradeoff." CVPR 2018.

[7] Jinpei Guo, Yifei Ji, Zheng Chen, Kai Liu, Min Liu, Wang Rao, Wenbo Li, Yong Guo, and Yulun Zhang. "OSCAR: One-Step Diffusion Codec Across Multiple Bit-rates." ArXiv preprint 2025.

**Questions:**

- Lemma 1: The authors seem to claim (L200) that the reconstruction is the MMSE itself, however the proof suggests that the mean of $Z_0$ converges to the MMSE. Can the authors clarify?
- Theorem 3 - this seems to imply a different $\rho$ parameter is required for each pixel (or PC direction?). Is this interpretation correct?
- Related - for the achievability proof - in Theis et al. work eq 13,14,15 (which are similar here) have different variance per dimension, which does not seem the case for the proof as stated here. However, the decomposition still seems similar. Don't you still need the orthogonality constraint (as in Theis' Thm 3) in the achievability proof?

---

> ### Author Response · Authors · 2025-11-21
> **Response to Reviewer 2 (Part 1)**
>
> **On the connection to (Freirich et al., 2021):** We thank the reviewer for the constructive comments and point out our oversight regarding the reference (Freirich et al., 2021) concerning DP control. We have now properly cited Freirich et al. (2021), revised the description on the proof of Proposition 2, and included a remark to discuss the connection between our results and this reference.
>
> *The modified description is also provided below.*
>
> > ``(After Proposition 2:) Proof. In Freirich et al. (2021), the authors derived the optimal DP tradeoff and the optimal estimators for multivariate Gaussian sources. Our results are equivalent to theirs in the case of AWGN degradation. The detailed closed-form in Eq. (8) and Lemma 7 in our proof offer insights into the achievability of our proposed method (Theorem 3). Therefore, we include the proof in Appendix B.2 for completeness.''
>
> > ``**Remark 1.** Although Freirich et al. (2021) have already derived the optimal DP tradeoff for the multivariate Gaussian case, they assumed the availability of two extreme point estimators (i.e., MMSE and perfect realism) and then constructed intermediate estimators via linear interpolation. In contrast, our achievability proof is constructive, as the score-scaled PF-ODE naturally provides a concrete scheme to achieve the two extremes of the DP tradeoff and offers flexible control over it.
> >
> > Our main goal is to construct a flexible lossy compression scheme. Leveraging the use of RCC, which produces a special noisy observation $Z_t=\sqrt{\bar{\alpha}_t}X+\sqrt{1-\bar{\alpha}_t}N$, our score-scaled ODE focuses on a special case of the denoising problem across various noise levels $t$. The progressive structure of the proposed ODE and the introduction of $\rho$ successfully enable traversing the distortion-perception tradeoff for any noise level $t$ using a single pre-trained model.''
>
> We would like to highlight the following key distinctions of our work:
>
> *   As discussed in Table 1 of the revised paper, Freirich et al. (2021) focus on the general denoising problem. In contrast, our main goal is to construct a comprehensive lossy compression scheme. While both DP performances are optimal at any *specific* noise level $Z_t=\sqrt{\bar{\alpha}_t}X+\sqrt{1-\bar{\alpha}_t}N$, our decoder offers an advantage in flexibility: a single model from our approach can adaptively operate across *all* varying noise levels $t$ due to the progressive structure of the ODE. Together with the introduction of $\rho$, our decoder effectively manages different compression rates and enables seamless control over the DP tradeoff using a single pre-trained model.
>
> *   Freirich et al. (2021) achieve optimal DP control by assuming the availability of two optimal estimators—one for MMSE and one for perfect perception. However, acquiring these idealized models in practice is often challenging. Their approach involved an empirical comparison, evaluating numerous existing methods and models to select the 'best' candidates for distortion and perception. In contrast, our scheme provides a concrete construction for these two extreme points (with $\rho=0$ and $\rho=1$). Furthermore, we can prove its optimality in the Gaussian case, all while utilizing only a single pre-trained model.

---

> ### Author Response · Authors · 2025-11-21
> **Response to Reviewer 2 (Part 2)**
>
> 2. **On the Perception Metric:** We thank the reviewer for this insightful comment regarding LPIPS. We acknowledge that the original DP and RDP papers (Blau and Michaeli, 2018; Blau and Michaeli, 2019) indeed treat LPIPS as a distortion metric due to its full-reference nature. However, we address this issue from two perspectives:
>
>     -   First, the original DP and RDP papers (Blau and Michaeli, 2018; Blau and Michaeli, 2019) indeed treat LPIPS as a distortion metric because it is a *full-reference* measure, requiring ground-truth and reconstruction pairs. However, in practice, LPIPS is often treated as a perceptual metric, a usage widely accepted in the literature for denoising problems (Park et al., 2023; Zhu et al., 2024), DP tradeoff (Wang et al., 2025; Zhu et al., 2025), compression problems (Yang and Mandt, 2024), and theoretical studies on perceptual metrics (Bhardwaj et al., 2020). It can be observed that LPIPS correlates more strongly with FID than with MSE or PSNR. Algorithms that achieve better LPIPS scores usually also achieve better FID scores, often at the expense of worse MSE (e.g., Figure 2 in Yang and Mandt (2024), and Appendix 2.2 in Wang (2025)). While we employ full-reference MSE and no-reference W2 distance in our theoretical analysis, we believe that in practice, LPIPS serves as a valuable perceptual metric.
>
>     -   Second, to further validate the effectiveness of our scheme, we plot the MSE-FID tradeoff on the Kodak dataset. The computation of FID on Kodak follows the configuration in DDCM, where we extract $64 \times 64$ patches from the images and compute FID on the resulting 2304 generated patches. It can be observed that our scheme exhibits a similar trend in its MSE-FID curves compared to our MSE-LPIPS curves, thereby demonstrating its effectiveness in controlling the RDP tradeoff. We have included the result in Figure 12 of the revised manuscript.
>
> ​	For the MSE-FID tradeoff at high BPPs on CIFAR-10 dataset, the reconstructions at high BPPs are expected to be similar across different $\rho$ values. When BPP is high, reconstructions have the potential to be both sharp and faithful, and the RDP tradeoff is not significant, which can also be observed in the theoretical curves in Figure 2 in the revised paper.
>
> > S. H. Park, et al., "Perception-Oriented Single Image Super-Resolution using Optimal Objective Estimation.” CVPR, 2023.
> >
> > Qiwen Zhu, et al., “Perceptual-Distortion Balanced Image Super-Resolution is a Multi-Objective Optimization Problem.” ACMMM, 2024.
> >
> > Yuhan Wang, et al., “Traversing Distortion-Perception Tradeoff using a Single Score-Based Generative Model.”, CVPR, 2025.
> >
> > Yuanzhi Zhu, et al., “OFTSR: One-Step Flow for Image Super-Resolution with Tunable Fidelity-Realism Trade-offs.”, arXiv, 2025.
> >
> > Ruihan Yang and Stephan Mandt, “Lossy Image Compression with Conditional Diffusion Models.” NeurIPS, 2023.
> >
> > Sangnie Bhardwaj, et al., “An Unsupervised Information-Theoretic Perceptual Quality Metric.” NeurIPS 2020.

---

> ### Author Response · Authors · 2025-11-21
> **Response to Reviewer 2 (Part 3)**
>
> **On Proposition 2:** We thank the reviewer for the constructive comments. As discussed in our previous response, we indeed overlooked the reference (Freirich et al., 2021). We have revised Proposition 2 and incorporated a discussion regarding the connection between our results and this reference. Given the specific noisy observation $Z_t = \sqrt{\bar{\alpha}_t}X + \sqrt{1-\bar{\alpha}_t}N$ generated by RCC, our optimality results are equivalent to those presented in (Freirich et al., 2021) for multivariate Gaussian distributions. A detailed derivation follows.
>
> Consider the source $X \sim N(0, \boldsymbol{\Sigma}\_0)$ and noisy observation $Y = \sqrt{\bar{\alpha}\_t}X + \sqrt{1-\bar{\alpha}\_t}N$.
>
> The MMSE estimator is: $X^\star = \sqrt{\bar{\alpha}\_t} \boldsymbol{\Sigma}\_0 \boldsymbol{\Sigma}\_t^{-1} Y$. This corresponds to our $\rho=0$ case.
>     The covariance matrix of the reconstruction $X^\star$ is $\boldsymbol{\Sigma}\_{X^\star} = \sqrt{\bar{\alpha}\_t} \boldsymbol{\Sigma}\_0 (\bar{\alpha}\_t \boldsymbol{\Sigma}\_0 + (1-\bar{\alpha}\_t)\mathbf{I})^{-1} \sqrt{\bar{\alpha}\_t} \boldsymbol{\Sigma}\_0 = \bar{\alpha}\_t \boldsymbol{\Sigma}\_0^2 \boldsymbol{\Sigma}\_t^{-1}$.
>
> The best perception estimator is defined as: $\hat{X}\_0 = \boldsymbol{\Sigma}\_X^\frac{1}{2} \boldsymbol{\Sigma}\_{X^\star}^{-\frac{1}{2}} X^\star = \boldsymbol{\Sigma}\_0^\frac{1}{2} (\bar{\alpha}\_t \boldsymbol{\Sigma}\_0^2 \boldsymbol{\Sigma}\_t)^{-\frac{1}{2}} \sqrt{\bar{\alpha}\_t} \boldsymbol{\Sigma}\_0 \boldsymbol{\Sigma}\_t^{-1} Y = \boldsymbol{\Sigma}\_0^\frac{1}{2} \boldsymbol{\Sigma}\_t^{-\frac{1}{2}} Y$. This corresponds to our $\rho=1$ case.
> The marginal covariance matrix of this estimator is $\boldsymbol{\Sigma}\_{\hat{X}\_0} = \boldsymbol{\Sigma}\_0^\frac{1}{2} \boldsymbol{\Sigma}\_t^{-\frac{1}{2}} \boldsymbol{\Sigma}\_t \boldsymbol{\Sigma}\_t^{-\frac{1}{2}} \boldsymbol{\Sigma}\_0^{\frac{1}{2}} = \boldsymbol{\Sigma}\_0$.
>
> The corresponding DP tradeoff can be computed as follows. The Gelbrich distance between $X$ and $X^\star$ is given by:
> $$
>     \begin{align*}(G^\star)^2 &= Tr\{\boldsymbol{\Sigma}\_0 + \bar{\alpha}\_t \boldsymbol{\Sigma}\_0^{-1} - 2 \cdot \boldsymbol{\Sigma}\_0^\frac{1}{2} \sqrt{\bar{\alpha}\_t} \boldsymbol{\Sigma}\_0 \cdot \boldsymbol{\Sigma}\_0^{-\frac{1}{2}}\} \\\\
>     &= Tr\{\boldsymbol{\Sigma}\_0 + \bar{\alpha}\_t \boldsymbol{\Sigma}\_0^{-1} - 2 \cdot \boldsymbol{\Sigma}\_0^\frac{1}{2} \sqrt{\bar{\alpha}\_t} \boldsymbol{\Sigma}\_0^{-\frac{1}{2}}\}\\\\
>     &= Tr\{(\boldsymbol{\Sigma}\_0^\frac{1}{2} - \sqrt{\bar{\alpha}\_t} \boldsymbol{\Sigma}\_0\boldsymbol{\Sigma}\_t^{-\frac{1}{2}})^2\}
>     \end{align*}
> $$
> Assuming the covariance matrix admits an eigenvalue decomposition $\boldsymbol{\Sigma}\_0 = \mathbf{Q} \boldsymbol{\Lambda}\_0 \mathbf{Q}^T$. Then, we have $\boldsymbol{\Sigma}\_t = \mathbf{Q}(\bar{\alpha}\_t \boldsymbol{\Lambda}\_0 + (1-\bar{\alpha}\_t)\mathbf{I})\mathbf{Q}^T$. The perception level achieved by the MMSE estimator is then:
> $$
>    \begin{align*}
>     (G^\star)^2 = &Tr\{(\mathbf{Q} (\boldsymbol{\Lambda}\_0^{\frac{1}{2}} - \sqrt{\bar{\alpha}\_t} \boldsymbol{\Lambda}\_0 (\bar{\alpha}\_t \boldsymbol{\Lambda}\_0 + (1-\bar{\alpha}\_t) \mathbf{I})^{-\frac{1}{2}}) \mathbf{Q}^T)^2\}\\\\
>     =  & Tr\{\mathbf{Q} (\dots)^2 \mathbf{Q}^T\} \\\\
>     =  & Tr\{(\boldsymbol{\Lambda}\_0^{\frac{1}{2}} - \sqrt{\bar{\alpha}\_t} \boldsymbol{\Lambda}\_0 (\bar{\alpha}\_t \boldsymbol{\Lambda}\_0 + (1-\bar{\alpha}\_t) \mathbf{I})^{-\frac{1}{2}})^2\}  \\\\
>     = &\sum\_{i=1}^d \Big(\sqrt{\lambda\_i} - \sqrt{\bar{\alpha}\_t} \frac{\lambda\_i}{\sqrt{\bar{\alpha}\_t \lambda\_i + (1-\bar{\alpha}\_t)}}\Big)^2 \\\\
>     = &\sum\_{i=1}^d \frac{\lambda\_i}{\lambda\_i^{(t)}} \big(\sqrt{\lambda\_i^{(t)}} - \sqrt{\bar{\alpha}\_t} \sqrt{\lambda\_i}\big)^2,
>     \end{align*}
> $$
> where $\lambda\_i$ denotes the $i$-th diagonal value of $\boldsymbol{\Lambda}\_0$ and $\lambda\_i^{(t)}= \bar{\alpha}\_t\lambda\_i + (1-\bar{\alpha}\_t)$.
>
> Thus, according to Theorem 1 in (Freirich et al., 2021), the DP tradeoff is given by:
>    $$
>     D(P) = D^\star + (G^\star - \sqrt{P})^2 = \Big(\sqrt{\sum\_{i=1}^d \frac{\lambda\_i}{\lambda\_i^{(t)}} \big(\sqrt{\lambda\_i^{(t)}} - \sqrt{\bar{\alpha}\_t} \sqrt{\lambda\_i}\big)^2} - \sqrt{P}\Big)^2 + \sum\_{i=1}^d \frac{(1-\bar{\alpha}\_t)\lambda\_i}{\lambda\_i^{(t)}},
>    $$
> This result aligns with our Proposition 2.
>
>  We have appropriately cited Freirich et al. (2021) in the revised version of our manuscript. We apologize for this oversight. Furthermore, we emphasize that our achievability proof is constructive, as the proposed score-scaled ODE provides a concrete scheme to achieve both extremes of the DP tradeoff and offers flexible control over it.

---

> ### Author Response · Authors · 2025-11-21
> **Response to Reviewer 2 (Part 4)**
>
> **Rephrasing to avoid overclaiming** We thank the reviewer for the valuable comments. We have proofread the manuscript and revised the description to avoid overclaiming. For instance, we have adjusted the previously mentioned statement to: "PF-ODE yields the best perception with relatively low distortion *when the input is corrupted by Gaussian noise*."
>
>  Regarding the optimality of DiffC-F, Theis et al. (2022) did not show that the overall scheme is optimal in achieving the RDP tradeoff. However, specifically for the DP tradeoff, they established a strong result concerning the optimality of the ODE decoder. They showed that, for input corrupted by Gaussian noise and under certain restrictions on the source, the ODE decoder provides an optimal reconstruction. More precisely, the output of the ODE decoder $\hat{X}_F$, when given $Z_t$ as an initial condition, achieves the lowest MSE under the perfect realism extreme, i.e., $\mathbb E[\|\hat{X}_F-X\|^2]\leq \mathbb E[\|\hat{X}^\prime-X\|]$ for any possible reconstruction $\hat{X}^\prime$ that satisfies the conditions $X-Z_t-\hat{X}^\prime$ (implying a Markov chain) and $\hat{X}^\prime\sim X$.
>
>  **Additional Baseline:** We thank the reviewer for their constructive feedback. In response, we have included a comparison with DDCM (Ohayon et al., 2025) in the revised version of the manuscript. We use their official implementation and the same Stable Diffusion 2.1 for comparison. The corresponding BPP-MSE, BPP-LPIPS, and RDP curves, which now incorporate these results, have been updated in Figures 5 and 6 in the revised manuscript.
>
> **High-Resolution Images for Higher Bitrates** We thank the reviewer for the valuable feedback. When BPP is high, reconstructions are both sharp and faithful across all $\rho$ values, and the RDP tradeoff is not significant, which can also be observed in the numerical curves in Figure 6 of the revised paper. We have added more high-resolution images for higher bitrates (including Flux with different $\rho$ and $t$ and benchmarks) on https://diffrdp.github.io/ to depict the detailed and vivid changes controlled by $\rho$ and $t$.
>
> **Revision on Figures:** We thank the reviewer for the helpful suggestion. In response, we have updated Figures 7, 14, and 15 in the revised version to include the PSNR values.
>
> **High Resolution Reconstructions for Flux:** We thank the reviewer for the valuable feedback. In Figure 6 of the original manuscript, we only illustrate the significant changes at low and medium BPPs. The reconstructions at high BPPs are expected to be similar across different $\rho$ values. We have added samples from Flux with different $t$ and $\rho$ in Figure 16 in Appendix E.2.2 of the revised version. We also provided more high-resolution images (including Flux visual samples with different $t$) on https://diffrdp.github.io/ to depict the detailed and vivid changes controlled by $\rho$ and $t$.

---

> ### Author Response · Authors · 2025-11-21
> **Response to Reviewer 2 (Part 5)**
>
> **Response to minor weakness:**
>
> 1.  We thank the reviewer for the valuable feedback. In the revised version, we have modified the structure and some details in Appendices A-D. Specifically, the changes are summarized as follows:
>     *   We put the discretization of score-scaled PF-ODE part to a separated appendix section (now beome Appendix A). The proof of Lemma 1 is now separately discussed in Appendix B in the revised version.
>     *   We included a discussion of two extreme cases ($\rho=0$ and $\rho=1$) in Lemma 1 and Appendix B.3 in the revised manuscript to better match the results between the main text and the proof, and provide more insights into the ODE reconstruction.
>     *   We revised Appendix D in the revised manuscript (Proof of Theorem 4) to include more details regarding the one-shot coding results.
> 2.  Due to the time limit, we were unable to include ImageNet experiments during the rebuttal phase. However, as discussed in previous responses, we have added a new benchmark (DDCM), a new metric (FID), and more high-resolution images on https://diffrdp.github.io/ to better validate the effectiveness of our scheme.
> 3.  We have added full tables with numerical results for each metric per BPP in Appendix E.2.2 (See Table 3 and Table 4) in the revised paper to enhance reproducibility.
> 4.  In general cases, it is indeed difficult to derive closed-form expressions due to the complex structure of the source distribution. Intuitively, we can observe that the variance of the reconstruction distribution depends on $\rho$. When $\rho$ is small, the variance is small, leading to better distortion. When $\rho$ is large, the variance is large, leading to better perception. We illustrate this phenomenon using a mixture of Gaussian sources where $p_X = w_1*\mathcal{N}(u_1, \sigma_1^2) + w_2*\mathcal{N}(u_2, \sigma_2^2)$ and $Y=\sqrt{\bar{\alpha}_t}X + \sqrt{1-\bar{\alpha}_t}N$ with $N\sim\mathcal{N}(\mathbf 0, \mathbf I)$. The effect of $t$ and $\rho$ on the sampling trajectories of ODE for a mixture Gaussian source is shown in Figure R1 in this link https://mixturegaussianfig.github.io/.
> 5. We have corrected the reference for Tweedie's formula in the revised version.
>
> **Response to Question 1:** When $\lambda \to 0$, the variance of the conditional distribution $p_{Z_0|Z_t}$ converges to zero. Consequently, the ODE reconstruction becomes a deterministic function of $Z_t$, specifically the mean of $p_{Z_0|Z_t}$.
>
> **Response to Question 2:** Yes. Here we use a similar technique as in Theis et al. (2022) when they proved their Theorem 3.
>
> **Response to Question 3:** Regarding the first point, Theis et al. (2022) employ distinct noise schedules across different dimensions to enhance rate-distortion performance (analogous to the water-filling solution presented in Qian et al. (2025) during their derivation of the RDP function for the multivariate Gaussian case). However, when our focus is solely on the DP aspect (as is the case in our Theorem 3), all dimensions utilize the same noise schedule, i.e., $Z_t=\sqrt{\bar{\alpha}_t}X+\sqrt{1-\bar{\alpha}_t}N$, where $N\sim\mathcal{N}(\mathbf 0, \mathbf I)$. For the second part, we assume the covariance matrix $\Sigma_0$ admits an eigen-decomposition with positive eigenvalues.
>
> > References:
> >
> > Dror Freirich, et al., “A theory of the distortion-perception tradeoff in wasserstein space.” NeurIPS, 2021.
> >
> > Yochai Blau and Tomer Michaeli, “The perception-distortion tradeoff.” CVPR, 2018.
> >
> > Yochai Blau and Tomer Michaeli, “Rethinking lossy compression: The rate-distortion-perception tradeoff.” ICML, 2019.

---

> > ### Comment · Reviewer_y2u7 · 2025-11-26
> >
> > **Denoising:** I appreciate the authors effort in changing the paper. However, using the term denoising to describe all restoration or inverse problems is a bit unclear. Inverse problems need not have noise (e.g. inpainting), the input and reconstruction doesn't even need to be of the same dimension. the DP tradeoff still applies.
> >
> > **LPIPS as a perception metrics:** This is a delicate and very important point which many papers sadly miss.
> > There's a distinction between the word "perception"/"perceptual", which relates to the semantic of and image, and "perception" as defined in the DP and RDP papers. As defined in the original papers, and you yourself repeat in eq 1, perception is the **distance between the distributions** of the reconstruction and the input, i.e. $d(p_X, p_\hat{X})$, while distortion is the per-sample distance, averaged, i.e. $\mathbb{E}[\Delta(X,\hat{X})]$. The latter can be taken in any space, be it pixel-space (MSE) or a feature-space (LPIPS).
> > In restoration works, its common to report MSE, LPIPS, and FID, since LPIPS complements MSE - yet it does not replace FID. Specifically, LPIPS cannot measure the distance between the distributions - since it strictly compares 2 samples. For example, a method can "restore" (or "decompress") images by simply synthesizing completely new images. These new images will be drawn from the image-distribution, so $P=0$ - meaning they are completely realistic, however LPIPS would flag them with a value $>0$, since they are *distorted* from their compared GT.
> > The entire DP theory relies on $P$ to be measured in distance between distributions - like W2 (which you used) does - so LPIPS just doesn't fit here. It is indeed, as you say, a valuable metric - just a **perceptual distortion** metric, not a **perception** metric.
> > Therefore, Figs 4, 5, 6, 8, and the new 11, 13 all report **distortion-distortion curves, and not perception-distortion curves**.
> >
> > *About the actual RDP curves:* The RDP curve for CIFAR in Fig 9 is not explained by your explanation. High BPPs, such as the orange curve - show a high FID for both small and large $\rho$. It shows a parabolic curve.
> > Interestingly, this phenomenon disappears in Fig. 12 for Kodak using SD2.
> >
> > **High-Resolution and Flux performances:** Thank you for adding the results. However, I remain unconvinced regarding Flux. In fig 16, setting $\rho=1$ yields large visual distortion which implies $P\neq0$. Why is that? Also, it seems that just setting different $\rho$ values e.g. in the demo for $t=99$ doesn't really change the distortion visually? It just seems to be perturbations of the same degraded quality. This might also explain the weird RDP curves of Flux.
> >
> > **Question 2:** From my understanding, Theis et al however had different parameters per dimensions? This if of course more general (while still limited)

---

> ### Author Response · Authors · 2025-12-02
> **Further Response to Reviewer 2 (Part 1)**
>
> We thank the reviewer for the prompt response. We have further revised the paper in response to the feedback, and we would like to highlight the following points.
>
> **1. Denoising:** We have updated the term "**Denoising**" to "**Image Restoration**" in Table 1 to more accurately reflect the problem considered in the corresponding references. In our main results, our score-scaled ODE decoder specifically addresses a denoising problem generated by the RCC algorithm, represented by $Z_t \sim \sqrt{\bar{\alpha}_t}X + \sqrt{1-\bar{\alpha}_t}N$.
>
> **2. LPIPS as a perception metric:** We thank the reviewer for further clarification. First, we maintain that LPIPS is a reasonable metric for perceptual quality. This is not only supported by numerous existing works (as listed in our previous response) but also aligns well with human perception (and consequently, with FID). Referring to the original definition of perceptual quality by Blau and Michaeli (2018, 2019), it describes the extent to which a sample is perceived by humans as a valid (natural) image. The divergence between distributions has been **shown** to correlate well with human opinion scores (Mittal et al., 2012). *In theory*, utilizing the divergence between the generated and original distributions is an effective approach. However, _in practice_, the high-dimensional distribution of image datasets remains inaccessible. Therefore, we must rely on metrics that align well with human perception, such as FID or LPIPS. FID computes the mean and variance in the feature space of two sets of images and then calculates the Fréchet distance, assuming both sets are drawn from two multidimensional Gaussian distributions. Although FID is a no-reference metric, it entirely ignores the shape of the distribution. LPIPS computes the distance between features of two images. These two images can differ significantly at the pixel level (e.g., two cat images with distinct details) yet still possess similar features. It is observed that LPIPS correlates more strongly with FID than with MSE or PSNR. Algorithms that achieve superior LPIPS scores typically also yield better FID scores, often at the expense of worse MSE (e.g., Figure 2 in Yang and Mandt (2024) and Appendix 2.2 in Wang (2025)).
>
> Second, acknowledging that this may be a debatable topic discussed across various papers, **we include RDP curves using both LPIPS and FID as perceptual metrics**, and both sets of results demonstrate valid RDP tradeoffs.
>
> Regarding the DP curves for CIFAR in Fig. 9, as discussed in the paper and our previous response, the RDP tradeoff is less significant for high bpp regions. The traversed FID intervals for high bpps (e.g., t=10, 20, 30 in Fig. 9) are much shorter than those for low bpps (e.g., t=780, 498, etc.). We want to emphasize that when bpp is low, there is room to control the DP balance, whereas for high bpp regions, we can expect the algorithm to produce samples with both low MSE and good perceptual quality. Figure 10 illustrates that the difference between reconstructions for different $\rho$ values and $t=10$ is barely observable. The observed parabolic curve may come from the following factors. First, FID computation on small-sized images, such as CIFAR-10, may not accurately capture the true perceptual quality. Second, since the score network is merely a learned approximation of the true score, the valid range of $\rho$ for controlling DP may not precisely be [0,1].
>
> **3. On Flux performance:** As can be observed in Fig. 6 and Fig. 12, the Flux model exhibits a worse DP curve than SD-2.1 within our framework, but it traverses a larger bpp region. This indicates its potential to achieve much lower MSE and better perceptual quality. We report the results from both models, and the choice of model can be guided by practical needs.
>
> Setting $\rho=1$ for $t=498$ in Fig. 16, the reconstruction appears sharper and more vivid than that with $\rho=0.75$. Furthermore, it numerically yields lower FID and LPIPS scores. It is challenging for the model to achieve perfect perceptual quality ($P=0$) in low bpp regions, especially when considering the interplay with distortion requirements (a similar observation holds for SD2.1 and other benchmarks).
>
> For the Flux model in the demo, $t=99$ already corresponds to a relatively high bpp, thus the changes in distortion and perception are not significant. However, it can still be observed that $\rho=0.75$ produces smoother details than $\rho=1$.

---

> ### Author Response · Authors · 2025-12-02
> **Further Response to Reviewer 2 (Part 2)**
>
> **4. Further response to Question 2:** When considering Theorem 3 in Theis et al. (2022) and our Theorem 3 (which pertains to the *denoising task with ODE, not the entire compression task*), both works assume the same observation with a uniform noise level across all dimensions. Additionally, both employ a similar technique of decomposing the ODE per-dimension. As quoted from Theorem 3 in Theis et al. (2022):
>
> >... Define $\mathbf Z_t=\sqrt{1-\sigma_t^2}\mathbf{X}+\sigma_t\mathbf{U}$, where $\mathbf U\sim\mathcal{N}(0, \mathbf I).$ If $\hat{\mathbf{X}}_F=\hat{\mathbf{Z}}_0$ is the solution to the ODE in Eq. (6) given $\mathbf{Z}_t$...
>
> It can be observed that the observation $\mathbf Z_t=\sqrt{1-\sigma_t^2}\mathbf{X}+\sigma_t\mathbf{U}$ has the same noise level across dimensions. In Sections 4.1 and 4.2 (not directly reflected in Theorem 3), Theis et al. (2022) utilize different noise parameters (e.g., $Z_{t,i}=\sqrt{1-\gamma_i^2}X_i+\gamma_i\sqrt{\lambda_i}U_i$) to demonstrate the potential for further reducing the *compression rate*.
>
> Similar to the original DiffC framework, in practice, our algorithm has the potential to achieve better RDP curves by adapting and fine-tuning dimension-specific noise levels and $\rho$ values.
>
> > Anish Mittal, et al., “No-Reference Image Quality Assessment in the Spatial Domain.” IEEE Trans. on Image Processing, 2012.

---

### Official Review · Reviewer_Vzta · 2025-11-10

**Soundness:** 2
**Presentation:** 2
**Contribution:** 2
**Rating:** 2
**Confidence:** 5

**Summary:**

The paper considers the rate-distortion-perception tradeoff using pre-trained diffusion-based models. It provides some results for Gaussian case and implements some experiments on a few datasets to explore the RDP tradeoff.

**Strengths:**

The paper studies an interesting problem of rate-distortion-perception tradeoff.

**Weaknesses:**

I believe that the paper has several major technical issues, and its contributions are not sufficiently compelling when compared to prior work such as Blau & Michaeli, Salehkalaibar et al., and Zhang et al.

1) The theoretical results lack sufficient rigor. For example, in the proof of Theorem 4, a reverse channel coding encoder is paired with a score-based decoder, where the central challenge is to establish a one-shot result for this encoder-decoder combination. However, the proof simply assumes that such a one-shot result holds, without justification. This assumption is non-trivial, and its validity is unclear for this specific encoder-decoder pair. Consequently, the subsequent simplification to the Gaussian case also lacks foundation in the absence of a rigorous one-shot argument.

2) Proposition 2 is not technically sound. Even at a fixed rate, the distortion-perception tradeoff does not admit the simple closed-form expression claimed in Eq. (8) (see Qian et al. for reference). Moreover, Proposition 2 entirely omits the compression rate, which contradicts the paper’s stated goal of analyzing the rate-distortion-perception (RDP) tradeoff. A result that ignores rate is not meaningful in this context.

3) The paper does not provide compelling theoretical or experimental insights. In particular, the experimental section only plots RDP curves without offering deeper analysis or qualitative insight into reconstruction characteristics, perceptual quality, or practical behavior. There is no clear experimental contribution beyond reproducing expected tradeoff plots.

4) The proofs in the appendix are difficult to follow and, based on the issues noted above, appear to contain technical gaps. The overall presentation of the theoretical arguments would benefit from significant revision for correctness and clarity.

5) The title of the paper claims a “training-free” method. However, while the encoder may be pretrained, the decoder still requires training. Therefore, the “training-free” characterization is misleading and should be reconsidered.

**Questions:**

what are the main contributions of the paper when diffusion-based models are used?

---

> ### Author Response · Authors · 2025-11-21
> **Response to Reviewer 1 (Part 1)**
>
> **1. On the Rigor of Theorem 4:** We thank the reviewer for the comment. We would like to emphasize that, similar to DiffC (Theis et al., 2022), our encoder is a conventional rate-constrained channel (RCC) encoder that produces a codeword $M$. The decoder consists of two parts: first, a conventional RCC decoder that produces samples of $Z_t=\sqrt{\bar{\alpha}_t}X+\sqrt{1-\bar{\alpha}_t}N$ based on $M$; and second, a score-based ODE decoder that restores $\hat{X}^\rho$ based on $Z_t$ and the chosen $\rho$.
>
>  Thus, up until the construction of $Z_t$, our scheme is identical to the conventional reverse channel coding problem, and the one-shot achievability result follows Li & Gamal (2018). Specifically, leveraging the strong functional representation lemma (Li & Gamal, 2018), the bound on $H(M)$ can be derived from the cross-entropy between the distribution of $M$ and the Zipf distribution $\text{Zipf}(1+1/(I(X;Z_t)+1))$. This bound is given by $I(X;Z_t)+\log_2 (I(X;Z_t)+1)+4$ bits. In the Gaussian case, where $X\sim \mathcal{N}(\mu_0, \sigma_0^2)$, we have $I(X;Z_t)=\frac{1}{2}\log\Big(\frac{\bar{\alpha}_t}{1-\bar{\alpha}_t}\sigma_0^2+1\Big)$. After producing $Z_t$, the achievable distortion and perception levels are given by Eqs. (16) and (17). We thank the reviewer for this comment and have modified Appendix D in the revised paper (Proof of Theorem 4) to include more details.
>
> **2. On Proposition 2:** We thank the reviewer for the comment. The _rate-distortion-perception function_ in the multivariate Gaussian case does not admit a closed-form expression according to Qian et al. (2025), as the rate allocation among different dimensions is non-trivial, necessitating a generalized water-filling method introduced by Qian. However, the _distortion-perception tradeoff_ does have a closed-form expression (Freirich et al., 2021). The RCC creates samples of $Z_t=\sqrt{\bar{\alpha}_t}X+\sqrt{1-\bar{\alpha}_t}N$. Thus, the score-based ODE decoder is designed to solve a specific denoising problem. We include these results because Proposition 2 and Theorem 3 demonstrate the optimality of our score-scaled ODE decoder for DP-control, which is an important but missing aspect in previous DiffC schemes. We have included a remark to better position the DP results within the whole compression framework and discuss their connection to previous DP results.
>
> The modified description is also shown below.
>
>  > **Remark 1.** Although Freirich et al. (2021) have already derived the optimal DP tradeoff for the multivariate Gaussian case, they assumed the availability of two extreme point estimators (i.e., MMSE and perfect realism) and then constructed intermediate estimators via linear interpolation. In contrast, our achievability proof is constructive, as the score-scaled PF-ODE naturally provides a concrete scheme to achieve the two extremes of the DP tradeoff and offers flexible control over it.
>  >
>  > Our main goal is to construct a flexible lossy compression scheme. Leveraging the use of RCC, which produces a special noisy observation $Z_t=\sqrt{\bar{\alpha}_t}X+\sqrt{1-\bar{\alpha}_t}N$, our score-scaled ODE focuses on a special case of the denoising problem across various noise levels $t$. The progressive structure of the proposed ODE and the introduction of $\rho$ successfully enable traversing the distortion-perception tradeoff for any noise level $t$ using a single pre-trained model.''
>
> **3. More Quantitative Discussion of Experimental Results:** We appreciate the reviewer's constructive comments. We have modified the paper to provide further insights and practical discussion regarding our scheme. The modifications are summarized below:
>
>  - We have incorporated discussions in the manuscript to demonstrate how the proposed scheme can flexibly adjust the compression rate and distortion-perception tradeoff based on physical restrictions (e.g., channel conditions in wireless communication) or user preferences. The added description in Section 5 is also provided below:
>
>    > ``_Our method provides an efficient way to seamlessly control compression rate and distortion-perception tradeoff with one pre-trained model and two parameters $t$ and $\rho$. In practice, users can select a suitable bitrate according to resource restrictions or physical wireless channel conditions and adjust the distortion-perception balance according to their specific needs without retraining._''
>
>  - We have included more high-resolution images on the http://diffrdp.github.io/ to illustrate the detailed and vivid changes controlled by $\rho$ and $t$.
>
>  - We have also added a table in Appendix E.2.2 in the revised manuscipt to compare the latency and memory of our scheme against benchmarks and to discuss practical issues in real-world deployment.

---

> ### Author Response · Authors · 2025-11-21
> **Response to Reviewer 1 (Part 2)**
>
> **4. Revisions to Appendices:** We thank the reviewer for the valuable feedback. In the revised version, we have modified the structure and some details in the appendices. Specifically, the changes are summarized as follows:
>
>  *   We have split Appendix A to provide a clearer explanation of the ODE discretization and the proof of Lemma 1. We put the discretization of score-scaled PF-ODE part in a separate appendix section (now Appendix A). The proof of Lemma 1 is now separately discussed in Appendix B.
>  *   We have included a discussion of two extreme cases ($\rho=0$ and $\rho=1$) in Lemma 1 and Appendix B.3 in the revised manuscript to offer more insights into the ODE reconstruction.
>  *   We have revised Proof of Theorem 4 to include more details regarding the one-shot coding results.
>
> **5. Training-Free Property of the Decoder:** We appreciate the reviewer's comments. As previously discussed, our scheme comprises three main components: an RCC encoder, an RCC decoder, and a score-scaled ODE decoder. A key advantage is that all three modules can be implemented using a single pre-trained diffusion model (e.g., Stable Diffusion) without requiring any additional training.
>
>  More specifically:
>
>  *   For the RCC encoder and decoder pair, the pre-trained model serves as a reference distribution from which to sample $p_{\theta}(\mathbf{z}_k|\mathbf{z}_{k-1})$.
>  *   Within the ODE decoder, the pre-trained model is utilized to compute the score $\nabla \log p_{Z_{k}}(\mathbf{z}_k)$, which enables the reverse sampling described in Eq. (6).
>
>  With a single pre-trained model, we can cover a wide range of RDP tradeoffs, thereby saving both training time and model storage costs. For instance, to achieve coverage for 10 different bitrates and 5 distinct distortion-perception tradeoffs, existing methods like HiFiC or CDC need training and storing 50 distinct models. This would result in an approximate total storage requirement of 2.5-9GB, which is much larger than that of our single-model approach.
>
> **On the Novelty of Our Scheme:** Our scheme introduces a novel approach that leverages a single pre-trained diffusion model to efficiently and seamlessly control both the compression rate and the distortion-perception tradeoff using just two parameters.
>
> - Theoretically, we prove the optimality of our score-scaled ODE and the overall compression framework in governing DP and RDP tradeoffs, specifically within the Gaussian setting.
>
> - From a practical standpoint, we demonstrate the scheme's effectiveness on high-resolution images by integrating it with existing state-of-the-art pre-trained models, such as Stable Diffusion and Flux.
>
> We have revised the introduction part to better clarify the novelty of our method.

---

### Author Response · Authors · 2025-11-21
**General Response:**

We thank all reviewers for their constructive and helpful comments. We have revised the paper in response to their feedback. Key revisions are summarized below:

- We have added a figure to illustrate our framework (see Figure 1 in the revised paper). The score-scaled PF-ODE decoder is proposed to control the DP tradeoff within a specific denoising problem, characterized by the observation $Z_t=\sqrt{\bar{\alpha}_t}X+\sqrt{1-\bar{\alpha}_t}N$ at any noise level $t$. The overall framework is designed to control the RDP tradeoff by adjusting the noise level $t$ (i.e., compression rate) and $\rho$ (i.e., DP tradeoff) using a single pre-trained diffusion model.

- We present a comparison of related works concerning denoising and lossy compression problems, focusing on DP and RDP tradeoffs, in Table 1 of the revised manuscript. Given that the results on DP and RDP tradeoffs are quite distinct and easily confused, we also provide a table summarizing existing results on DP and RDP, as shown below.

> Table R1: A summary of theoretical results of DP and RDP tradeoffs.

|         | Type                    | Specific Results                                             |
| --------------- | --------------------------- | ------------------------------------------------------------ |
| DP Tradeoff  | Closed-Form Solution | Scalar Gaussian with KL divergence, MSE, and linear estimator (Blau & Michaeli, 2018) |
|                 |                             | Multivariate Gaussian with W2 distance and MSE (Freirich et al., 2021) |
| RDP Function | Closed-Form Solution | Binary sources with TV divergence and Hamming distortion (Blau & Michaeli, 2019) |
|                 |                             | Scalar Gaussian with W2 distance / KL divergence and MSE (Zhang et al., 2021) |
|                 |                             | Multivariate Gaussian with W2 distance / KL divergence and MSE solved by water-filling (Qian et al., 2025) |
|                 | Coding Theorem           | One-shot and asymptotic achievability with common randomness (Theis & Wagner, 2021) |


- We have revised the description of the proof of Proposition 2 and included a remark to better contextualize our DP results within the overall compression framework and discuss their connection to previous DP results (Freirich et al., 2021), as reviewers highlighted.

- We have conducted experiments to compare the latency of our method with other baselines. The detailed results are shown in the table below. Notably, the comparison between the original DiffC and other benchmarks was already provided in Vonderfecht & Liu (2025), and we reproduce the results on our devices.

| Method       | # of Parameters | Encoding (s) | Decoding (s) | Overall Time (s) |
| :----------- | --------------: | -----------: | -----------: | ---------------: |
| HiFiC        |            181M |         0.67 |         1.53 |             2.20 |
| CDC          |           53.8M |         0.07 |         3.25 |             3.32 |
| Ours (SD2.1) |            950M |    0.22-9.14 |    0.33-2.10 |        2.31-9.47 |
| DDCM (SD2.1) |            950M |        37.76 |        37.91 |            75.67 |

- Following the reviewers' suggestions, we include a new benchmark, DDCM (Ohayon et al., 2025), and report the MSE-FID curves of the proposed scheme and baselines on the Kodak dataset, as suggested. We have also included more high-resolution images on https://diffrdp.github.io/ to depict the detailed and vivid changes controlled by $\rho$ and $t$.

The detailed responses to reviewers are referred to the following official comments. The major revised parts are marked in blue.

Guy Ohayon, et al., "Compressed image generation with denoising diffusion codebook models." ICML, 2025.

---

### Meta-Review · Area_Chair_Jo7H · 2025-12-28

**Summary:**

This submission received **mixed but overall below-threshold initial scores**, with one reject (Reviewer Vzta: overall), multiple marginal rejects (Reviewers y2u7, 63wi, Z2xu), and two borderline accepts (Reviewers D1im). Importantly, no clear consensus emerged in favor of acceptance.

While reviewers acknowledge the importance of studying the rate–distortion–perception (RDP) tradeoff and find the idea of leveraging a single pre-trained diffusion model, they raised fundamental concerns about (i) theoretical rigor and correctness, (ii) overstated novelty relative to prior work, and (iii) the validity and interpretation of the empirical RDP evaluation, especially the use of LPIPS as a perception metric.

Despite an extensive rebuttal and multiple revisions, core disagreements remain unresolved, particularly regarding theoretical soundness and proper positioning within existing RDP theory. Based on the initial scores and the remaining unresolved concerns, my judgment as AC is to **Reject**.

**Reviewer Concerns:**

**Concerns partially addressed by the rebuttal:**

* **Clarity, presentation, and missing baselines** were improved by adding framework figures, latency tables, additional comparisons, and new appendices
  *(Reviewers y2u7, 63wi, D1im)*.

* **Attribution and positioning relative to Freirich et al. (2021)** were corrected by adding citations
  *(Reviewer y2u7)*.

* **Latency and training-free claims** were clarified with newexperiments and explanations of storage tradeoffs across multiple bitrates
  *(Reviewers 63wi, D1im, Z2xu)*.


**Concerns that remain outstanding:**

* **Novelty relative to prior RDP theory**:
  Reviewer y2u7 emphasizes that the key idea, interpolating between MMSE and perfect-perception extremes, was already formalized in prior work (e.g., Freirich et al., 2021), and that Proposition 2 appears largely redundant. While citations were added, reviewers still question whether the contribution is incremental rather than substantive.

* **Validity of empirical RDP evaluation**:
  Reviewer y2u7 strongly objects to using LPIPS as a perception metric, arguing that it measures distortion rather than distributional realism, and that many reported curves are therefore distortion–distortion rather than true RDP curves. Follow-up discussions indicate this disagreement remains unresolved.

* **Practical motivation and efficiency**:
  Reviewers 63wi and Z2xu remain unconvinced that the method offers a compelling advantage over existing codecs (e.g., HiFiC) in latency, model size, or deployment practicality, even after additional experiments.

* **Explicit concerns raised during the discussion**:
Reviewer y2u7 remains unconvinced by the reported high-resolution and Flux performance. Reviewer Z2xu is not convinced that the proposed method outperforms HiFiC in terms of either model size or latency, finds the theoretical analysis difficult to follow, and remains unconvinced by the authors’ claims.

**Reviewer Scores:**

* **Reviewer Vzta (initial: 2)** → **Likely  Unchanged**
  Strong confidence; maintains objections on theoretical soundness and misleading claims.

* **Reviewer y2u7 (initial: 4)** → **Likely unchanged**
  Acknowledges revisions but maintains concerns about novelty and misuse of LPIPS in RDP evaluation.

* **Reviewer 63wi (initial: 4)** → **Likely unchanged**
  Clarifications on latency and storage help, but novelty and efficiency concerns remain.

* **Reviewer Z2xu (initial: 4)** → **Unchanged**
  Explicitly states that new experiments do not change their assessment.

* **Reviewer D1im (initial: 6)** → **Possibly unchanged**
  Generally positive, but concerns about inference cost and ablation remain; unlikely to shift overall outcome.

---

### Decision · Program_Chairs · 2026-01-26

Reject